# Unbiased constrained sampling with
# Self-Concordant Barrier Hamiltonian Monte Carlo

**Maxence Noble**[*]
CMAP, CNRS, École polytechnique,
Institut Polytechnique de Paris,
91120 Palaiseau, France

**Valentin De Bortoli**
Computer Science Department,
ENS, CNRS, PSL University

**Alain Oliviero Durmus**
CMAP, CNRS, École polytechnique,
Institut Polytechnique de Paris,
91120 Palaiseau, France

## Abstract

In this paper, we propose Barrier Hamiltonian Monte Carlo (BHMC), a version of the HMC algorithm which aims at sampling from a Gibbs distribution $\pi$ on a manifold M, endowed with a Hessian metric $\mathfrak{g}$ derived from a self-concordant barrier. Our method relies on Hamiltonian dynamics which comprises $\mathfrak{g}$. Therefore, it incorporates the constraints defining M and is able to exploit its underlying geometry. However, the corresponding Hamiltonian dynamics is defined via non separable Ordinary Differential Equations (ODEs) in contrast to the Euclidean case. It implies unavoidable bias in existing generalization of HMC to Riemannian manifolds. In this paper, we propose a new filter step, called "involution checking step", to address this problem. This step is implemented in two versions of BHMC, coined continuous BHMC (c-BHMC) and numerical BHMC (n-BHMC) respectively. Our main results establish that these two new algorithms generate reversible Markov chains with respect to $\pi$ and do not suffer from any bias in comparison to previous implementations. Our conclusions are supported by numerical experiments where we consider target distributions defined on polytopes.

## 1  Introduction

Markov Chain Monte Carlo (MCMC) methods is one of the primary algorithmic approaches to obtain approximate samples from a target distribution $\pi$. They have been successively applied over these past decades in a large panel of practical settings, (Liu & Liu, 2001). In particular, gradient-based MCMC methods have shown their efficiency and robustness in high-dimensional settings, and come nowadays with strong theoretical guarantees, (Dalalyan, 2017; Durmus & Moulines, 2017). However, they still struggle in facing the case where the target distribution is supported on a *constrained* subset M of $\mathbb{R}^d$, (Gelfand et al., 1992; Pakman & Paninski, 2014; Lan & Shahbaba, 2015). Yet, this problem appears in various fields; see e.g., (Morris, 2002; Lewis et al., 2012; Thiele et al., 2013) with some important applications in computational statistics and biology.

Drawing samples from such distributions is indeed a challenging problem that has been intensively studied in the literature, (Dyer & Frieze, 1991; Lovász & Simonovits, 1993; Lovász & Kannan, 1999; Lovász & Vempala, 2007; Brubaker et al., 2012; Cousins & Vempala, 2014; Pakman & Paninski, 2014; Lan & Shahbaba, 2015; Bubeck et al., 2015). In particular, some recent extensions of the

---

[*]Corresponding author. Contact at: maxence.noble-bourillot@polytechnique.edu

37th Conference on Neural Information Processing Systems (NeurIPS 2023).

popular Metropolis-Hastings (MH) algorithm to constrained spaces consist in designing proposals based on dynamics for which $\pi$ is invariant, (Zappa et al., 2018; Lelièvre et al., 2019, 2022). In practice, the corresponding solutions are numerically integrated based on implicit and symplectic schemes, which however come with additional difficulties. As first observed by Zappa et al. (2018), an additional "involution checking step" in the usual MH filter is necessary to ensure that the resulting Markov kernel admits $\pi$ as a stationary distribution.

In this paper, we aim at generalizing this family of methods by taking into account the geometry of the constrained subspace, building on the Riemannian Manifold Hamiltonian Monte Carlo (RMHMC) algorithm introduced by Girolami & Calderhead (2011). Similarly to HMC, (Duane et al., 1987; Neal et al., 2011; Betancourt, 2017), RMHMC aims to target a positive distribution $\pi$ on $\mathbb{R}^d$ and relies on the integration of a canonical Hamiltonian equation. In contrast, RMHMC incorporates some *geometrical* information in the definition of the Hamiltonian via a Riemannian metric $\mathfrak{g}$ on $\mathbb{R}^d$. When considering applications of RMHMC to a convex, open and bounded subset M, a natural choice for this metric is the Hessian metric associated with a self-concordant barrier on M, (Nesterov & Nemirovskii, 1994), as suggested by Kook et al. (2022a). We adopt here the same approach and now focus on a constrained subspace M which is assumed to be equipped with an appropriately designed "self-concordant" metric, (Nesterov & Nemirovskii, 1994).

Given this setting, we propose the BHMC (Barrier HMC) algorithm. To the best of our knowledge, this is **the first version of RMHMC incorporating an "involution checking step"** to assess the issue of asymptotic bias arising from the use of implicit integration methods to solve the Hamiltonian dynamics. In this work, we focus on the *numerical* version of BHMC (n-BHMC), for which we provide a numerical implementation. We refer to Appendix D for an introduction to the ideal version of BHMC, called *continuous* BHMC (c-BHMC), where it is assumed that we have access to the continuous Hamiltonian dynamics but which cannot be computed in practice. For each algorithm, we rigorously prove that the associated Markov chain preserves $\pi$.

The rest of the paper is organized as follows. In Section 2, we introduce our sampling framework and review some background on self-concordance and Riemannian geometry. In Section 3, we present n-BHMC, and derive theoretical results for this algorithm in Section 4. We review related works in Section 5 and provide numerical experiments for n-BHMC in Section 6.

**Notation.** For any $f \in \mathrm{C}^3(\mathbb{R}^d, \mathbb{R})$, we denote by $Df$ (and $D^2 f$) the Jacobian (resp. the Hessian) of $f$. For any $\mathrm{A} \in \mathbb{R}^{d \times d}$ and any $(x, y) \in \mathbb{R}^d \times \mathbb{R}^d$, we denote by $\mathrm{A} : D^3 f(x)$ the vector $(\mathrm{Tr}(\mathrm{A}^\top \{D^3 f(x)\}_i))_{i \in [d]} \in \mathbb{R}^d$, and define $D^3 f(x)[x, y]$ as the vector $x^\top D^3 f(x) y \in \mathbb{R}^d$. For any positive-definite matrix $\mathrm{A} \in \mathbb{R}^{d \times d}$, $\langle \cdot, \cdot \rangle_\mathrm{A}$ stands for the scalar product induced by A on $\mathbb{R}^d$, defined by $\langle x, y \rangle_\mathrm{A} = \langle x, \mathrm{A}y \rangle$. The "momentum reversal" operator $s : \mathbb{R}^d \times \mathbb{R}^d \to \mathbb{R}^d \times \mathbb{R}^d$ is defined for any $(x, p) \in \mathbb{R}^d \times \mathbb{R}^d$ by $s(x, p) = (x, -p)$. Let $\mathsf{E}, \mathsf{F}$ be two sets and $h : \mathsf{E} \to 2^\mathsf{F}$, where $2^\mathsf{F}$ is the set of sets of F. We say that $h$ is a set-valued map. Note that any map $g : \mathsf{E} \to \mathsf{F}$ can be extended to a set-valued map by identifying, for any $x \in \mathsf{E}$, $g(x)$ and $\{g(x)\}$. Let $f : \mathsf{F} \to 2^\mathsf{G}$. We define $f \circ h : \mathsf{E} \to 2^\mathsf{G}$ for any $x \in \mathsf{E}$ by $f \circ h(x) = \cup_{y \in h(x)} f(y)$, where by convention $\cup_\emptyset = \emptyset$. For any topological space X, we denote $\mathcal{B}(\mathsf{X})$ its Borel sets. For any probability measure $\mu \in \mathscr{P}(\mathsf{X})$ and measurable map $\varphi : \mathsf{X} \to \mathsf{Y}$, we denote $\varphi_\# \mu \in \mathscr{P}(\mathsf{Y})$ the pushforward of $\mu$ by $\varphi$. In general, we will equivalently denote by $z$ or $(x, p)$ a state of the Hamiltonian system.

## 2 Setting and Background

In this paper, we consider an open subset $\mathsf{M} \subset \mathbb{R}^d$, and we aim at sampling from a target distribution $\pi$ given for any $x \in \mathsf{M}$ by

$$\mathrm{d}\pi(x)/\mathrm{d}x = \exp[-V(x)]/Z ,$$

where $V \in \mathrm{C}^2(\mathsf{M}, \mathbb{R})$ and $Z = \int_\mathsf{M} \exp[-V(x)]\mathrm{d}x$. We view M as an embedded $d$-dimensional submanifold of $\mathbb{R}^d$, equipped with a metric $\mathfrak{g}$, satisfying the following assumptions.

**A1.** M *is an open convex bounded subset of* $\mathbb{R}^d$.

**A2.** *There exists* $\phi$, *a* $\alpha$-*regular and* $\nu$-*self-concordant barrier on* M *such that* $\mathfrak{g} = D^2 \phi$.

We provide in Section 2.1 basic Riemannian facts along with the definition of the Hamiltonian dynamics of RMHMC and introduce self-concordance and $\alpha$-regularity in Section 2.2.

## 2.1 Riemannian Manifold Hamiltonian dynamics

**Basics on Riemannian geometry.** Let $\mathsf{M}$ be a $d$-dimensional smooth manifold, endowed with a metric $\mathfrak{g}$. Denoting by $\mathrm{d}x$ a dual coframe, (Lee, 2006, Lemma 3.2.), we recall that the *Riemannian volume element* corresponding to $(\mathsf{M}, \mathfrak{g})$ is given in local coordinates by

$$\mathrm{dvol}_\mathsf{M}(x) = \sqrt{\det(\mathfrak{g})}\mathrm{d}x \ .$$

We denote by $\mathrm{T}_x^\star\mathsf{M}$ the dual of the tangent space at $x \in \mathsf{M}$, *i.e.*, the cotangent space. For any $x \in \mathsf{M}$, $\mathrm{T}_x^\star\mathsf{M}$ is a vector space naturally endowed with the scalar product $\langle \cdot, \cdot \rangle_{\mathfrak{g}(x)^{-1}}$, (Mok, 1977). We recall that the cotangent bundle $\mathrm{T}^\star\mathsf{M}$ is defined by $\mathrm{T}^\star\mathsf{M} = \sqcup_{x\in\mathsf{M}}\{x\} \cup \mathrm{T}_x^\star\mathsf{M}$. This set is a $2d$-dimensional manifold endowed with a Riemannian metric $\mathfrak{g}^\star$ given by Mok (1977), inherited from $\mathfrak{g}$. With this metric, the volume form on $\mathrm{T}^\star\mathsf{M}$ does not depend on $\mathfrak{g}$ anymore, and satisfies for any $(x, p) \in \mathrm{T}^\star\mathsf{M}$

$$\mathrm{dvol}_{\mathrm{T}^\star\mathsf{M}}(x, p) = \mathrm{d}x\mathrm{d}p \ \ ,$$

where $\mathrm{d}x\mathrm{d}p$ is a dual coframe for $\mathrm{T}^\star\mathsf{M}$.

Therefore, under **A**1, identifying $\mathrm{T}^\star\mathsf{M}$ with $\mathsf{M} \times \mathbb{R}^d$, the volume form on $\mathrm{T}^\star\mathsf{M}$ induced by $\mathfrak{g}^\star$ is the Lebesgue measure of $\mathbb{R}^d \times \mathbb{R}^d$ restricted to $\mathrm{T}^\star\mathsf{M}$. Despite adopting at first a Riemannian perspective on $\mathsf{M}$, we actually recover the Euclidean setting on $\mathrm{T}^\star\mathsf{M}$, which motivates us to consider $\mathrm{T}^\star\mathsf{M}$ as our sampling space. We refer to Appendix B for details on the cotangent bundle and its metric $\mathfrak{g}^\star$.

**Hamiltonian dynamics.** Note that with the previously introduced notation, $\pi$ can be expressed as

$$\mathrm{d}\pi/\mathrm{dvol}_\mathsf{M}(x) = \exp[-V(x) - \tfrac{1}{2}\log(\det(\mathfrak{g}(x)))]/Z \ .$$

This motivates the introduction of the following Hamiltonian on $\mathrm{T}^\star\mathsf{M}$

$$H(x, p) = V(x) + \tfrac{1}{2}\log\left(\det\mathfrak{g}(x)\right) + \tfrac{1}{2}\|p\|_{\mathfrak{g}(x)^{-1}}^2 \ . \tag{1}$$

In this definition, we notably take into account the scalar product $\langle \cdot, \cdot \rangle_{\mathfrak{g}(x)^{-1}}$ on $\mathrm{T}_x^\star\mathsf{M}$. Finally, we consider the joint distribution $\bar{\pi}$ on $\mathrm{T}^\star\mathsf{M}$

$$\mathrm{d}\bar{\pi}(x, p) = (1/\bar{Z})\exp[-H(x, p)]\mathrm{dvol}_{\mathrm{T}^\star\mathsf{M}}(x, p) \ , \tag{2}$$

where $\bar{Z} = \int_{\mathrm{T}^\star\mathsf{M}} \exp[-H(x, p)]\mathrm{dvol}_{\mathrm{T}^\star\mathsf{M}}(x, p)$, for which the first marginal is $\pi$. Indeed, for any $\varphi \in \mathrm{C}(\mathsf{M}, \mathbb{R})$, we have

$$\int_{\mathrm{T}^\star\mathsf{M}} \varphi(x)\mathrm{d}\bar{\pi}(x, p) = \int_{\mathrm{T}^\star\mathsf{M}} \varphi(x)\mathrm{d}\pi(x)\mathrm{N}_x(p; 0, \mathrm{Id})\mathrm{d}p = \int_\mathsf{M} \varphi(x)\mathrm{d}\pi(x) \ ,$$

where we denote by $\mathrm{N}_x(0, \mathrm{I}_d)$ the centered standard Gaussian distribution w.r.t. $\| \cdot \|_{\mathfrak{g}(x)^{-1}}$. The Hamiltonian dynamics for $H$ is given by the coupled Ordinary Differential Equations (ODEs)

$$\dot{x} = \partial_p H(x, p) \ , \qquad \dot{p} = -\partial_x H(x, p) \ , \tag{3}$$

where the derivatives of $H$ can be computed explicitly as

$$\partial_p H(x, p) = \mathfrak{g}(x)^{-1}p \ , \qquad \partial_x H(x, p) = -\tfrac{1}{2}D\mathfrak{g}(x)[\mathfrak{g}(x)^{-1}p, \mathfrak{g}(x)^{-1}p] + L(x) \ ,$$

where $L(x) = \nabla V(x) + \tfrac{1}{2}\mathfrak{g}(x)^{-1} : D\mathfrak{g}(x)$.

## 2.2 Self-concordance and regularity

Until now, we have considered an arbitrary Riemannian metric $\mathfrak{g}$. In the rest of this work, we focus on metrics given by Hessian of self-concordant barriers.

**Self-concordance.** We first introduce self-concordant barriers, a family of smooth convex functions which are well-suited for minimization by the Newton method.

**Definition 1** (Nesterov & Nemirovskii (1994)). *Let $\mathsf{U}$ be a non-empty open convex domain in $\mathbb{R}^d$. A function $\phi : \mathsf{U} \to \mathbb{R}$ is said to be a $\nu$-self-concordant barrier (with $\nu \geq 1$) on $\mathsf{U}$ if it satisfies:*
*(a) $\phi \in \mathrm{C}^3(\mathsf{U}, \mathbb{R})$ and $\phi$ is convex,*
*(b) $\phi(x) \to +\infty$ as $x \to \partial\mathsf{U}$,*
*(c) $|D^3\phi(x)[h, h, h]| \leq 2\|h\|_{\mathfrak{g}(x)}^3$, for any $x \in \mathsf{M}$, $h \in \mathbb{R}^d$,*
*(d) $|D\phi(x)[h]|^2 \leq \nu\|h\|_{\mathfrak{g}(x)}^2$, for any $x \in \mathsf{M}$, $h \in \mathbb{R}^d$,*
*where $\mathfrak{g}(x) = D^2\phi(x)$.*

Balls for $\|\cdot\|_{\mathfrak{g}(x)}$, called Dikin ellipsoids, are key for the study of self-concordance, see Appendix C.

**Regularity.** The property of $\alpha$-regularity for some $\alpha \geq 1$ is shared by many self-concordant barriers, including logarithmic and quadratic programming barriers. It ensures stability for interior point polynomial-time methods. Definition and properties of $\alpha$-regularity are recalled in Appendix C.

**Example of the polytope.** Let us assume that M is the polytope $\mathsf{M} = \{x : \mathrm{A}x < b\}$, where $\mathrm{A} \in \mathbb{R}^{m \times d}$ and $b \in \mathbb{R}^m$. We endow it with the Riemannian metric $\mathfrak{g}(x) = D^2\phi(x)$ where $\phi : \mathsf{M} \to \mathbb{R}$ is the logarithmic barrier given for any $x \in \mathsf{M}$ by $\phi(x) = -\sum_{i=1}^m \log(b_i - \mathrm{A}_i^\top x)$. The barrier $\phi$ is both a $m$-self-concordant barrier and a 2-regular function, (Nesterov & Nemirovski, 1998, page 3). Moreover, we have for any $x \in \mathsf{M}$, $\mathfrak{g}(x) = \mathrm{A}^\top S(x)^{-2}\mathrm{A}$, where $S(x) = \mathrm{Diag}\left(b_i - \mathrm{A}_i^\top x\right)_{i \in [m]}$. We provide in Figure 1 an illustration of this barrier.

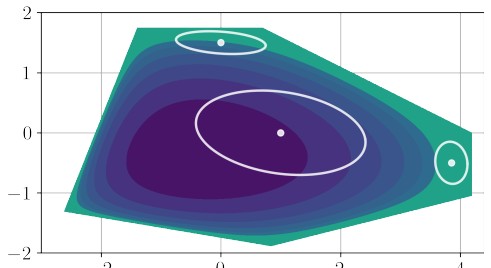

Figure 1: A polytope $\mathsf{M} \subset \mathbb{R}^2$ with three Dikin ellipsoids $\{y \in \mathbb{R}^d : y^\top \mathfrak{g}(x)y < 1\}$ derived from its logarithmic barrier.

## 3 The n-BHMC algorithm

In practice, it is not possible to *exactly* compute the Riemannian Hamiltonian dynamics (3). We thus introduce a *numerical* version of BHMC (n-BHMC), in which we replace the continuous ODE integration by a symplectic numerical scheme. We first define the Hamiltonian integrators used in n-BHMC in Section 3.1 and provide details on the different steps of our algorithm in Section 3.2.

### 3.1 Hamiltonian integrators of n-BHMC

In the same spirit as Shahbaba et al. (2014), we first rewrite the Hamiltonian $H$ given in (1) as $H = H_1 + H_2$, where we highlight the non separable aspect of $H$ in $H_2$, *i.e.*, for any $(x, p) \in \mathrm{T}^\star\mathsf{M}$

$$H_1(x, p) = V(x) + \tfrac{1}{2}\log(\det\mathfrak{g}(x)) , \quad H_2(x, p) = \tfrac{1}{2}\|p\|^2_{\mathfrak{g}(x)^{-1}} .$$

Therefore, we have

$$\partial_x H_1(x, p) = \nabla V(x) + \tfrac{1}{2}\mathfrak{g}(x)^{-1} : D\mathfrak{g}(x) , \qquad \partial_p H_1(x, p) = 0 , \qquad (4)$$

$$\partial_x H_2(x, p) = -\tfrac{1}{2}D\mathfrak{g}(x)[\mathfrak{g}(x)^{-1}p, \mathfrak{g}(x)^{-1}p] , \qquad \partial_p H_2(x, p) = \mathfrak{g}(x)^{-1}p . \qquad (5)$$

Leveraging the separable structure of $H_1$, we now define an implicit scheme to discretize the Hamiltonian dynamics (3) by *splitting* it into the Hamiltonian dynamics (4) and (5). In the rest of this section, we consider some step-size $h \in \mathbb{R}$.

**Explicit integrator of $H_1$.** We first approximate the dynamics (4) on a step-size $h/2$ using a first-order Euler method (Hairer et al., 2006, Theorem VI.3.3.). Since $H_1$ is separable, this integrator simply reduces to the map $\mathrm{S}_{h/2} : \mathrm{T}^\star\mathsf{M} \to \mathrm{T}^\star\mathsf{M}$ defined by

$$\mathrm{S}_{h/2}(x, p) = (x, p - \tfrac{h}{2}\partial_x H_1(x, p)) .$$

We have: (i) $\mathrm{S}_{h/2}(\mathrm{T}^\star\mathsf{M}) \subset \mathrm{T}^\star\mathsf{M}$, (ii) $\mathrm{S}_{h/2}$ is symplectic (we refer to Appendix F for more details on symplecticity), (iii) $\mathrm{S}_{-h/2} \circ \mathrm{S}_{h/2} = \mathrm{Id}$ and (iv) $\mathrm{S}_{-h/2} = s \circ \mathrm{S}_{h/2} \circ s$. Note that $s \circ \mathrm{S}_{h/2}$ is an involution on $\mathrm{T}^\star\mathsf{M}$, which inherits from properties (i) and (ii) of $\mathrm{S}_{h/2}$.

**Implicit integrator of $H_2$.** Since $H_2$ is not separable, a common discretization scheme such as the Euler method is not symplectic anymore, which requires to use an implicit method instead. We thus approximate these dynamics on a step-size $h$ with a symplectic second-order integrator denoted by $\mathrm{G}_h$. For theoretical purposes, we focus on the Störmer-Verlet scheme, (Hairer et al., 2006, Theorem VI.3.4.), also known as the *generalized Leapfrog* integrator, which is widely used in geometric integration, (Girolami & Calderhead, 2011; Betancourt, 2013; Cobb et al., 2019; Brofos & Lederman, 2021a). We refer to Appendix F for a discussion on other numerical schemes. For any $z^{(0)} \in \mathrm{T}^\star\mathsf{M}$, $\mathrm{G}_h(z^{(0)}) \subset \mathrm{T}^\star\mathsf{M}$ consists of points $z^{(1)} = (x^{(1)}, p^{(1)})$ which are solution of

$$p^{(1/2)} = p^{(0)} - \tfrac{h}{2}\partial_x H_2(x^{(0)}, p^{(1/2)}) ,$$
$$x^{(1)} = x^{(0)} + \tfrac{h}{2}[\partial_p H_2(x^{(0)}, p^{(1/2)}) + \partial_p H_2(x^{(1)}, p^{(1/2)})] ,$$
$$p^{(1)} = p^{(1/2)} - \tfrac{h}{2}\partial_x H_2(x^{(1)}, p^{(1/2)}) . \qquad (6)$$

As defined, there is no guarantee that (6) admits a (unique) solution. As a matter of fact, the integrator $G_h$ can be seen as a set-valued map. Moreover, it is easy to check that: (i) $G_h(T^\star M) \subset 2^{T^\star M}$, (ii) $G_{-h} = s \circ G_h \circ s$, since $\partial_p H_2(s(z)) = -\partial_p H_2(z)$ and $\partial_x H_2(s(z)) = \partial_x H_2(z)$, and (iii) if $|G_h(z)| > 0$ then $z \in (G_{-h} \circ G_h)(z)$.

**Implicit integrator of $H$.**   Relying on the integrators $S_{h/2}$ and $G_h$ previously defined, we now explain how we approximate the Hamiltonian dynamics (3) on a step-size $h$. We first define the set-valued maps $F_h : T^\star M \to 2^{T^\star M}$ and $R_h : T^\star M \to 2^{T^\star M}$ by

$$F_h = G_h \circ s , \quad R_h = (s \circ S_{h/2}) \circ F_h \circ (s \circ S_{h/2}) .$$

Using properties (ii) and (iii) of $G_h$, we have for any $z \in T^\star M$ such that $|F_h(z)| > 0$, $z \in (F_h \circ F_h)(z)$, and for any $z' \in T^\star M$ such that $|(F_h \circ s \circ S_{h/2})(z')| > 0$, we have $z' \in (R_h \circ R_h)(z')$. Thus, $F_h$ and $R_h$ are *involutive* in the sense of set-valued maps. Note also that $s \circ R_h = S_{h/2} \circ G_h \circ S_{h/2}$ boils down to the *Strang splitting* (Strang, 1968) of the dynamics (3) and thus, is a symplectic scheme approximating the dynamics of Hamiltonian $H$ on a step-size $h$.

**Numerical integrators.**   In practice, we do not have access to $F_h$ but approximate it with a *numerical map* $\Phi_h$. For clarity's sake, we denote by $\mathrm{dom}_{\Phi_h} \subset T^\star M$ the domain of this integrator with $\Phi_h(\mathrm{dom}_{\Phi_h}) \subset T^\star M$. In practice, $\mathrm{dom}_{\Phi_h}$ corresponds to the set of points for which the numerical integration of $F_h$ outputs a solution. We also approximate $R_h$ with the *numerical map* $R_h^\Phi : (s \circ S_{h/2})(\mathrm{dom}_{\Phi_h}) \subset T^\star M \to T^\star M$

$$R_h^\Phi = (s \circ S_{h/2}) \circ \Phi_h \circ (s \circ S_{h/2}) .$$

Similarly to $s \circ R_h$, $s \circ R_h^{\Phi\,2}$ approximates the dynamics of Hamiltonian $H$ on a step-size $h$. In our experiments, we design $\Phi_h$ using a fixed-point solver with a given number of iterations following Brofos & Lederman (2021a,b). We refer to Appendix K for details on computations of $\Phi_h$.

### 3.2   Steps of the algorithm

For any $z = (x, p) \in T^\star M$, we define on $T^\star M$ the norm $\|\cdot\|_z$ by

$$\|(x', p')\|_z = \|x'\|_{\mathfrak{g}(x)} + \|p'\|_{\mathfrak{g}(x)^{-1}} , \quad \forall (x', p') \in T^\star M , \tag{7}$$

where $\|\cdot\|_{\mathfrak{g}(x)^{-1}}$ is the canonical norm on $T_x^\star M$ induced by the Riemannian metric $\mathfrak{g}$ (see Section 2.1) and $\|\cdot\|_{\mathfrak{g}(x)}$ is the common norm used on $M$ to study properties of self-concordance (see Section 2.2). As defined, this norm will be crucial to correct the bias of the numerical integration in n-BHMC, presented in Algorithm 1, for which we are now ready to detail the steps.

**Steps 1 and 5: applying a momentum update.**   Assume that the current state at stage $n \geq 1$ is $(X_{n-1}, P_{n-1}) \in T^\star M$. Then, the momentum is first partially refreshed such that $\tilde{P}_n \leftarrow \sqrt{1-\beta} P_{n-1} + \sqrt{\beta} G_n$, where $G_n \sim N_x(0, I_d)$ is independent from $P_{n-1}$. This update is applied both at the beginning and the end of the iteration, similarly to Lelièvre et al. (2022).

**Step 2: solving a discretized version of ODE (3).**   Starting from $(X_n', P_n') = (X_{n-1}, \tilde{P}_n)$, we now approximate the dynamics of $H$ on a step-size $h$ with the integrators presented in Section 3.1, while ensuring that the proposal state belongs to $T^\star M$. We proceed as follows:

1. We first run the *explicit* integrator $S_{h/2}$ and compute $Z_n^{(0)} = (s \circ S_{h/2})(X_{n-1}, \tilde{P}_n)$.

2. If $Z_n^{(0)} \notin \mathrm{dom}_{\Phi_h}$, then $(X_n', P_n')$ is not updated and we directly go to Step 3. Otherwise, we define $Z_n^{(1)} = \Phi_h(Z_n^{(0)})$. To ensure the reversibility of this step, we then perform an "involution checking step", (Zappa et al., 2018; Lelièvre et al., 2019), highlighted in yellow in Algorithm 1, *i.e.*, we verify (a) $Z_n^{(1)} \in \mathrm{dom}_{\Phi_h}$ and (b) $Z_n^{(0)} = \Phi_h(Z_n^{(1)})$. In practice, we replace condition (b) by combining an explicit tolerance threshold $\eta$ with the norm defined in (7) and accept the proposal if

$$\|Z_n^{(0)} - \Phi_h(Z_n^{(1)})\|_{Z_n^{(0)}} + \|Z_n^{(0)} - \Phi_h(Z_n^{(1)})\|_{\Phi_h(Z_n^{(1)})} \leq \eta . \tag{8}$$

We discuss the choice of this norm to perform the "involution checking step" in Appendix K. If both of these conditions are satisfied, we finally set $(X_n', P_n') = (s \circ S_{h/2})(Z_n^{(1)})$, to ensure the reversibility of the update of $(X_n', P_n')$. Otherwise, $(X_n', P_n')$ is not updated and we go to Step 3.

---

[2]Note that $s \circ R_h^\Phi$ is a *function* and not a set-valued map.

---

**Algorithm 1:** n-BHMC with Momentum Refresh

**Input:** $(X_0, P_0) \in \mathrm{T}^\star \mathsf{M}$, $\beta \in (0,1]$, $N \in \mathbb{N}$, $h > 0$, $\eta > 0$, $\Phi_h$ with domain $\mathrm{dom}_{\Phi_h}$
**Output:** $(X_n, P_n)_{n \in [N]}$

1 **for** $n = 1, ..., N$ **do**
2     Step 1:   $G_n \sim \mathrm{N}_x(0, \mathrm{I}_d), \tilde{P}_n \leftarrow \sqrt{1-\beta} P_{n-1} + \sqrt{\beta} G_n$
3     Step 2:   solving a discretized version of ODE (3)
4     $X'_n, P'_n \leftarrow X_{n-1}, \tilde{P}_n; \qquad X_n^{(0)}, P_n^{(0)} \leftarrow (s \circ \mathrm{S}_{h/2})(X_n, \tilde{P}_n)$
5     **if** $Z_n^{(0)} = (X_n^{(0)}, P_n^{(0)}) \in \mathrm{dom}_{\Phi_h}$ **then**
6        $Z_n^{(1)} = \Phi_h(Z_n^{(0)})$
7        $\mathrm{err}_0 = \|Z_n^{(0)} - \Phi_h(Z_n^{(1)})\|_{Z_n^{(0)}}, \quad \mathrm{err}_1 = \|Z_n^{(0)} - \Phi_h(Z_n^{(1)})\|_{\Phi_h(Z_n^{(1)})}$ (if defined)
8        **if** $\boxed{Z_n^{(1)} \in \mathrm{dom}_{\Phi_h} \ \& \ \mathrm{err}_0 + \mathrm{err}_1 \leq \eta}$ **then** $X'_n, P'_n \leftarrow (s \circ \mathrm{S}_{h/2})(Z_n^{(1)})$
9     **end**
10    Step 3:   $A_n \leftarrow \min(1, \exp[-H(X'_n, P'_n) + H(X_{n-1}, \tilde{P}_n)]); \qquad U_n \sim \mathrm{U}[0,1]$
11    **if** $U_n \leq A_n$ **then** $\bar{X}_n, \bar{P}_n \leftarrow X'_n, P'_n$
12    **else** $\bar{X}_n, \bar{P}_n \leftarrow X_{n-1}, \tilde{P}_n$
13    Step 4:   $X_n, \hat{P}_n \leftarrow s(\bar{X}_n, \bar{P}_n)$
14    Step 5:   $G'_n \sim \mathrm{N}_x(0, \mathrm{I}_d), P_n \leftarrow \sqrt{1-\beta} \hat{P}_n + \sqrt{\beta} G'_n$
15 **end**

---

**Step 3: computing the acceptance filter.** We denote by $a(x', p'|x, p)$ the acceptance probability to move from $(x, p) \in \mathrm{T}^\star \mathsf{M}$ to $(x', p') \in \mathrm{T}^\star \mathsf{M}$, which is given by

$$a(x', p'|x, p) = 1 \wedge \exp[-H(x', p') + H(x, p)] . \tag{9}$$

After Step 2, we perform a simple MH filter by accepting the proposal with probability $A_n = a(X'_n, P'_n | X_{n-1}, \tilde{P}_n)$ similarly to a classical HMC algorithm.

**Step 4: applying momentum reversal.** To ensure a move along the dynamics of $H$ with step-size $h$, we reverse the momentum after the acceptance step. Indeed, if the acceptance filter is successful, then $(X_n, \hat{P}_n) = (s \circ \mathrm{R}_h^\Phi)(X_{n-1}, \tilde{P}_{n-1})$ approximates the Hamiltonian dynamics (3) on a step-size $h$ starting at $(X_{n-1}, \tilde{P}_{n-1})$, as seen in Section 3.1. Otherwise, $(X_n, \hat{P}_n) = (X_{n-1}, -\tilde{P}_{n-1})$.

### 3.3 About the usefulness of the "involution checking step"

In their paper, Kook et al. (2022a) also incorporate self-concordance into RMHMC to sample from distributions supported on polytopes. To integrate the Hamiltonian dynamics, they propose to use a numerical version of the *Implicit Midpoint integrator* (Hairer et al., 2006, Theorem VI.3.5.), which shares the same theoretical properties as the Störmer-Verlet scheme. This results in CRHMC (Kook et al., 2022a, Algorithm 3), whose main difference compared to n-BHMC is the lack of the "involution checking step" (Line 8 in Algorithm 1). Crucially, **Kook et al. (2022a) assume that their implicit integrator admits a unique solution** given any starting point and any step-size $h$, which is then approximated by their numerical scheme. However, this statement may not be true in the self-concordant setting, as we explain below with a simple example. This explains why we always refer to the implicit integrator as a *set-valued map*. In their setting, the numerical integrator is not guaranteed to be involutive, which results in an asymptotic bias that the MH filter cannot solve.

In our paper, we do not make such an assumption on the Störmer-Verlet integrator, and consider a numerical solver which outputs *one* approximate solution to this implicit scheme. By doing so, our analysis is meant to be as close as possible to the practical implementation of RMHMC. Then, the "involution checking step" is critical to make the numerical integrator locally involutive, which enables us to derive rigorous reversibility results, see Theorem 3, and to solve the issue of asymptotic bias caused by implicit integration, see Section 6. We refer to Appendix J for a detailed comparison between n-BHMC and CRHMC. We now present a simple setting where the assumption made by Kook et al. (2022a) does not hold in the case of the Störmer-Verlet integrator, which is defined in (6).

Consider the 1-dimensional setting $\mathsf{M} = (-\infty, 0)$ with $\mathfrak{g} : x \mapsto 1/x^2$. Assume that $\beta = 1$. Let $h > 0$ and let $X_0 \in \mathsf{M}$ be the (position) state of the Markov chain at the beginning of the first iteration in n-BHMC. In this case, $\tilde{P}_0 \sim \mathrm{N}(0, 1/X_0^2)$ and we have $P_0^{(0)} = \tilde{P}_0 - (h/2)\partial_x H_1(X_0, \tilde{P}_0)$, where the function $H_1$ is defined in Section 3.1. Using (4), we have $P_0^{(0)} \sim \mathrm{N}(\mu_0, 1/X_0^2)$, where

$\mu_0 = -(h/2)\partial_x H_1(X_0, \tilde{P}_0)$ only depends on $X_0$. Then, the implicit equation in $P_0^{(1/2)}$, given by the first equation of (6), reads as $P_0^{(1/2)} = P_0^{(0)} - (h/2)\partial_x H_2(X_0, P_0^{(1/2)})$. Using (5), it can be rewritten as a polynomial equation of degree 2 in $P_0^{(1/2)}$

$$\tfrac{h}{2} X_0 (P_0^{(1/2)})^2 + P_0^{(1/2)} - P_0^{(0)} = 0 \quad . \tag{10}$$

Denote $\Delta = 1 + 2hX_0 P_0^{(0)}$. Then, (10) may admit 0,1 or 2 solutions depending on the sign of $\Delta$. If $P_0^{(0)} \le -1/(2hX_0)$, i.e., $\Delta \ge 0$, then (10) admits 1 or 2 solutions. However, if $h$ is small enough, only one solution is valid when considering the constraint on the position update in (6). Whenever $P_0^{(0)} > -1/(2hX_0)$, i.e., $\Delta < 0$, (10) admits no solution. Recalling that $P_0^{(0)} \sim \mathrm{N}(\mu_0, 1/X_0^2)$, this event occurs with positive probability, thus violating the assumption made by Kook et al. (2022a).

## 4 Theoretical results

We now study the reversibility of n-BHMC with respect to the target distribution. To do so, we first present theoretical results on the *exact* integrators of the discretized Hamiltonian dynamics in Section 4.1, from which we derive our assumption on the numerical integrator used in n-BHMC. Finally, we state our main result on n-BHMC in Section 4.2.

### 4.1 From implicit to numerical integrators

We show in Proposition 2 that even though $\mathrm{F}_h$ is a *set-valued* map, it can *locally* be identified with a $\mathrm{C}^1$-diffeomorphism. In a manner akin to Lelièvre et al. (2022), this justifies our assumption **A**3 that the *numerical* map $\Phi_h$ used to approximate $\mathrm{F}_h$ in Step 2 of Algorithm 1 is *locally* a $\mathrm{C}^1$-involution.

**Proposition 2.** *Assume* **A**1, **A**2. *Let* $z^{(0)} \in \mathrm{T}^\star\mathsf{M}$, *then there exists* $h^\star > 0$ *(explicit in Appendix G) such that for any* $h \in (0, h^\star)$, *there exist* $z_h^{(1)} \in \mathrm{F}_h(z^{(0)})$, *a neighborhood* $\mathsf{U} \subset \mathrm{T}^\star\mathsf{M}$ *of* $z^{(0)}$ *and a* $\mathrm{C}^1$-*diffeomorphism* $\gamma_h : \mathsf{U} \to \gamma_h(\mathsf{U}) \subset \mathrm{T}^\star\mathsf{M}$ *with*

*(a)* $\gamma_h(z^{(0)}) = z_h^{(1)}$ *and* $|\det \mathrm{Jac}(\gamma_h)| = 1$.

*(b)* $\gamma_h(z)$ *is the only element of* $\mathrm{F}_h(z)$ *in* $\gamma_h(\mathsf{U})$ *for any* $z \in \mathsf{U}$.

Proposition 2 shows that while $\mathrm{F}_h$ can take multiple (or none) values on $\mathrm{T}^\star\mathsf{M}$, for any $z^{(0)} \in \mathrm{T}^\star\mathsf{M}$, there exists $h$ small enough and a neighborhood $\mathsf{U}$ of $z^{(0)}$ such that the set-valued map $\mathrm{F}_h$ is locally symplectic on $\mathsf{U}$. The proof of Proposition 2 is given in Appendix G. It first relies on considering the Störmer-Verlet scheme introduced in (6) as the composition of maps for which we derive the existence of solutions and secondly applying the implicit function theorem on these maps. Motivated by this result on the *implicit map* $\mathrm{F}_h$ and the fact that for any $z \in \mathrm{T}^\star\mathsf{M}$ with $|\mathrm{F}_h(z)| > 0$, $z \in \mathrm{F}_h \circ \mathrm{F}_h(z)$ (see Section 3.1), we make the following assumption on the *numerical map* $\Phi_h$.

**A**3. *There exists* $\lambda \in (0,1)$ *s.t. for any* $z^{(0)} \in \mathrm{T}^\star\mathsf{M}$, *there exists* $h_\star > 0$ *and for any* $h \in (0, h_\star)$

*(a)* $\mathsf{B} = \mathsf{B}_{\|\cdot\|_{z^{(0)}}}(z^{(0)}, \lambda r^\star(x^{(0)})) \subset \mathrm{dom}_{\Phi_h}$,

*(b)* $\Phi_h \in \mathrm{C}^1(\mathsf{B}, \mathrm{T}^\star\mathsf{M})$ *and* $\Phi_h \circ \Phi_h = \mathrm{Id}$ *on* $\mathsf{B}$,

*with* $r^\star : \mathsf{M} \to (0, +\infty)$ *depending only on* $\mathfrak{g}$ *and defined in Appendix I.*

Assumption **A**3 can be thought as a strengthening of Proposition 2-(a), where (i) $h_\star$ refers to $h^\star$, and (ii) $\mathsf{B}$ and $\Phi_h$ correspond to *explicit* versions of $\mathsf{U}$ and $\gamma_h$. We conjecture that the involution condition in **A**3-(b) could in fact be replaced by the condition that for any $z \in \mathsf{B}$, $\Phi_h(z) \in \mathrm{F}_h(z)$, in a manner akin to Lelièvre et al. (2022). We leave this study for future work.

### 4.2 Reversibility results

**Beyond n-BHMC.** While Algorithm 1 can be implemented practically, it cannot be easily analysed under **A**1, **A**2 and **A**3. The main reason for this limitation is that the self-concordance properties are defined locally, whereas deriving reversibility results require global control. In particular, we cannot ensure that $\Phi_h$ is locally an involution around $z^{(0)}$ if $h > h_\star$. To circumvent this issue, we enforce a condition on $h$ to be small enough in n-BHMC. Using the notation from **A**3, we define the set

$$\mathsf{A}_h = \{z \in \mathrm{T}^\star\mathsf{M} : h < \min_{\tilde{z} \in \mathsf{B}_{\|\cdot\|_z}(z,1)} h_\star(\tilde{z})\} \subset \mathrm{dom}_{\Phi_h} .$$

Let $z^{(0)} \in \mathsf{A}_h$. By **A**3, we know that $\Phi_h$ is an involution on a neighbourhood of $z^{(0)}$; in particular, it comes that $(\Phi_h \circ \Phi_h)(z^{(0)}) = z^{(0)}$. Hence, the condition (b) of the "involution checking step" in Algorithm 1 is *de facto* satisfied. This naturally leads to replace "$Z_n^{(1)} \in \mathrm{dom}_{\Phi_h}$", *i.e.*, the condition (a) of the "involution checking step", by the more restrictive condition "$Z_n^{(1)} \in \mathsf{A}_h$", as presented in Algorithm 3 (see Appendix H), for which we are able to derive a reversibility result.

We denote by $\mathrm{Q} : \mathrm{T}^\star \mathsf{M} \times \mathcal{B}(\mathrm{T}^\star \mathsf{M}) \to [0, 1]$, the transition kernel of the (homogeneous) Markov chain $(X_n, P_n)_{n \in \mathbb{N}}$ obtained with Algorithm 3. We now state our main result on n-BHMC.

**Theorem 3.** *Assume* **A**1*,* **A**2*,* **A**3*. Then,* $\mathrm{Q}$ *is reversible up to momentum reversal, i.e., we have for any* $f \in \mathrm{C}(\mathrm{T}^\star \mathsf{M} \times \mathrm{T}^\star \mathsf{M}, \mathbb{R})$ *with compact support*
$$\int_{\mathrm{T}^\star \mathsf{M} \times \mathrm{T}^\star \mathsf{M}} f(z, z') \mathrm{d}\bar{\pi}(z) \mathrm{Q}(z, \mathrm{d}z') = \int_{\mathrm{T}^\star \mathsf{M} \times \mathrm{T}^\star \mathsf{M}} f(s(z'), s(z)) \mathrm{d}\bar{\pi}(z) \mathrm{Q}(z, \mathrm{d}z') .$$
*In particular,* $\bar{\pi}$ *is an invariant measure for* $\mathrm{Q}$.

*Proof.* We provide here a sketch of the proof, technical details being postponed to Appendix I. The reversibility (up to momentum reversal) of the momentum update is straightforward. To establish the reversibility up to momentum reversal of the *numerical* Hamiltonian integration step, we cover the compact support of $f$ with a finite family of open balls with respect to the metric $\mathfrak{g}$. Combining **A**2 and **A**3, we show that $\Phi_h$ is a volume-preserving $\mathrm{C}^1$-diffeomorphism on each one of these sets. We then conclude upon combining this result with the fact that $\mathrm{R}_h^\Phi \circ s \circ \mathrm{S}_{h/2} = s \circ \mathrm{S}_{h/2} \circ \Phi_h$. $\square$

Although Algorithm 3 may be implemented, this would come with a huge (and unrealistic) computational cost since verifying that $z \in \mathsf{A}_h$ implies to find the minimal critical step-size $h_\star$ on a neighborhood of $z$. The question of the existence of a global step-size $h$ such that Algorithm 1 can be properly analyzed is not the topic of this paper and is left for future work.

**Comparison with Theorem 8 in Kook et al. (2022a).** Although Kook et al. (2022a) present a theoretical result similar to Theorem 3, we claim that *their statement is not correct*. Indeed, the authors make a critical confusion between the *ideal* integrator as considered in Hairer et al. (2006) and the *numerical* version they implement. Then, in the proof of reversibility for their scheme, they act as if the two algorithms were the same while it is not true (see Appendix J for further details). On the other hand, (i) we make a clear distinction between the ideal integrator and its *numerical* implementation (see Section 3.1), and (ii) implement the "involution checking step" to enforce reversibility (Line 8 in Algorithm 1).

## 5 Related work

**Sampling on manifolds.** Traditional *constrained* sampling methods in Euclidean spaces include the Hit-and-Run algorithm (Lovász & Vempala, 2004), the Random Walk Metropolis-Hastings (RWMH) algorithm, also referred to as Ball Walk (Lee & Vempala, 2017a), and HMC, (Duane et al., 1987). However, it has been empirically demonstrated that RWMH and HMC require small step-sizes in order to correctly sample from a target distribution $\pi$ over a *submanifold* $\mathsf{M} \subset \mathbb{R}^d$, thus resulting in poor mixing time (Girolami & Calderhead, 2011, Figures 1 and 3). In the specific case where $\mathsf{M} = \{x \in \mathbb{R}^d : c(x) = 0\}$ for some $c : \mathbb{R}^d \to \mathbb{R}^m$, Brubaker et al. (2012) combine HMC with the RATTLE integrator (Leimkuhler & Skeel, 1994), incorporating the constraints of $\mathsf{M}$ in the Hamiltonian dynamics via Lagrange multipliers. Girolami & Calderhead (2011) adopt an original approach by endowing $\mathsf{M}$ with a Riemannian metric $\mathfrak{g}$ and propose RMHMC (Riemannian Manifold HMC), a version of HMC where the Hamiltonian depends on $\mathfrak{g}$ as in (1). It consists of integrating the Hamiltonian dynamics of (3) on short-time steps using the generalized Leapfrog integrator (6) and the acceptance filter defined in (9). This method does not include an "involution checking step", and thus the reversibility of the algorithm cannot be ensured in practice and in theory. Similarly, Byrne & Girolami (2013) propose to design a numerical integrator for RMHMC using geodesics.

**Choice of the metric in RMHMC.** There are various ways to design a *meaningful* metric $\mathfrak{g}$ given a submanifold $\mathsf{M}$. In a Bayesian perspective, Girolami & Calderhead (2011) aim at computing the *a posteriori* distribution of a statistical model of interest and thus choose $\mathfrak{g}$ to be the Fisher-Rao metric. In this paper, we consider another approach which exploits the *geometrical* structure of $\mathsf{M}$. For instance, one can always define a self-concordant barrier $\phi$ when $\mathsf{M}$ is a convex body, (Nesterov & Nemirovskii, 1994), and choose $\mathfrak{g} = D^2\phi$, as done by Kook et al. (2022a).

**Self-concordance in RMHMC.** Elaborating on Riemannian geodesics, Lee & Vempala (2018) provide theoretical guarantees of fast mixing time for RMHMC when (i) M is a polytope, and (ii) $\mathfrak{g}$ is the Hessian of a self-concordant function $\phi$ (as in our setting). Their result notably improves the complexity of uniform polytope sampling from algorithms relying on self-concordance such as the Dikin Walk (Kannan & Narayanan, 2009) and the Geodesic Walk (Lee & Vempala, 2017b). However, they only consider the case where the *exact* continuous Hamiltonian dynamics is used, as in c-BHMC (see Appendix D). On the other hand, Kook et al. (2022a) integrate the Hamiltonian dynamics via implicit schemes without any "involution checking step" in a similar self-concordant setting. In particular, they consider a convex body K equipped with a self-concordant barrier $\phi$, combined with a linear equality constraint $Ax = b$. This is a special case of our framework (see **A**1 and **A**2) by rewriting this whole set as $K' = \{A^\dagger b + u \in K \ : \ u \in \text{Ker}(A)\}$[3], equipped with the self-concordant barrier $u \mapsto \phi(A^\dagger b + u)$. Besides this, Kook et al. (2022a) provide an efficient implementation of their algorithm (CRHMC) in the case of a convex bounded manifold of the form $M = \{x \in \mathbb{R}^d \ : \ Ax = b, \ell < x < u\}$. Although CRHMC demonstrates empirical fast mixing time, we show in Section 6 that it suffers from an asymptotic bias. More recently, Kook et al. (2022b) built upon the empirical results of Kook et al. (2022a) to derive theoretical results of fast mixing time. However, they do not question the issue of asymptotic bias in CRHMC.

**Enforcing reversibility.** Zappa et al. (2018) propose a version of RWMH including projection steps after each proposal. To our knowledge, they are the first to practice "involution checking" of the proposal, and thus *enforce the reversibility* of the Markov chain with respect to $\pi$. Lelièvre et al. (2019) notably combine this method with the discretization suggested by Brubaker et al. (2012) to also *enforce the constraints* of the manifold. Lelièvre et al. (2022) elaborate on this framework, by designing a symplectic numerical integrator with multiple possible outputs, and provide a rigorous proof of reversibility. Note that it only applies when $\mathfrak{g}$ is induced by the flat metric of $\mathbb{R}^d$ and therefore cannot be combined with our approach as such.

## 6 Numerical experiments

In our experiments[4], we illustrate the performance of n-BHMC (Algorithm 1) to sample from target distributions which are supported on polytopes. We compare our method with the numerical implementation of CRHMC[5] provided by Kook et al. (2022a). In all of our settings, we compute $\mathfrak{g}$ as the Hessian of the logarithmic barrier, see Section 2.2. The algorithms are always initialized at the center of mass of the considered polytope. At each iteration of n-BHMC, we perform one step of numerical integration, using the Störmer-Verlet scheme with $K = 30$ fixed-point steps and keep the refresh parameter $\beta$ equal to 1. We refer to Appendix K for more details on the setting of our experiments and additional results.

**Synthetic data.** We first consider the problem of sampling from the truncated Gaussian distribution

$$(\mathrm{d}\pi/\mathrm{dLeb})(x) = \exp[-\|x-\mu\|^2/2]\mathbb{1}_M(x)/\int_M \exp[-\|\tilde{x}-\mu\|^2/2]\mathrm{d}\tilde{x}, \quad \forall x \in \mathbb{R}^d,$$

where M is the hypercube or the simplex, and $\mu \in \mathbb{R}^d$ is defined by

$$\mu = (10/\sqrt{d-1}) \times \{\mathbf{1} - e_1 + (\sqrt{d-1}-1)e_2\},$$

where $e_i$ stands for the $i$-th canonical vector of $\mathbb{R}^d$ and $\mathbf{1} = \sum_{i=1}^d e_i$. In particular, $\mu_1 = 0$, $\mu_2 = 10$ and $\mu_j = \mu_2/\sqrt{d-1}$ for any $j \in \{3, \ldots, d\}$. Therefore, the mass of $\pi$ is not evenly distributed on the boundary of M, which is key to observe the impact of the reversibility condition. For this experiment, we consider the low-dimensional setting $d \in \{5, 10\}$. To sample from $\pi$, we run 10 times the algorithms CRHMC and n-BHMC for $N = 10^5$ iterations, and update the step-size $h$ such that the average acceptance rate in the MH filter is roundly equal to 0.5, following Roberts & Rosenthal (2001). We discuss the setting of the tolerance parameter $\eta$ for n-BHMC in Appendix K. In order to correctly assess the bias of each numerical method, we aim at accurately computing the target quantity $Q = \int_M \langle x, \mu \rangle_2 \mathrm{d}\pi(x)$. Although our choice of $Q$ is arbitrary, it highlights well the bias in CRHMC, which is corrected when using n-BHMC.

---

[3]$A^\dagger$ is the pseudo-inverse of A.

[4]Our code: https://github.com/maxencenoble/barrier-hamiltonian-monte-carlo.

[5]https://github.com/ConstrainedSampler/PolytopeSamplerMatlab

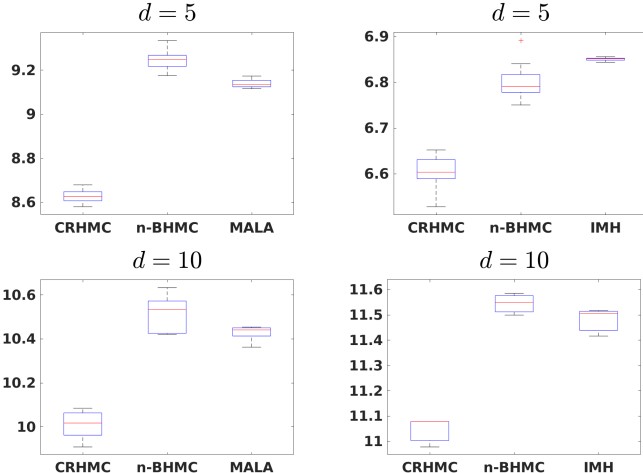

Figure 2: Comparison between n-BHMC and CRHMC on the hypercube (left) and the simplex (right).

We evaluate their performance by taking as ground truth the estimate given by (i) the Metropolis-Adjusted Langevin Algorithm (MALA) (Roberts & Stramer, 2002) for the hypercube and (ii) the Independent MH (IMH) algorithm (Liu, 1996) for the simplex, which are also run 10 times, but 10 times longer than n-BHMC and CRHMC, *i.e.*, for $N = 10^6$ iterations. To compute the target quantity $Q$, we keep the whole trajectories and report the confidence intervals in Figure 2. For both polytopes, we notably observe that our method is less biased than CRHMC.

**Real-world data.** We then consider 10 polytopes given in the COBRA Toolbox v3.0 (Heirendt et al., 2019), which model molecular systems, and follow the method provided in (Kook et al., 2022a, Appendix A) to pre-process them. Here, we aim at uniformly sampling from these polytopes. However, we do not have access to a realistic baseline that yields an unbiased estimator, since the sampling dimension is too high and running MALA or IMH would be too costly. Hence, instead of assessing the bias of the algorithms, we rather want to highlight that the "involution checking step" does not hurt the convergence properties of BHMC compared to CRHMC. We evaluate the efficiency of the

| Model | $d$ | NNZ | n-BHMC | CRHMC |
|---|---|---|---|---|
| ecoli | 95 | 291 | **0.08822** | 5.719 |
| cardiac-mit | 220 | 228 | **0.1905** | 5.304 |
| Aci-D21 | 851 | 1758 | 141.9 | **46.17** |
| Aci-MR95 | 994 | 2859 | 325.1 | **61.34** |
| Abi-49176 | 1069 | 2951 | 179.4 | **74.22** |
| Aci-20731 | 1090 | 2946 | **0.7124** | 88.82 |
| Aci-PHEA | 1561 | 4640 | **2.343** | 100.8 |
| iAF1260 | 2382 | 6368 | 537.9 | **170.5** |
| iJO1366 | 2583 | 7284 | 537.9 | **169.4** |
| Recon1 | 3742 | 8717 | **3.740** | 3280 |

Table 1: Real-world data setting: comparison with Kook et al. (2022a).

algorithms by computing *the sampling time per effective sample* (in seconds), defined as the total sampling time until termination divided by the Effective Sample Size. We set the initial step-size to 0.01 for both algorithms and $\eta = 10$ in n-BHMC. We then follow the exact same procedure as in (Kook et al., 2022a, Table 1) by drawing 1000 uniform samples with limit on running time set to 1 day. Results are given in Table 1. For each molecular model, we specify by NNZ the number of non-zero entries in the matrix A defining the corresponding pre-processed polytope, which thus reflects its complexity. In particular, the sampling dimension $d$ corresponds to the number of columns of A. Note that the sampling time values are not of interest in themselves, but are only meant to compare CRHMC and n-BHMC. While it is clear that adding an "involution checking step" implies a trade-off between accuracy and complexity of the method, our results demonstrate that it does not penalize BHMC in the considered settings, and may even make it more efficient in some cases.

## 7 Discussion

In this paper, we introduced a novel version of RMHMC, Barrier HMC (BHMC), which addresses the problem of sampling from a distribution $\pi$ over a constrained convex subset $\mathsf{M} \subset \mathbb{R}^d$ equipped with a self-concordant barrier $\phi$. Our contribution highlights that the use of well known implicit schemes for ODE integration combined with these space constraints leads to asymptotic bias when it comes to their numerical implementation. We propose to solve it in a straightforward manner via our algorithm, n-BHMC, which relies on an additional "involution checking step". Under reasonable assumptions, our theory shows that this critical step removes the asymptotic bias. This result is supported by numerical experiments where we highlight this lack of bias compared to state-of-the-art methods. In future work, we would like to investigate the "coupled" behaviour of the hyperparameters $h$ and $\eta$ in practice, the study of irreductiblity of n-BHMC and the influence of $\eta$ on the bias. Moreover, we are interested in the application of n-BHMC for efficient polytope volume computation. More generally, we are convinced that our study is a first step for future work on designing implicit integration schemes for unbiased constrained sampling methods.

## Acknowledgments

AD acknowledges support from the Lagrange Mathematics and Computing Research Center. AD and MN would like to thank the Isaac Newton Institute for Mathematical Sciences for support and hospitality during the programme *The mathematical and statistical foundation of future data-driven engineering* when work on this paper was undertaken. MN acknowledges funding from the grant SCAI (ANR-19-CHIA-0002).

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

## Organisation of the supplementary

This appendix is organized as follows. Appendix A summarizes general facts that will be useful for proofs. Appendix B and Appendix C provide additional details on Riemannian metrics and technicalities on self-concordance, respectively. The c-BHMC algorithm is presented in Appendix D with some results whose proofs are given in Appendix E. Appendix F presents general facts about numerical integration and specific facts on the integrators used in n-BHMC. Appendix G dispenses the proof of the result on implicit integrators of n-BHMC, stated in Section 4.1. Appendix H presents the modified version of n-BHMC, incorporating a step-size condition, for which we state the reversibility with respect to the target distribution in Section 4.2. Proof of this last result is given in Appendix I. A detailed comparison between n-BHMC and CRHMC (Kook et al., 2022a) is given in Appendix J. Finally, Appendix K provides more details on the experiments of Section 6.

**Notation.** For any Riemannian manifold $(\mathsf{M}, \mathfrak{g})$ and any $z = (x, p) \in \mathrm{T}^\star \mathsf{M}$, we denote by $\|\cdot\|_z$ the norm (7) defined on $\mathrm{T}^\star \mathsf{M}$ by $\|(x', p')\|_z = \|x'\|_{\mathfrak{g}(x)} + \|p'\|_{\mathfrak{g}(x)^{-1}}$ for any $(x', p') \in \mathrm{T}^\star \mathsf{M}$. For any open subset $\mathsf{U} \subset \mathbb{R}^d$ and any $k \in \mathbb{N}$, we denote by $\mathrm{C}^k(\mathsf{U}, \mathbb{R}^{d'})$ the set of functions $f : \mathsf{U} \to \mathbb{R}^{d'}$ such that $f$ is $k$ times continuously differentiable.

## A   Useful facts and lemmas

We recall here some basic knowledge on linear algebra and probability, and state useful inequalities for our proofs.

**Linear algebra reminders.** For any matrices $\mathrm{A}, \mathrm{B} \in \mathbb{R}^{d \times d}$, we write $\mathrm{A} \preceq \mathrm{B}$ if for any $x \in \mathbb{R}^d$, $x^\top (\mathrm{B} - \mathrm{A}) x \geq 0$. Any positive-definite matrix $\mathrm{A} \in \mathbb{R}^{d \times d}$ induces a scalar product $\langle \cdot, \cdot \rangle_\mathrm{A}$ on $\mathbb{R}^d$, defined by $\langle x, y \rangle_\mathrm{A} = \langle x, \mathrm{A}y \rangle$. This scalar product induces the norm $\|\cdot\|_\mathrm{A}$ on $\mathbb{R}^d$, defined by $\|x\|_\mathrm{A} = \sqrt{\langle x, x \rangle_\mathrm{A}} = \|\mathrm{A}^{1/2} x\|_2$. For any positive-definite matrices $\mathrm{A}, \mathrm{B} \in \mathbb{R}^{d \times d}$ and for any $\alpha \geq 0$, $\mathrm{A} \preceq \alpha \mathrm{B}$ is equivalent to $\|\cdot\|_\mathrm{A} \leq \sqrt{\alpha} \|\cdot\|_\mathrm{B}$. The canonical norm of $\mathbb{R}^d$ induces the norm $\|\cdot\|_2$ on $\mathbb{R}^{d \times d}$, defined by $\|\mathrm{A}\|_2 = \sup_{\|u\|_2 = 1} \|\mathrm{A}u\|_2$. In particular, for any matrices $\mathrm{A}, \mathrm{B} \in \mathbb{R}^{d \times d}$ and any vector $x \in \mathbb{R}^d$, $\|\mathrm{A}x\|_2 \leq \|\mathrm{A}\|_2 \|x\|_2$ and $\|\mathrm{A}\mathrm{B}\|_2 \leq \|\mathrm{A}\|_2 \|\mathrm{B}\|_2$. Moreover, if $\mathrm{A}$ is non-negative-definite, then $\|\mathrm{A}\|_2 = \lambda(\mathrm{A})$, where $\lambda(\mathrm{A})$ is the largest eigenvalue of $\mathrm{A}$, and $\|\mathrm{A}^\alpha\|_2 = \lambda(\mathrm{A})^\alpha$ for any $\alpha > 0$.

**Probability reminders.** In this section, we consider a smooth manifold $\mathsf{M} \subset \mathbb{R}^d$. We first recall the definition of reversibility (Douc et al., 2018, Definition 1.5.1) before stating general results on reversibility.

**Definition 4.** *Let $\mathrm{Q} : \mathrm{T}^\star \mathsf{M} \times \mathcal{B}(\mathrm{T}^\star \mathsf{M}) \to [0, 1]$ be a transition probability kernel and let $\bar{\pi}$ be a probability distribution on $\mathrm{T}^\star \mathsf{M}$. Then,*

*(a) $\mathrm{Q}$ is said to be reversible with respect to $\bar{\pi}$ if for any $f \in \mathrm{C}(\mathrm{T}^\star \mathsf{M} \times \mathrm{T}^\star \mathsf{M}, \mathbb{R})$ with compact support*

$$\int_{\mathrm{T}^\star \mathsf{M} \times \mathrm{T}^\star \mathsf{M}} f(z, z') \mathrm{Q}(z, \mathrm{d}z') \bar{\pi}(\mathrm{d}z) = \int_{\mathrm{T}^\star \mathsf{M} \times \mathrm{T}^\star \mathsf{M}} f(z', z) \mathrm{Q}(z, \mathrm{d}z') \bar{\pi}(\mathrm{d}z) \ .$$

*(b) $\mathrm{Q}$ is said to be reversible up to momentum reversal with respect to $\bar{\pi}$ if for any $f \in \mathrm{C}(\mathrm{T}^\star \mathsf{M} \times \mathrm{T}^\star \mathsf{M}, \mathbb{R})$ with compact support*

$$\int_{\mathrm{T}^\star \mathsf{M} \times \mathrm{T}^\star \mathsf{M}} f(z, z') \mathrm{Q}(z, \mathrm{d}z') \bar{\pi}(\mathrm{d}z) = \int_{\mathrm{T}^\star \mathsf{M} \times \mathrm{T}^\star \mathsf{M}} f(s(z'), s(z)) \mathrm{Q}(z, \mathrm{d}z') \bar{\pi}(\mathrm{d}z) \ ,$$

*where we recall that for any $(x, p) \in \mathbb{R}^d \times \mathbb{R}^d$, $s(x, p) = (x, -p)$.*

**Lemma 5.** *Let $\mathrm{Q} : \mathrm{T}^\star \mathsf{M} \times \mathcal{B}(\mathrm{T}^\star \mathsf{M}) \to [0, 1]$ be a transition probability kernel and let $\bar{\pi}$ be a probability distribution on $\mathrm{T}^\star \mathsf{M}$. Assume that $s_\# \bar{\pi} = \bar{\pi}$ and that $\mathrm{Q}$ is reversible up to momentum reversal with respect to $\bar{\pi}$. Then $\bar{\pi}$ is an invariant measure for $\mathrm{Q}$, i.e., for any $f \in \mathrm{C}(\mathrm{T}^\star \mathsf{M}, \mathbb{R})$ with compact support*

$$\int_{\mathrm{T}^\star \mathsf{M} \times \mathrm{T}^\star \mathsf{M}} f(z') \mathrm{Q}(z, \mathrm{d}z') \bar{\pi}(\mathrm{d}z) = \int_{\mathrm{T}^\star \mathsf{M} \times \mathrm{T}^\star \mathsf{M}} f(z) \bar{\pi}(\mathrm{d}z) \ .$$

*Proof.* Let $f \in \mathrm{C}(\mathrm{T}^\star\mathsf{M}, \mathbb{R})$ with compact support. We have

$$\int_{\mathrm{T}^\star\mathsf{M}\times\mathrm{T}^\star\mathsf{M}} f(z')\mathrm{Q}(z, \mathrm{d}z')\bar{\pi}(\mathrm{d}z)$$
$$= \int_{\mathrm{T}^\star\mathsf{M}\times\mathrm{T}^\star\mathsf{M}} f(s(z))\mathrm{Q}(z, \mathrm{d}z')\bar{\pi}(\mathrm{d}z) \qquad \text{(Definition 4)}$$
$$= \int_{\mathrm{T}^\star\mathsf{M}\times\mathrm{T}^\star\mathsf{M}} f(z)\mathrm{Q}(s(z), \mathrm{d}z')(s_\#\bar{\pi})(\mathrm{d}z) = \int_{\mathrm{T}^\star\mathsf{M}} f(z)\bar{\pi}(\mathrm{d}z) \quad \text{(momentum reversal on } z) \,,$$

which concludes the proof. $\qquad\square$

**Lemma 6.** *Let* $\mathrm{Q} : \mathrm{T}^\star\mathsf{M} \times \mathcal{B}(\mathrm{T}^\star\mathsf{M}) \to [0, 1]$ *be a transition probability kernel and let* $\bar{\pi}$ *be a probability distribution on* $\mathrm{T}^\star\mathsf{M}$. *Assume that* $s_\#\mathrm{Q} = \mathrm{Q}$ *and that* $\mathrm{Q}$ *is reversible with respect to* $\bar{\pi}$. *Then,* $\mathrm{Q}$ *is reversible up to momentum reversal with respect to* $\bar{\pi}$.

*Proof.* Let $f \in \mathrm{C}(\mathrm{T}^\star\mathsf{M}, \mathbb{R})$ with compact support. We have

$$\int_{\mathrm{T}^\star\mathsf{M}\times\mathrm{T}^\star\mathsf{M}} f(z, z')\mathrm{Q}(z, \mathrm{d}z')\bar{\pi}(\mathrm{d}z)$$
$$= \int_{\mathrm{T}^\star\mathsf{M}\times\mathrm{T}^\star\mathsf{M}} f(z, s(z'))(s_\#\mathrm{Q})(z, \mathrm{d}z')\bar{\pi}(\mathrm{d}z) \qquad \text{(momentum reversal on } z')$$
$$= \int_{\mathrm{T}^\star\mathsf{M}\times\mathrm{T}^\star\mathsf{M}} f(z', s(z))\mathrm{Q}(z, \mathrm{d}z')\bar{\pi}(\mathrm{d}z) \qquad \text{(Definition 4)}$$
$$= \int_{\mathrm{T}^\star\mathsf{M}\times\mathrm{T}^\star\mathsf{M}} f(s(z'), s(z))(s_\#\mathrm{Q})(z, \mathrm{d}z')\bar{\pi}(\mathrm{d}z) \qquad \text{(momentum reversal on } z')$$
$$= \int_{\mathrm{T}^\star\mathsf{M}\times\mathrm{T}^\star\mathsf{M}} f(s(z'), s(z))\mathrm{Q}(z, \mathrm{d}z')\bar{\pi}(\mathrm{d}z) \,,$$

which concludes the proof. $\qquad\square$

**Useful inequalities.** The following inequalities hold:

(a) For any $u \in [0, 1/5]$, $(1 - u)^{-2} \le 1 + 3u$ and $(1 - u)^{-3} \le 1 + 5u$.

(b) For any $u \in [0, 1/2]$, $(1 - u)^{-1} \le 1 + 2u$.

(c) For any $u \in [0, 1]$, $|(1 - u)^2 - 1| \le 3u$.

(d) For any $u \ge 0$, we have $1 - (1 + u)^{-1} \le u$ and $(1 - u)^2 - 1 \ge -2u$.

# B  Details on Riemannian metrics

Let $\mathsf{M}$ be a smooth $d$-dimensional manifold, endowed with a metric $\mathfrak{g}$. We recall that the *Riemannian volume element* corresponding to $(\mathsf{M}, \mathfrak{g})$ is given in local coordinates by $\mathrm{dvol}_\mathsf{M}(x) = \sqrt{\det(\mathfrak{g})}\mathrm{d}x$, where $\mathrm{d}x$ is a dual coframe, (Lee, 2006, Lemma 3.2.). For any $x \in \mathsf{M}$, we respectively denote by $\mathrm{T}_x\mathsf{M}$ and $\mathrm{T}_x^\star\mathsf{M}$, the tangent space at $x$ and its dual space, *i.e.*, the cotangent space at $x$. Note that $\mathrm{T}_x\mathsf{M}$ and $\mathrm{T}_x^\star\mathsf{M}$ are space vectors, and $\mathrm{T}_x\mathsf{M}$ is endowed with the scalar product $\langle\cdot, \cdot\rangle_{\mathfrak{g}(x)}$ by definition of the Riemannian metric. For clarity sake, we denote by $v$ (resp. $p$) an element of the tangent (resp. cotangent) space. We recall that the tangent bundle $\mathrm{T}\mathsf{M}$ and the cotangent bundle $\mathrm{T}^\star\mathsf{M}$ are respectively defined by $\mathrm{T}\mathsf{M} = \sqcup_{x\in\mathsf{M}}\{x\} \cup \mathrm{T}_x\mathsf{M}$ and $\mathrm{T}^\star\mathsf{M} = \sqcup_{x\in\mathsf{M}}\{x\} \cup \mathrm{T}_x^\star\mathsf{M}$. These two sets are $2d$-dimensional manifolds.

**Metric on the tangent bundle** $\mathrm{T}\mathsf{M}$. Sasaki (1958) originally introduced on $\mathrm{T}\mathsf{M}$ a Riemannian metric $\hat{\mathfrak{g}}$, which, among other properties, preserves the Euclidean metric induced by $\mathfrak{g}$ on each tangent space. This metric is defined by

$$\hat{\mathfrak{g}} = \begin{pmatrix} \mathfrak{g}_{ij} + v^k v^l \Gamma_{is}^k \Gamma_{tj}^l \mathfrak{g}^{st} & -\mathfrak{g}^{js}v^k\Gamma_{is}^k \\ -\mathfrak{g}^{is}v^k\Gamma_{js}^k & \mathfrak{g}_{ij} \end{pmatrix} \,,$$

where $\mathfrak{g}_{ij}$ and $\mathfrak{g}^{ij}$ respectively refer to $\mathfrak{g}$ and $\mathfrak{g}^{-1}$, and $\Gamma$ corresponds to the Christoffel symbol. Since $(\mathrm{T}\mathsf{M}, \hat{\mathfrak{g}})$ is a Riemannian manifold, the volume form on $\mathrm{T}\mathsf{M}$ satisfies for any $(x, v) \in \mathrm{T}\mathsf{M}$

$$\mathrm{dvol}_{\mathrm{T}\mathsf{M}}(x, v) = \sqrt{\det(\hat{\mathfrak{g}}(x, v))}\mathrm{d}x\mathrm{d}v = \det(\mathfrak{g}(x))\mathrm{d}x\mathrm{d}v \,.$$

**Metric on the cotangent bundle** $T^\star M$.    Elaborating on the metric defined by Sasaki (1958), Mok (1977) showed that for any $x \in M$, $T_x^\star M$ is naturally endowed with the scalar product $\langle \cdot, \cdot \rangle_{\mathfrak{g}(x)^{-1}}$, and proposed a Riemannian metric $\mathfrak{g}^\star$ on $T^\star M$, which notably preserves this result on each cotangent space. This metric is closely related to $\hat{\mathfrak{g}}$ and is defined by

$$\mathfrak{g}^\star = \begin{pmatrix} \mathfrak{g}_{ij} + p^k p^l \Gamma_{is}^k \Gamma_{tj}^l \mathfrak{g}^{st} & -\mathfrak{g}^{js} p^k \Gamma_{is}^k \\ -\mathfrak{g}^{is} p^k \Gamma_{js}^k & \mathfrak{g}^{ij} \end{pmatrix} .$$

Since $(T^\star M, \mathfrak{g}^\star)$ is a Riemannian manifold, the volume form on $T^\star M$ satisfies for any $(x, p) \in T^\star M$

$$\mathrm{dvol}_{T^\star M}(x, p) = \sqrt{\det(\mathfrak{g}^\star(x, p))} \mathrm{d}x \mathrm{d}p = \mathrm{d}x \mathrm{d}p .$$

In contrast to the tangent bundle, the volume form on the cotangent bundle does not depend on $\mathfrak{g}$, which motivates to augment on $T^\star M$ (instead of $TM$) any target measure $\pi$ defined on $M$.

## C    Properties of self-concordance

We first recall the definition of self-concordance and derive some of its properties in Lemmas 8 and 9.

**Definition 7** (Nesterov & Nemirovskii (1994))**.**  *Let* $U$ *be a non-empty open convex domain in* $\mathbb{R}^d$. *A function* $\phi : U \to \mathbb{R}$ *is said to be a* $\nu$-*self-concordant barrier (with* $\nu \geq 1$*) on* $U$ *if it satisfies:*

*(a)* $\phi \in C^3(U, \mathbb{R})$ *and* $\phi$ *is convex,*

*(b)* $\phi(x) \to +\infty$ *as* $x \to \partial U$,

*(c)* $|D^3 \phi(x)[h, h, h]| \leq 2 \|h\|_{\mathfrak{g}(x)}^3$, *for any* $x \in M$, $h \in \mathbb{R}^d$,

*(d)* $|D\phi(x)[h]|^2 \leq \nu \|h\|_{\mathfrak{g}(x)}^2$, *for any* $x \in M$, $h \in \mathbb{R}^d$,

*where* $\mathfrak{g}(x) = D^2 \phi(x)$.

Self-concordance can be thought as a certain kind of regularity. Indeed, if $\phi$ is a $\nu$-self-concordant barrier on a convex body $U$, then $D^2 \phi$ is 2-Lipschitz continuous on $U$ with respect to the local norm induced by $\mathfrak{g}$ (see Property (c)), and more restrictively, $\phi$ is $\nu$-Lipschitz continuous with respect to the same norm (see Property (d)).

**Lemma 8.**  *Let* $\phi : U \to \mathbb{R}$ *be a* $\nu$-*self-concordant barrier with* $\mathfrak{g} = D^2 \phi$. *Assume that* $U$ *is bounded. Then,* $\|\nabla \phi(x)\|_2 \to +\infty$ *and* $\|\mathfrak{g}(x)\|_2 \to +\infty$ *as* $x \to \partial U$.

*Proof.*  Since $\phi$ is convex, we have, for any $(x, y) \in U^2$

$$\begin{aligned} \phi(x) &\leq \phi(y) + \nabla \phi(x)(x - y) , \\ |\phi(x)| &\leq |\phi(y)| + \|\nabla \phi(x)\|_2 \|x - y\|_2 \\ &\leq |\phi(y)| + \|\nabla \phi(x)\|_2 \mathrm{diam}(U) , \end{aligned}$$

where we used Cauchy-Schwartz inequality in the second line. Using Property (b) of $\phi$, we obtain that $\|\nabla \phi(x)\|_2 \to +\infty$ as $x \to \partial U$. By combining this result with Property (d) of $\phi$, we obtain that $\|\mathfrak{g}(x)\|_2 \to +\infty$ as $x \to \partial U$. $\qquad \square$

According to Nesterov (2003) (see Lemma 4.1.2), Property (c) of self-concordance is equivalent to

$$|D^3 \phi(x)[h_1, h_2, h_3]| \leq 2 \prod_{i=1}^3 \|h_i\|_{\mathfrak{g}(x)} ,$$

for any $x \in M$ and any $(h_1, h_2, h_3) \in \mathbb{R}^d \times \mathbb{R}^d \times \mathbb{R}^d$.

Let $u : \mathbb{R}^d \to \mathbb{R}^d$ be a linear operator. For any $x \in \mathbb{R}^d$, we define the operator norms $\|u\|_{\mathfrak{g}(x)}$ and $\|u\|_{\mathfrak{g}(x)^{-1}}$ by

$$\begin{aligned} \|u\|_{\mathfrak{g}(x)} &= \sup\{u(\mathfrak{g}(x)h) \ : \ \|h\|_{\mathfrak{g}(x)} = 1\} , \\ \|u\|_{\mathfrak{g}(x)^{-1}} &= \sup\{u(\mathfrak{g}(x)^{-1}h) \ : \ \|h\|_{\mathfrak{g}(x)^{-1}} = 1\} . \end{aligned}$$

In addition, a set $V \subset \mathbb{R}^d$ contains no straight line if for any $D_{x_0} = \{t x_0 \ : \ t \in \mathbb{R}\}$ with $x_0 \neq 0$, we have $V^c \cap D_{x_0} \neq \emptyset$.

**Lemma 9.** *Let $\phi : \mathsf{U} \to \mathbb{R}$ be a $\nu$-self-concordant barrier with $\mathfrak{g} = D^2\phi$. Assume that $\mathsf{U}$ contains no straight line. For any $x \in \mathsf{U}$ and any $r > 0$, we denote by $\mathsf{W}^0(x,r)$ the open Dikin ellipsoid of $\phi$ at $x$, given by $\mathsf{W}^0(x,r) = \{y \in \mathbb{R}^d \mid \|y - x\|_{\mathfrak{g}(x)} < r\}$. Then, the following properties hold:*

*(a) For any $x \in \mathsf{U}$ and any $(h_1, h_2) \in \mathbb{R}^d \times \mathbb{R}^d$*

$$\|D^3\phi(x)[h_1, h_2]\|_{\mathfrak{g}(x)^{-1}} \leq 2\|h_1\|_{\mathfrak{g}(x)}\|h_2\|_{\mathfrak{g}(x)} .$$

*(b) For any $x \in \mathsf{U}$, $\mathsf{W}^0(x,1) \subset \mathsf{U}$, and for any $y \in \mathsf{W}^0(x,1)$*

$$(1 - \|y - x\|_{\mathfrak{g}(x)})^2 \mathfrak{g}(x) \preceq \mathfrak{g}(y) \preceq (1 - \|y - x\|_{\mathfrak{g}(x)})^{-2}\mathfrak{g}(x) .$$

*(c) For any $x \in \mathsf{U}$, any $y \in \mathsf{W}^0(x,1)$ and any $h \in \mathbb{R}^d$, the following hold*

$$\|h\|_{\mathfrak{g}(x)} \leq (1 - \|y - x\|_{\mathfrak{g}(x)})^{-1}\|h\|_{\mathfrak{g}(y)} ,$$
$$\|h\|_{\mathfrak{g}(y)} \leq (1 - \|y - x\|_{\mathfrak{g}(x)})^{-1}\|h\|_{\mathfrak{g}(x)} ,$$
$$\|h\|_{\mathfrak{g}(x)^{-1}} \leq (1 - \|y - x\|_{\mathfrak{g}(x)})^{-1}\|h\|_{\mathfrak{g}(y)^{-1}} ,$$
$$\|h\|_{\mathfrak{g}(y)^{-1}} \leq (1 - \|y - x\|_{\mathfrak{g}(x)})^{-1}\|h\|_{\mathfrak{g}(x)^{-1}} .$$

*Proof.* The result is a direct consequence of (Nesterov, 2003, Theorem 4.1.3, Theorem 4.1.5, Theorem 4.1.6). $\qquad\square$

We now introduce $\alpha$-regularity, which slightly strengthens self-concordance, by ensuring that $D^3\phi$ is $\alpha(\alpha + 1)$-Lipschitz continuous with respect to the local norm induced by $\mathfrak{g}$ (see (11)). We state below the definition of $\alpha$-regularity as well as some of its properties in Lemma 11.

**Definition 10** (Nesterov & Nemirovski (1998))**.** *Let $\alpha \geq 1$ and $\mathsf{U}$ be non-empty open convex domain in $\mathbb{R}^d$. A self-concordant function $\phi$ on $\mathsf{U}$ is said $\alpha$-regular, if $\phi \in \mathrm{C}^4(\mathsf{U}, \mathbb{R})$ and for any $x \in \mathsf{U}$ and any $h \in \mathbb{R}^d$*

$$|D^4\phi(x)[h, h, h, h]| \leq \alpha(\alpha + 1)\|h\|^2_{\mathfrak{g}(x)}\|h\|^2_{\mathsf{U},x} ,$$

*where $\|h\|_{\mathsf{U},x} := \inf\{t^{-1} \mid t > 0, x \pm th \in \mathsf{U}\}$.*

**Lemma 11.** *Let $\phi : \mathsf{U} \to \mathbb{R}$ be a $\alpha$-regular function with $\mathfrak{g} = D^2\phi$. Assume that $\mathsf{U}$ contains no straight line.*

*(a) For any $x \in \mathsf{U}$ and any $h \in \mathbb{R}^d$*

$$|D^4\phi(x)[h, h, h, h]| \leq \alpha(\alpha + 1)\|h\|^4_{\mathfrak{g}(x)} . \tag{11}$$

*(b) For any $x \in \mathsf{U}$, any $(h_1, h_2, h_3, h_4) \in \mathbb{R}^d \times \mathbb{R}^d \times \mathbb{R}^d \times \mathbb{R}^d$*

$$|D^4\phi(x)[h_1, h_2, h_3, h_4]| \leq \alpha(\alpha + 1)\prod_{i=1}^{4}\|h_i\|_{\mathfrak{g}(x)} . \tag{12}$$

*(c) For any $x \in \mathsf{U}$, any $y \in \mathsf{W}^0(x,1)$ and any $(h_1, h_2) \in \mathbb{R}^d \times \mathbb{R}^d$*

$$\|D^3\phi(x)[h_1, h_2] - D^3\phi(y)[h_1, h_2]\|_{\mathfrak{g}(x)^{-1}} \leq \alpha(\alpha + 1)/3\|h_1\|_{\mathfrak{g}(x)}\|h_2\|_{\mathfrak{g}(x)}((1 - \|y - x\|_{\mathfrak{g}(x)})^{-3} - 1) . \tag{13}$$

*Proof.* We first prove (11). First remark that the result is true for $h = 0$. Consider now $h \neq 0$. Let $\varepsilon > 0$. We define $t_\varepsilon = (1 + \varepsilon)^{-1}\|h\|^{-1}_{\mathfrak{g}(x)} > 0$, $y_\varepsilon^+ = x + t_\varepsilon h$ and $y_\varepsilon^- = x - t_\varepsilon h$. We have $\|x - y_\varepsilon^+\|_{\mathfrak{g}(x)} = \|x - y_\varepsilon^-\|_{\mathfrak{g}(x)} = 1/(1 + \varepsilon) < 1$. Hence, $y_\varepsilon^- \in \mathsf{U}$ and $y_\varepsilon^+ \in \mathsf{U}$ by Lemma 9. Therefore, $\|h\|_{\mathsf{U},x} \leq t_\varepsilon^{-1}$ and

$$|D^4\phi(x)[h, h, h, h]| \leq \alpha(\alpha + 1)\|h\|^4_{\mathfrak{g}(x)}(1 + \varepsilon)^2 .$$

We conclude upon taking $\varepsilon \to 0$. Then, (12) is an immediate consequence of (11) using (Nesterov & Nemirovskii, 1994, Proposition 9.1.1). We now prove (13). Using Lemma 9, we have

$$\|D^3\phi(x)[h_1, h_2] - D^3\phi(y)[h_1, h_2]\|_{\mathfrak{g}(x)^{-1}}$$
$$\leq \int_0^1 \|D^4\phi(x + t(y-x))[h_1, h_2, y-x]\|_{\mathfrak{g}(x)^{-1}} \mathrm{d}t$$
$$\leq \int_0^1 (1 - t\|y-x\|_{\mathfrak{g}(x)})^{-1} \|D^4\phi(x + t(y-x))[h_1, h_2, y-x]\|_{\mathfrak{g}(x+t(y-x))^{-1}} \mathrm{d}t$$
$$\leq \int_0^1 \alpha(\alpha+1)/(1 - t\|y-x\|_{\mathfrak{g}(x)}) \|h_1\|_{\mathfrak{g}(x+t(y-x))} \|h_2\|_{\mathfrak{g}(x+t(y-x))} \|y-x\|_{\mathfrak{g}(x+t(y-x))} \mathrm{d}t$$
$$\leq \alpha(\alpha+1)\|h_1\|_{\mathfrak{g}(x)} \|h_2\|_{\mathfrak{g}(x)} \int_0^1 \|y-x\|_{\mathfrak{g}(x)}/(1 - t\|y-x\|_{\mathfrak{g}(x)})^4 \mathrm{d}t$$
$$\leq \alpha(\alpha+1)/3 \|h_1\|_{\mathfrak{g}(x)} \|h_2\|_{\mathfrak{g}(x)}((1 - \|y-x\|_{\mathfrak{g}(x)})^{-3} - 1) ,$$

which concludes the proof. $\qquad\square$

We now prove the equivalence between the Euclidean norm and the norms induced by $\mathfrak{g}$ and $\mathfrak{g}^{-1}$.

**Lemma 12.** *Let* $(\mathsf{U}, \mathfrak{g})$ *be non-empty Riemannian manifold in* $\mathbb{R}^d$*. For any* $x_0 \in \mathsf{U}$*, any* $(x, p) \in \mathsf{U} \times \mathbb{R}^d$*, we have*

$$\|\mathfrak{g}(x_0)^{-1}\|_2^{-1/2}\|x\|_2 \leq \|x\|_{\mathfrak{g}(x_0)} \quad \leq \|\mathfrak{g}(x_0)\|_2^{1/2}\|x\|_2 ,$$
$$\|\mathfrak{g}(x_0)\|_2^{-1/2}\|p\|_2 \leq \|p\|_{\mathfrak{g}(x_0)^{-1}} \leq \|\mathfrak{g}(x_0)^{-1}\|_2^{1/2}\|p\|_2 .$$

*In addition, let*

$$C_{x_0} = \max(\|\mathfrak{g}(x_0)\|_2^{1/2}, \|\mathfrak{g}(x_0)^{-1}\|_2^{1/2}) > 0 .$$

*Then, we have*

$$1/C_{x_0}(\|x\|_2 + \|p\|_2) \leq \|x\|_{\mathfrak{g}(x_0)} + \|p\|_{\mathfrak{g}(x_0)^{-1}} \leq C_{x_0}(\|x\|_2 + \|p\|_2) .$$

# D   C-BHMC: Algorithm and Results

In this section, we first state general results on the Hamiltonian dynamics (3) under our main assumptions. It allows us to introduce the *continuous* version of BHMC (c-BHMC), for which we derive the reversibility with respect to $\pi$. Full proofs of this section are provided in Appendix E.

**Results on the Hamiltonian dynamics.**   Using Cauchy-Lipschitz's theorem, we obtain in Proposition 13 the existence and uniqueness of the Hamiltonian dynamics (3), starting from any $z_0 \in \mathrm{T}^\star\mathsf{M}$.

**Proposition 13.** *Assume* **A**1*,* **A**2*. Let* $z_0 \in \mathrm{T}^\star\mathsf{M}$*. Then* (3) *admits a unique maximal solution* $(J_{z_0}, z)$*, where (i)* $J_{z_0} \subset \mathbb{R}$ *is an open neighbourhood of* 0*, (ii)* $z \in \mathrm{C}^1(J_{z_0}, \mathrm{T}^\star\mathsf{M})$*, (iii)* $z(0) = z_0$ *and (iv) for any* $t \in J_{z_0}$*,* $H(z(t)) = H(z_0)$*.*

In contrast to the Euclidean case (*i.e.*, $\mathfrak{g}(x) = \mathrm{I}_d$), the Hamiltonian dynamics (3) relies on a metric derived from a self-concordant barrier, and thus is defined up to an explosion time. In the next result, we characterize the behaviour of the solutions when this explosion time is finite.

**Proposition 14.** *Assume* **A**1*,* **A**2*. Let* $z_0 \in \mathrm{T}^\star\mathsf{M}$*,* $(J_{z_0}, z)$ *as in Proposition 13. Assume* $T_{z_0} = \sup J_{z_0} < +\infty$*, then we have*

*(a)* $\lim_{t \to T_{z_0}} \|p(t)\|_2 = +\infty$*.*

*(b) There exists* $x_\mathsf{M} \in \partial\mathsf{M}$ *such that* $\lim_{t \to T_{z_0}} x(t) = x_\mathsf{M}$*.*

In the rest of this section, we define the *Hamiltonian flow* as the map $\mathrm{T} : \mathsf{D} \to \mathrm{T}^\star\mathsf{M}$, where $\mathsf{D} = \{(h, z_0) : z_0 \in \mathrm{T}^\star\mathsf{M}, h < \sup J_{z_0}\}$, such that $\mathrm{T}_h(z_0) = z(h)$ for any $(h, z_0) \in \mathsf{D}$, $z$ being uniquely defined from $z_0$ in Proposition 13.

**Introduction to c-BHMC.**   In the case where $\mathsf{D} = \mathrm{T}^\star\mathsf{M} \times \mathbb{R}$, the definition of the Hamiltonian dynamics ensures that $\mathrm{T}_h$ is involutive (up to momentum reversal) for any $h > 0$, which is key to derive reversibility in BHMC. However, in our manifold setting, there is one subtlety that needs to be checked, namely that the Hamiltonian flow does not leave the cotangent bundle in finite time. As detailed in Proposition 14, there is indeed no guarantee that this flow is defined for all times, *i.e.*, that

we have $D = T^\star M \times \mathbb{R}$, due to the ill-conditioned behavior of the dynamics near the boundary of the manifold. Therefore, given a time horizon $h > 0$, we restrict the cotangent bundle to the points where the Hamiltonian flow is defined up to time $h$. Hence, we define, for any $h > 0$, the sets

$$\mathsf{E}_h = \{z_0 \in T^\star M \,:\, h < \sup J_{z_0}\}\,, \quad \bar{\mathsf{E}}_h = \{z_0 \in \mathsf{E}_h \,:\, h < \sup J_{(s \circ T_h)(z_0)}\}\,.$$

Note that elements of $\bar{\mathsf{E}}_h$ correspond to points $z_0$ for which:

(a) we are able to compute *exactly* the Hamiltonian dynamics (3) starting from $z_0$ on the time interval $[0, h]$, by considering $\{T_t(z_0)\}_{t \in [0,h]}$ (since $\bar{\mathsf{E}}_h \subset \mathsf{E}_h$),

(b) we are also able to compute *exactly* the **reversed** dynamics - up to momentum reversal - starting from $(s \circ T_h)(z_0)$ on the time interval $[0, h]$.

In Appendix E, we prove that $T_h$ is symplectic (up to momentum reversal) on $\bar{\mathsf{E}}_h$ for any $h > 0$, which thus ensures that the Hamiltonian dynamics is *reversible* with respect to $\bar{\pi}$, see Lemma 16.

Let $h > 0$. Our algorithm c-BHMC, whose pseudo-code is provided in Algorithm 2, then proceeds as follows at stage $n \geq 1$. Assume that the current state is $(X_{n-1}, P_{n-1}) \in T^\star M$. First define a partial refreshment of the momentum $\tilde{P}_n \leftarrow \sqrt{1 - \beta} P_{n-1} + \sqrt{\beta} G_n$, where $G_n \sim N_x(0, I_d)$ is independent from $P_{n-1}$. Then, verify that $(X_{n-1}, \tilde{P}_n) \in \bar{\mathsf{E}}_h$, which ensures that the Hamiltonian flow is involutive on the time interval $[0, h]$, starting at this state. In the same spirit as in Algorithm 1, we refer to this step as an "involution checking step", which is highlighted in yellow in Algorithm 2, although the involution property directly derives from the Hamiltonian dynamics. If this step is valid, define $(X_n, \bar{P}_n) = T_h(X_{n-1}, \tilde{P}_n)$; otherwise, set $(X_n, \bar{P}_n) \leftarrow s(X_{n-1}, \tilde{P}_n)$. Finally, update the momentum as in the beginning of the iteration to obtain the new state $(X_n, P_n)$.

---

**Algorithm 2:** c-BHMC with Momentum Refresh

**Input:** $(X_0, P_0) \in T^\star M$, $\beta \in (0, 1]$, $N \in \mathbb{N}$, $h > 0$
**Output:** $(X_n, P_n)_{n \in [N]}$

1 **for** $n = 1, ..., N$ **do**
2    Step 1:   $G_n \sim N_x(0, I_d), \tilde{P}_n \leftarrow \sqrt{1 - \beta} P_{n-1} + \sqrt{\beta} G_n$
3    Step 2:   solving continuous ODE (3)
4    **if** $(X_{n-1}, \tilde{P}_n) \in \bar{\mathsf{E}}_h$ **then**   $(X_n, \bar{P}_n) \leftarrow T_h(X_{n-1}, \tilde{P}_n)$
5    **else**   $(X_n, \bar{P}_n) \leftarrow s(X_{n-1}, \tilde{P}_n)$
6    Step 3:   $G'_n \sim N_x(0, I_d), P_n \leftarrow \sqrt{1 - \beta} \bar{P}_n + \sqrt{\beta} G'_n$
7 **end**

---

Denote by $Q_c : T^\star M \times \mathcal{B}(T^\star M) \to [0, 1]$, the transition kernel of the (homogeneous) Markov chain $(X_n, P_n)_{n \in [N]}$ obtained with Algorithm 2. Using the properties of the Hamiltonian dynamics, we get the following result.

**Theorem 15.** *Assume* **A**1, **A**2*. Then, $Q_c$ is reversible up to momentum reversal, i.e., we have for any $f \in C(T^\star M \times T^\star M, \mathbb{R})$ with compact support*

$$\int_{T^\star M \times T^\star M} f(z, z') \mathrm{d}\bar{\pi}(z) Q_c(z, \mathrm{d}z') = \int_{T^\star M \times T^\star M} f(s(z'), s(z)) \mathrm{d}\bar{\pi}(z) Q_c(z, \mathrm{d}z')\,.$$

*In particular, $\bar{\pi}$ is an invariant measure for $Q_c$.*

# E   Proofs of Appendix D

## E.1   Existence, uniqueness and explosion time of Hamiltonian dynamics

We prove below Proposition 13 and Proposition 14.

*Proof of Proposition 13.* Assume **A**1, **A**2. In particular, M contains no straight line, and Lemmas 9 and 11 apply here. We rewrite the ODE (3) as a Cauchy problem defined on Banach space $\mathsf{E} = (\mathbb{R}^d \times \mathbb{R}^d, \|\cdot\|_\mathsf{E})$ where $\|(a, b)\|_\mathsf{E} = \|a\|_2 + \|b\|_2$ for any pair $(a, b) \in \mathbb{R}^d \times \mathbb{R}^d$. We define $\Omega = M \times \mathbb{R}^d$, which is an open set in $\mathsf{E}$. A solution $(J, z)$ to this Cauchy problem is defined for all $t \in J$ by

$$\dot{z}(t) = F(z(t))\,, \;\; z(0) = z_0\,, \quad \text{where} \quad F(x, p) = (\partial_p H(x, p), -\partial_x H(x, p))\,,$$

such that $J$ is an open interval of $\mathbb{R}$ with $0 \in J$, and $z = (x, p) : J \to \Omega$ is differentiable on $J$. We now prove two results on $F$ to establish existence and uniqueness of a maximal solution $(J, z)$ for this Cauchy problem:

(a) $F$ is continuously differentiable on $\Omega$.

(b) $F$ is locally Lipschitz-continuous on $\Omega$ with respect to $\|\cdot\|_\mathsf{E}$.

Since $\nabla V, D\mathfrak{g} \in \mathrm{C}^1(\mathsf{M}, \mathbb{R})$, we directly obtain Item (a). We now prove Item (b). Let $\bar{z} = (\bar{x}, \bar{p}) \in \Omega$. We endow $\mathsf{E}$ with the norm $\|\cdot\|_{\bar{z}}$. Since $\mathsf{M}$ is an open subset of $\mathbb{R}^d$, there exists $0 < r \le 1/(11 C_{\bar{x}})$, where we recall that $C_{\bar{x}}$ is defined in Lemma 12, such that $\mathrm{B}_{\|\cdot\|_2}(\bar{x}, r) \subset \mathsf{M}$. We consider such $r$, and we define $\mathsf{B} = \mathrm{B}_{\|\cdot\|_\mathsf{E}}(\bar{z}, r)$ and $\mathsf{B}_\mathfrak{g} = \mathrm{B}_{\|\cdot\|_{\bar{z}}}(\bar{z}, \bar{r})$, where $\bar{r} = C_{\bar{x}} r \le 1/11$. In particular, we have by Lemma 9-(b) that $\mathsf{B}_\mathfrak{g} \subset \mathrm{B}_{\|\cdot\|_{\mathfrak{g}(\bar{x})}}(\bar{x}, \bar{r}) \times \mathrm{B}_{\|\cdot\|_{\mathfrak{g}(\bar{x})^{-1}}}(\bar{p}, \bar{r}) \subset \Omega$ since $\bar{r} < 1$. Moreover, using Lemma 12, we have $\mathsf{B} \subset \mathsf{B}_\mathfrak{g} \subset \Omega$.

Since $V \in \mathrm{C}^2(\mathsf{M}, \mathbb{R})$, $\nabla V$ is Lipschitz-continuous on $\mathsf{B}_\mathfrak{g}$, $i.e.$, there exists $C_V > 0$ such that for any $(z, z') \in \mathsf{B}_\mathfrak{g} \times \mathsf{B}_\mathfrak{g}$,
$$\|\nabla V(x) - \nabla V(x')\|_{\mathfrak{g}(\bar{x})^{-1}} \le C_V \|x - x'\|_{\mathfrak{g}(\bar{x})} . \tag{14}$$

**We first show that $F$ is Lipschitz-continuous on $\mathsf{B}_\mathfrak{g}$ with respect to $\|\cdot\|_{\bar{z}}$.** Consider $(z, z') \in \mathsf{B}_\mathfrak{g} \times \mathsf{B}_\mathfrak{g}$ denoted by $z = (x, p)$ and $z' = (x', p')$. Note that $(x, x') \in W^0(\bar{x}, 1) \times W^0(\bar{x}, 1)$, where $W^0(\bar{x}, 1)$ is defined in Lemma 9. We first bound $\|F^{(1)}(z) - F^{(1)}(z')\|_{\mathfrak{g}(\bar{x})} = \|\partial_p H(z) - \partial_p H(z')\|_{\mathfrak{g}(\bar{x})}$. We have
$$\partial_p H(z) - \partial_p H(z') = \mathfrak{g}(x)^{-1} p - \mathfrak{g}(x')^{-1} p' = \mathfrak{g}(x)^{-1}(p - p') + (\mathfrak{g}(x)^{-1} - \mathfrak{g}(x')^{-1})p' ,$$
then,
$$\|\partial_p H(z) - \partial_p H(z')\|_{\mathfrak{g}(\bar{x})} \le \|\mathfrak{g}(\bar{x})^{1/2}\mathfrak{g}(x)^{-1}\mathfrak{g}(\bar{x})^{1/2}\|_2 \|p - p'\|_{\mathfrak{g}(\bar{x})^{-1}} \tag{15}$$
$$+ \|\mathfrak{g}(\bar{x})^{1/2}(\mathfrak{g}(x)^{-1} - \mathfrak{g}(x')^{-1})\mathfrak{g}(\bar{x})^{1/2}\|_2 \|p'\|_{\mathfrak{g}(\bar{x})^{-1}} .$$

Using Lemma 9-(b), we have
$$(1 - \|\bar{x} - x\|_{\mathfrak{g}(x)})^2 \mathfrak{g}(\bar{x})^{-1} \preceq \mathfrak{g}(x)^{-1} \preceq (1 - \|\bar{x} - x\|_{\mathfrak{g}(x)})^{-2} \mathfrak{g}(\bar{x})^{-1} ,$$
and thus,
$$(1 - \|\bar{x} - x\|_{\mathfrak{g}(x)})^2 \mathrm{I}_d \preceq \mathfrak{g}(\bar{x})^{1/2}\mathfrak{g}(x)^{-1}\mathfrak{g}(\bar{x})^{1/2} \preceq (1 - \|\bar{x} - x\|_{\mathfrak{g}(x)})^{-2} \mathrm{I}_d$$
$$(1 - \bar{r})^2 \mathrm{I}_d \prec \mathfrak{g}(\bar{x})^{1/2}\mathfrak{g}(x)^{-1}\mathfrak{g}(\bar{x})^{1/2} \prec (1 - \bar{r})^{-2} \mathrm{I}_d .$$

Therefore, we have
$$\|\mathfrak{g}(\bar{x})^{1/2}\mathfrak{g}(x)^{-1}\mathfrak{g}(\bar{x})^{1/2}\|_2 \le (1 - \bar{r})^{-2} . \tag{16}$$

In addition, we have
$$\|x - x'\|_{\mathfrak{g}(x)} \le (1 - \|\bar{x} - x'\|_{\mathfrak{g}(\bar{x})})^{-1} \|x - x\|_{\mathfrak{g}(\bar{x})}$$
$$< (1 - \bar{r})^{-1} 2\bar{r}$$
$$< 1/5, \tag{17}$$
where we used Lemma 9-(c) in the first inequality and $\bar{r} \le 1/11$ in the last inequality. Therefore, $x' \in W^0(x, 1)$ and we have using Lemma 9-(b)
$$\{(1 - \|x' - x\|_{\mathfrak{g}(x)})^2 - 1\}\mathfrak{g}(x')^{-1} \preceq \mathfrak{g}(x)^{-1} - \mathfrak{g}(x')^{-1}$$
$$\preceq \{(1 - \|x' - x\|_{\mathfrak{g}(x)})^{-2} - 1\}\mathfrak{g}(x')^{-1}$$
$$\{(1 - \|x' - x\|_{\mathfrak{g}(x)})^2 - 1\}(1 - \bar{r})^2 \mathrm{I}_d \preceq \mathfrak{g}(\bar{x})^{1/2}(\mathfrak{g}(x)^{-1} - \mathfrak{g}(x')^{-1})\mathfrak{g}(\bar{x})^{1/2}$$
$$\preceq \{(1 - \|x' - x\|_{\mathfrak{g}(x)})^{-2} - 1\}(1 - \bar{r})^{-2} \mathrm{I}_d .$$

Then,
$$\|\mathfrak{g}(\bar{x})^{1/2}(\mathfrak{g}(x)^{-1} - \mathfrak{g}(x')^{-1})\mathfrak{g}(\bar{x})^{1/2}\|_2 \tag{18}$$
$$\le \max(|(1 - \|x' - x\|_{\mathfrak{g}(x)})^2 - 1|(1 - \bar{r})^2, \{(1 - \|x' - x\|_{\mathfrak{g}(x)})^{-2} - 1\}(1 - \bar{r})^{-2})$$
$$\le (1 - \bar{r})^{-2} \max(|(1 - \|x' - x\|_{\mathfrak{g}(x)})^2 - 1|, (1 - \|x' - x\|_{\mathfrak{g}(x)})^{-2} - 1) .$$

Using Inequalities (a) and (c) with $u = \|x' - x\|_{\mathfrak{g}(x)}$, where $u \le 1/5$ by (17), we obtain

(a) $|(1 - \|x' - x\|_{\mathfrak{g}(x)})^2 - 1| \le 3\|x' - x\|_{\mathfrak{g}(x)} < 3(1 - \bar{r})^{-1}\|x - x'\|_{\mathfrak{g}(\bar{x})}$,

(b) $(1 - \|x' - x\|_{\mathfrak{g}(x)})^{-2} - 1 \le 3\|x' - x\|_{\mathfrak{g}(x)} < 3(1 - \bar{r})^{-1}\|x - x'\|_{\mathfrak{g}(\bar{x})}$.

Moreover, we have

$$\|p'\|_{\mathfrak{g}(\bar{x})^{-1}} = \|p' - \bar{p} + \bar{p}\|_{\mathfrak{g}(\bar{x})^{-1}} \le \bar{r} + \|\bar{p}\|_{\mathfrak{g}(\bar{x})^{-1}}.$$

Combining (15), (16) and (18), it comes that

$$\|F^{(1)}(z) - F^{(1)}(z')\|_{\mathfrak{g}(\bar{x})} \le (1 - \bar{r})^{-2}\|p - p'\|_{\mathfrak{g}(\bar{x})^{-1}} + 3(1 - \bar{r})^{-3}(\bar{r} + \|\bar{p}\|_{\mathfrak{g}(\bar{x})^{-1}})\|x - x'\|_{\mathfrak{g}(\bar{x})}. \quad (19)$$

We define $A_{\bar{x},1} = (1 - \bar{r})^{-2} + 3(1 - \bar{r})^{-3}(\bar{r} + \|\bar{p}\|_{\mathfrak{g}(\bar{x})^{-1}})$ such that $\|F^{(1)}(z) - F^{(1)}(z')\|_{\mathfrak{g}(\bar{x})} \le A_{\bar{x},1}\|z - z'\|_{\bar{z}}$ and we aim to bound $\|F^{(2)}(z) - F^{(2)}(x')\|_{\mathfrak{g}(\bar{x})^{-1}} = \|\partial_x H(z) - \partial_x H(z')\|_{\mathfrak{g}(\bar{x})^{-1}}$. We have

$$\partial_x H(z) - \partial_x H(z') = \nabla V(x) - \nabla V(x') \quad (20)$$
$$+ \tfrac{1}{2}D\mathfrak{g}(x')[\mathfrak{g}(x')^{-1}p', \mathfrak{g}(x')^{-1}p'] - \tfrac{1}{2}D\mathfrak{g}(x)[\mathfrak{g}(x)^{-1}p, \mathfrak{g}(x)^{-1}p]$$
$$+ \tfrac{1}{2}\mathfrak{g}(x)^{-1} : D\mathfrak{g}(x) - \tfrac{1}{2}\mathfrak{g}(x')^{-1} : D\mathfrak{g}(x')$$

We have

$$\|D\mathfrak{g}(x')[\mathfrak{g}(x')^{-1}p', \mathfrak{g}(x')^{-1}p'] - D\mathfrak{g}(x)[\mathfrak{g}(x)^{-1}p, \mathfrak{g}(x)^{-1}p]\|_{\mathfrak{g}(\bar{x})^{-1}}$$
$$\le \|D\mathfrak{g}(x)[\mathfrak{g}(x)^{-1}p, \mathfrak{g}(x)^{-1}p] - D\mathfrak{g}(x)[\mathfrak{g}(x')^{-1}p', \mathfrak{g}(x')^{-1}p']\|_{\mathfrak{g}(\bar{x})^{-1}}$$
$$+ \|D\mathfrak{g}(x)[\mathfrak{g}(x')^{-1}p', \mathfrak{g}(x')^{-1}p'] - D\mathfrak{g}(x')[\mathfrak{g}(x')^{-1}p', \mathfrak{g}(x')^{-1}p']\|_{\mathfrak{g}(\bar{x})^{-1}} \quad (21)$$

Note that we have $\mathfrak{g}(x)^{-1}p = F^{(1)}(z)$ (resp. $\mathfrak{g}(x')^{-1}p' = F^{(1)}(z')$). Using Lemma 9-(c), the first upper bound in (21) is bounded by

$$(1 - \bar{r})^{-1}\|D\mathfrak{g}(x)[F^{(1)}(z), F^{(1)}(z)] - D\mathfrak{g}(x)[F^{(1)}(z'), F^{(1)}(z')]\|_{\mathfrak{g}(x)^{-1}}$$
$$\le 2(1 - \bar{r})^{-1}\|D\mathfrak{g}(x)[F^{(1)}(z') - F^{(1)}(z), F^{(1)}(z')]\|_{\mathfrak{g}(x)^{-1}}$$
$$+ (1 - \bar{r})^{-1}\|D\mathfrak{g}(x)[F^{(1)}(z') - F^{(1)}(z), F^{(1)}(z') - F^{(1)}(z)]\|_{\mathfrak{g}(x)^{-1}}$$
$$\le 4(1 - \bar{r})^{-1}\|F^{(1)}(z') - F^{(1)}(z)\|_{\mathfrak{g}(x)}\|F^{(1)}(z')\|_{\mathfrak{g}(x)}$$
$$+ 2(1 - \bar{r})^{-1}\|F^{(1)}(z') - F^{(1)}(z)\|_{\mathfrak{g}(x)}^2 \qquad\qquad \text{(Lemma 9-(a))}$$
$$\le 4(1 - \bar{r})^{-3}\|F^{(1)}(z') - F^{(1)}(z)\|_{\mathfrak{g}(\bar{x})}\|F^{(1)}(z')\|_{\mathfrak{g}(\bar{x})}$$
$$+ 2(1 - \bar{r})^{-3}\|F^{(1)}(z') - F^{(1)}(z)\|_{\mathfrak{g}(\bar{x})}^2 \qquad\qquad \text{(Lemma 9-(c))}$$
$$\le 4(1 - \bar{r})^{-3}A_{\bar{x},1}\|z - z'\|_{\bar{z}}\|F^{(1)}(z')\|_{\mathfrak{g}(\bar{x})} + 2(1 - \bar{r})^{-3}A_{\bar{x},1}^2\|z - z'\|_{\bar{z}}^2 \qquad \text{(see (19))}$$
$$\le \{4A_{\bar{x},1}(1 - \bar{r})^{-5}(\bar{r} + \|\bar{p}\|_{\mathfrak{g}(\bar{x})^{-1}}) + 4(1 - \bar{r})^{-3}A_{\bar{x},1}^2\bar{r}\}\|z - z'\|_{\bar{z}},$$

where we used that $\|F^{(1)}(z')\|_{\mathfrak{g}(\bar{x})} \le (1 - \bar{r})^{-2}(\bar{r} + \|\bar{p}\|_{\mathfrak{g}(\bar{x})^{-1}})$ in the last inequality. Combining Lemma 9-(c) and Lemma 11-(c), the second upper bound in (21) is bounded by

$$(1 - \bar{r})^{-1}\|D\mathfrak{g}(x)[F^{(1)}(z'), F^{(1)}(z')] - D\mathfrak{g}(x')[F^{(1)}(z'), F^{(1)}(z')]\|_{\mathfrak{g}(x)^{-1}}$$
$$\le \alpha(\alpha + 1)(1 - \bar{r})^{-1}/3\left\{(1 - \|x' - x\|_{\mathfrak{g}(x)})^{-3} - 1\right\}\|F^{(1)}(z')\|_{\mathfrak{g}(x)}^2$$
$$\le \alpha(\alpha + 1)(1 - \bar{r})^{-3}/3\left\{(1 - (1 - \bar{r})^{-1}\|x' - x\|_{\mathfrak{g}(\bar{x})})^{-3} - 1\right\}\|F^{(1)}(z')\|_{\mathfrak{g}(\bar{x})}^2$$
$$\le \alpha(\alpha + 1)(1 - \bar{r})^{-7}(\bar{r} + \|\bar{p}\|_{\mathfrak{g}(\bar{x})^{-1}})^2/3\left\{(1 - (1 - \bar{r})^{-1}\|x' - x\|_{\mathfrak{g}(\bar{x})})^{-3} - 1\right\}.$$

Using Inequality (a) with $u = (1 - \bar{r})^{-1}\|x' - x\|_{\mathfrak{g}(\bar{x})}$, where $u \le 1/5$ by (17), we obtain

$$(1 - (1 - \bar{r})^{-1}\|x' - x\|_{\mathfrak{g}(\bar{x})})^{-3} - 1 \le 5(1 - \bar{r})^{-1}\|x' - x\|_{\mathfrak{g}(\bar{x})}. \quad (22)$$

Then, the second upper bound in (21) is bounded by

$$(5/3)\alpha(\alpha + 1)(1 - \bar{r})^{-8}(\bar{r} + \|\bar{p}\|_{\mathfrak{g}(\bar{x})^{-1}})^2\|x' - x\|_{\mathfrak{g}(\bar{x})}.$$

We now define $h : y \in \mathsf{M} \to \mathfrak{g}(y)^{-1} : D\mathfrak{g}(y)$. Since $\phi \in \mathrm{C}^4(\mathsf{M})$, $h \in \mathrm{C}^1(\mathsf{M})$. Moreover, $[x, x'] \in \mathsf{M}$ by convexity of $\mathsf{M}$ and we can define $A_\phi = \sup_{y \in [x,x']} \|Dh(y)\|$, where $\|Dh(y)\| = \sup_{\|u\|_{\mathfrak{g}(\bar{x})}=1} \|Dh(y)[u]\|_{\mathfrak{g}(\bar{x})^{-1}}$. Using the mean value theorem on $h$, we have

$$\|h(x) - h(x')\|_{\mathfrak{g}(\bar{x})^{-1}} \leq A_\phi \|x' - x\|_{\mathfrak{g}(\bar{x})} . \tag{23}$$

By combining (14), (20), (21) and (23), we have

$$\begin{aligned}
\|F^{(2)}&(z) - F^{(2)}(z')\|_{\mathfrak{g}(\bar{x})^{-1}} \\
&\leq C_V \|x - x'\|_{\mathfrak{g}(\bar{x})} \\
&\quad + (1/2)\{4 A_{\bar{x},1}(1 - \bar{r})^{-5}(\bar{r} + \|\bar{p}\|_{\mathfrak{g}(\bar{x})^{-1}}) + 4 A_{\bar{x},1}^2(1 - \bar{r})^{-3}\bar{r}\}\|z - z'\|_{\bar{z}} \\
&\quad + (1/2)\cdot(5/3)\alpha(\alpha + 1)(1 - \bar{r})^{-8}(\bar{r} + \|\bar{p}\|_{\mathfrak{g}(\bar{x})^{-1}})^2\|x' - x\|_{\mathfrak{g}(\bar{x})} \\
&\quad + (1/2)A_\phi\|x' - x\|_{\mathfrak{g}(\bar{x})} .
\end{aligned}$$

We define $A_{\bar{x},2} = C_V + 2 A_{\bar{x},1}(1 - \bar{r})^{-5}(\bar{r} + \|\bar{p}\|_{\mathfrak{g}(\bar{x})^{-1}}) + 2 A_{\bar{x},1}^2(1 - \bar{r})^{-3}\bar{r} + (5/6)\alpha(\alpha + 1)(1 - \bar{r})^{-8}(\bar{r} + \|\bar{p}\|_{\mathfrak{g}(\bar{x})^{-1}})^2 + A_\phi/2$ such that

$$\|F^{(2)}(z) - F^{(2)}(z')\|_{\mathfrak{g}(\bar{x})} \leq A_{\bar{x},2}\|z - z'\|_{\bar{z}} .$$

Finally, we have

$$\begin{aligned}
\|F(z) - F(z')\|_{\bar{z}} &= \|F^{(1)}(z) - F^{(1)}(z')\|_{\mathfrak{g}(\bar{x})} + \|F^{(2)}(z) - F^{(2)}(z')\|_{\mathfrak{g}(\bar{x})^{-1}} \\
&\leq (A_{\bar{x},1} + A_{\bar{x},2})\|z - z'\|_{\bar{z}} ,
\end{aligned}$$

*i.e.*, $F$ is $(A_{\bar{x},1} + A_{\bar{x},2})$-Lipschitz-continuous on $\mathsf{B}_\mathfrak{g}$ with respect to $\|\cdot\|_{\bar{z}}$.

**We now show that $F$ is Lipschitz-continuous on $\mathsf{B}$ with respect to $\|\cdot\|_\mathsf{E}$.** Consider $(z, z') \in \mathsf{B} \times \mathsf{B}$ denoted by $z = (x, p)$ and $z' = (x', p')$. In particular, $(z, z') \in \mathsf{B}_\mathfrak{g} \times \mathsf{B}_\mathfrak{g}$. Using the previous result and Lemma 12, we have

$$\|F(z)-F(z')\|_\mathsf{E} \leq C_{\bar{x}}\|F(z)-F(z')\|_{\bar{z}} \leq C_{\bar{x}}(A_{\bar{x},1}+A_{\bar{x},2})\|z-z'\|_{\bar{z}} \leq C_{\bar{x}}^2(A_{\bar{x},1}+A_{\bar{x},2})\|z-z'\|_\mathsf{E} ,$$

which proves that $F$ is $C_{\bar{x}}^2(A_{\bar{x},1} + A_{\bar{x},2})$-Lipschitz-continuous on $\mathsf{B}$ with respect to $\|\cdot\|_\mathsf{E}$.

**Conclusion.** Combining Items (a) and (b), we obtain the results (i), (ii) and (iii) of Proposition 13 upon using Cauchy-Lipschitz's theorem. Moreover, for any $t \in J_{z_0}$, $\partial_t H(z(t)) = \partial_x H(z(t))^\top \dot{x}(t) + \partial_p H(z(t))^\top \dot{p}(t) = -\dot{p}(t)^\top \dot{x}(t) + \dot{p}(t)^\top \dot{x}(t) = 0$, which proves the result (iv) of Proposition 13. $\quad\square$

*Proof of Proposition 14.* Assume **A**1, **A**2. Let $z_0 \in \mathrm{T}^\star\mathsf{M}$, $(J_{z_0}, z)$ as in Proposition 13. Assume that $T_{z_0} = \sup J_{z_0} < +\infty$. Then, by Cauchy-Lipschitz's theorem, $z$ leaves any compact set of $\mathsf{E}$ at the neighbourhood of $T_{z_0}$. By construction of $\|\cdot\|_\mathsf{E}$, this property is induced on both variables $x \in \mathsf{M}$ and $p \in \mathbb{R}^d$, each with respect to $\|\cdot\|_2$. We directly obtain the result of Proposition 14 by continuity of $(x, p)$. $\quad\square$

### E.2 Proof of reversibility in Algorithm 2

In (RM)HMC, one often sets the time horizon of the Hamiltonian dynamics with a hyperparameter $h$. However, under **A**1 and **A**2, we are not able to prove that the Hamiltonian dynamics (3) is defined at time $h > 0$, given *any* starting point $z_0 \in \mathrm{T}^\star\mathsf{M}$. Actually, this dynamics explodes if the time horizon is finite (see Proposition 14). This theoretical limitation requires to implement some sort of "involution checking step" (see Line 4 in Algorithm 2), which is theoretically explained below.

In the rest of this section, for any $z_0 \in \mathrm{T}^\star\mathsf{M}$, we will denote by $z(\cdot, z_0) \in \mathrm{C}^1(J_{z_0}, \mathrm{T}^\star\mathsf{M})$ the maximal solution to (3) with starting point $z_0$, uniquely defined in Proposition 13. For any $h > 0$, we recall below the definition of the sets $\mathsf{E}_h$ and $\bar{\mathsf{E}}_h$ the map $\mathrm{T}_h$ introduced in Appendix D:

(a) $\mathsf{E}_h = \{z_0 \in \mathrm{T}^\star\mathsf{M} : h < \sup J_{z_0}\}$,

(b) $\bar{\mathsf{E}}_h = \{z_0 \in \mathsf{E}_h : h < \sup J_{(s \circ \mathrm{T}_h)(z_0)}\}$,

(c) the map $\mathrm{T}_h : \mathsf{E}_h \to \mathrm{T}^\star\mathsf{M}$ is such that $\mathrm{T}_h(z_0) = z(h, z_0)$ for any $z_0 \in \mathsf{E}_h$.

In particular, we have

$$\bar{\mathsf{E}}_h = \mathsf{E}_h \cap (s \circ \mathrm{T}_h)^{-1}(\mathsf{E}_h) . \tag{24}$$

We first prove that the restriction of $s \circ \mathrm{T}_h$ to $\bar{\mathsf{E}}_h$ is an involutive integrator, which is crucial to derive the "reversibility" of Algorithm 2 in Theorem 15.

**Lemma 16.** *Assume* **A**1, **A**2. *Then, for any* $h > 0$, $s \circ \mathrm{T}_h$ *is a* $\mathrm{C}^1$-*diffeomorphism on* $\bar{\mathsf{E}}_h$ *such that* $|\det \mathrm{Jac}[s \circ \mathrm{T}_h]| \equiv 1$.

*Proof.* Assume **A**1, **A**2. Let $h > 0$. It is clear that $\mathrm{T}_h$ (and thus $s \circ \mathrm{T}_h$) is well defined and continuously differentiable on $\mathsf{E}_h$ (in particular on $\bar{\mathsf{E}}_h$), by elaborating on the proof of Proposition 13 combined with Cauchy-Lipschitz's theorem. Let $z_0 \in \bar{\mathsf{E}}_h$. By definition of $\bar{\mathsf{E}}_h$, $\mathrm{T}_h(z_0)$, resp. $J_{z_0}$, and $(\mathrm{T}_h \circ s \circ \mathrm{T}_h)(z_0)$, resp. $J_{(s \circ \mathrm{T}_h)(z_0)}$, are well defined. We now aim to prove that $(\mathrm{T}_h \circ s \circ \mathrm{T}_h)(z_0) = s(z_0)$.

We define $z' : t \in [0, h] \mapsto s(z(h - t, z_0))$, which existence is straightforward since $(h - t) \in J_{z_0}$ for any $t \in [0, h]$. In particular, $z'(0) = (s \circ \mathrm{T}_h)(z_0)$ and $z'$ is a solution of the ODE (3) on the interval $[0, h]$. Yet, $[0, h] \subset J_{(s \circ \mathrm{T}_h)(z_0)}$, and $z'(0) = z(0, (s \circ \mathrm{T}_h)(z_0))$. Then, by unicity of $z(\cdot, (s \circ \mathrm{T}_h)(z_0))$ (see Proposition 13), $z(\cdot, (s \circ \mathrm{T}_h)(z_0))$ and $z'$ coincide on $[0, h]$. In particular, we have at time $h$

$$s(z_0) = z'(h) = z(h, (s \circ \mathrm{T}_h)(z_0)) = (\mathrm{T}_h \circ s \circ \mathrm{T}_h)(z_0) .$$

Then for any $z_0 \in \bar{\mathsf{E}}_h$, $(s \circ \mathrm{T}_h \circ s \circ \mathrm{T}_h)(z_0) = z_0$, *i.e.*, $s \circ \mathrm{T}_h$ is an involution on $\bar{\mathsf{E}}_h$. Since $s \circ \mathrm{T}_h \in \mathrm{C}^1(\bar{\mathsf{E}}_h, \mathrm{T}^\star \mathsf{M})$, $s \circ \mathrm{T}_h$ is a $\mathrm{C}^1$-diffeomorphism on $\bar{\mathsf{E}}_h$ such that $|\det \mathrm{Jac}[s \circ \mathrm{T}_h]| \equiv 1$. $\square$

We recall that we denote by $\mathrm{Q}_c : \mathrm{T}^\star \mathsf{M} \times \mathcal{B}(\mathrm{T}^\star \mathsf{M}) \to [0, 1]$, the transition kernel of the (homogeneous) Markov chain $(X_n, P_n)_{n \in [N]}$ obtained with Algorithm 2. We also denote by:

(a) $\mathrm{Q}_0 : \mathrm{T}^\star \mathsf{M} \times \mathcal{B}(\mathrm{T}^\star \mathsf{M}) \to [0, 1]$, the transition kernel referring to Step 1 (also Step 3) in Algorithm 2.

(b) $\mathrm{Q}_{c,1} : \mathrm{T}^\star \mathsf{M} \times \mathcal{B}(\mathrm{T}^\star \mathsf{M}) \to [0, 1]$, the transition kernel referring to Step 2 in Algorithm 2.

We provide below details on Markov kernels $\mathrm{Q}_0$, $\mathrm{Q}_{c,1}$ and $\mathrm{Q}_c$.

**Kernel** $\mathrm{Q}_0$. This kernel corresponds to the Gaussian momentum update with refreshing rate $\beta$. For any $(z, z') \in \mathrm{T}^\star \mathsf{M} \times \mathrm{T}^\star \mathsf{M}$, we have

$$\begin{aligned}
\mathrm{Q}_0(z, \mathrm{d}z') &= \mathrm{N}_x(p'; \sqrt{1 - \beta}p, \beta \mathrm{Id})\mathrm{d}p'\delta_x(\mathrm{d}x') \\
&= (2\pi\beta)^{-d/2} \det(\mathfrak{g}(x))^{-1/2}\exp[-(2\beta)^{-1}\|p' - \sqrt{1 - \beta}p\|^2_{\mathfrak{g}(x)^{-1}}]\mathrm{d}p'\delta_x(\mathrm{d}x') .
\end{aligned} \tag{25}$$

**Kernel** $\mathrm{Q}_{c,1}$. This kernel is deterministic and corresponds to the *exact* integration of the Hamiltonian up until time $h$. For any $(z, z') \in \mathrm{T}^\star \mathsf{M} \times \mathrm{T}^\star \mathsf{M}$, we have

$$\mathrm{Q}_{c,1}(z, \mathrm{d}z') = \mathbb{1}_{\bar{\mathsf{E}}_h}(z)\delta_{\mathrm{T}_h(z)}(\mathrm{d}z') + \mathbb{1}_{\bar{\mathsf{E}}_h^c}(z)\delta_{s(z)}(\mathrm{d}z') . \tag{26}$$

**Kernel** $\mathrm{Q}_c$. This kernel corresponds to one step of Algorithm 2 (*i.e.*, comprising Steps 1 to 3). For any $(z, z') \in \mathrm{T}^\star \mathsf{M} \times \mathrm{T}^\star \mathsf{M}$, we have

$$\mathrm{Q}_c(z, \mathrm{d}z') = \int_{\mathrm{T}^\star \mathsf{M} \times \mathrm{T}^\star \mathsf{M}} \mathrm{Q}_0(z, \mathrm{d}z_1)\mathrm{Q}_{c,1}(z_1, \mathrm{d}z_2)\mathrm{Q}_0(z_2, \mathrm{d}z') . \tag{27}$$

We recall that $\bar{\pi}$, as defined in (2), admits a density with respect to the product Lebesgue measure given for any $z = (x, p) \in \mathrm{T}^\star \mathsf{M}$ by

$$(\mathrm{d}\bar{\pi}/(\mathrm{d}x\mathrm{d}p))(x, p) = (1/Z) \exp[-(1/2)\|p\|^2_{\mathfrak{g}(x)^{-1}}] \det(\mathfrak{g}(x))^{-1/2} \exp[-V(x)] .$$

**Lemma 17.** *Assume* **A**1. *Then, the Markov kernel* $\mathrm{Q}_0$, *defined in* (25), *is reversible (up to momentum reversal) with respect to* $\bar{\pi}$.

*Proof.* Assume **A**1. For any $(z, z') \in T^\star M \times T^\star M$, we have

$$Q_0(z, \mathrm{d}z')\bar{\pi}(\mathrm{d}z) = (1/Z)(2\pi)^{-d/2}\beta^{-d/2}\det(\mathfrak{g}(x))^{-1}\exp[-V(x)]\mathrm{d}x\mathrm{d}p\mathrm{d}p'\delta_x(\mathrm{d}x')$$
$$\exp[-(2\beta)^{-1}\|p' - \sqrt{1-\beta}p\|^2_{\mathfrak{g}(x)^{-1}}]\exp[-(1/2)\|p\|^2_{\mathfrak{g}(x)^{-1}}]$$
$$= (1/Z)(2\pi)^{-d/2}\beta^{-d/2}\det(\mathfrak{g}(x))^{-1}\exp[-V(x)]\mathrm{d}x\mathrm{d}p\mathrm{d}p'\delta_x(\mathrm{d}x')$$
$$\exp[-(2\beta)^{-1}\{\|p\|^2_{g(x)^{-1}} - 2\sqrt{1-\beta}\langle p', p\rangle_{g(x)^{-1}} + \|p'\|^2_{g(x)^{-1}}\}]$$
$$= (1/Z)(2\pi)^{-d/2}\beta^{-d/2}\det(\mathfrak{g}(x'))^{-1}\exp[-V(x')]\mathrm{d}x'\mathrm{d}p'\mathrm{d}p\delta_x(\mathrm{d}x)$$
$$\exp[-(2\beta)^{-1}\{\|p'\|^2_{g(x')^{-1}} - 2\sqrt{1-\beta}\langle p, p'\rangle_{g(x')^{-1}} + \|p\|^2_{g(x')^{-1}}\}]$$
$$= Q_0(z', \mathrm{d}z)\bar{\pi}(\mathrm{d}z') ,$$

which proves the reversibility of $Q_0$ with respect to $\bar{\pi}$. Since $s_\# Q_0 = Q_0$, we conclude the proof with Lemma 6. $\square$

**Lemma 18.** *Assume* **A**1, **A**2. *Then, for any $h > 0$, the Markov kernel $Q_{c,1}$ with step-size $h$, defined in* (26)*, is reversible up to momentum reversal with respect to $\bar{\pi}$.*

*Proof.* Assume **A**1, **A**2. We recall that the set $\bar{\mathsf{E}}_h$ is defined in (24). Let $f \in C(T^\star M \times T^\star M, \mathbb{R})$ with compact support. According to Definition 4, we aim to show that

$$\int_{T^\star M \times T^\star M} f(z, z')Q_{c,1}(z, \mathrm{d}z')\bar{\pi}(\mathrm{d}z) = \int_{T^\star M \times T^\star M} f(s(z'), s(z))Q_{c,1}(z, \mathrm{d}z')\bar{\pi}(\mathrm{d}z) . \quad (28)$$

We denote by $I$ the left integral of (28). We have $I = I_1 + I_2$ where

$$I_1 = \int_{\bar{\mathsf{E}}_h} \bar{\pi}(z)f(z, \mathrm{T}_h(z))\mathrm{d}z , \qquad I_2 = \int_{\bar{\mathsf{E}}_h^c} \bar{\pi}(z)f(z, s(z))\mathrm{d}z .$$

We denote by $J$ the right integral of (28). By symmetry, we have $J = J_1 + J_2$ where

$$J_1 = \int_{\bar{\mathsf{E}}_h} \bar{\pi}(z)f((s \circ \mathrm{T}_h)(z), s(z))\mathrm{d}z , \qquad J_2 = \int_{\bar{\mathsf{E}}_h^c} \bar{\pi}(z)f(z, s(z))\mathrm{d}z .$$

We directly have $I_2 = J_2$. Let us now prove that $I_1 = J_1$. By change of variable $z \mapsto (s \circ \mathrm{T}_h)(z)$ in $I_1$, we have

$$I_1 = \int_{\bar{\mathsf{E}}_h} \bar{\pi}(z)f(z, \mathrm{T}_h(z))\mathrm{d}z$$
$$= \int_{(s \circ \mathrm{T}_h)(\bar{\mathsf{E}}_h)} \bar{\pi}((s \circ \mathrm{T}_h)(z))f((s \circ \mathrm{T}_h)(z), s(z))\mathrm{d}z \quad \text{(Lemma 16)}$$
$$= \int_{\bar{\mathsf{E}}_h} \bar{\pi}(z)f((s \circ \mathrm{T}_h)(z), s(z))\mathrm{d}z \quad \text{(Proposition 13-(iv) \& } (s \circ \mathrm{T}_h)(\bar{\mathsf{E}}_h) = \bar{\mathsf{E}}_h)$$
$$= J_1 .$$

Finally, we obtain $I = J$ and thus prove (28) for any continuous function with compact support, which concludes the proof.

$\square$

We are now ready to prove Theorem 15, which states that $Q_c$ is reversible up to momentum reversal with respect to $\bar{\pi}$.

*Proof of Theorem 15.* Assume **A**1, **A**2. Let $f : T^\star M \times T^\star M \to \mathbb{R}$ be a continuous function with compact support. We have

$$\int_{T^\star M \times T^\star M} f(z, z')Q_c(z, \mathrm{d}z')\bar{\pi}(\mathrm{d}z)$$
$$= \int_{(T^\star M)^4} f(z, z')Q_0(z, \mathrm{d}z_1)Q_{c,1}(z_1, \mathrm{d}z_2)Q_0(z_2, \mathrm{d}z')\bar{\pi}(\mathrm{d}z) \quad \text{(see (27))}$$
$$= \int_{(T^\star M)^4} f(s(z_1), z')Q_0(z, \mathrm{d}z_1)Q_{c,1}(s(z), \mathrm{d}z_2)Q_0(z_2, \mathrm{d}z')\bar{\pi}(\mathrm{d}z) \quad \text{(Lemma 17)}$$
$$= \int_{(T^\star M)^4} f(s(z_1), z')Q_0(s(z), \mathrm{d}z_1)Q_{c,1}(z, \mathrm{d}z_2)Q_0(z_2, \mathrm{d}z')\bar{\pi}(\mathrm{d}z) \quad \text{(momentum reversal on } z)$$
$$= \int_{(T^\star M)^4} f(s(z_1), z')Q_0(z_2, \mathrm{d}z_1)Q_{c,1}(z, \mathrm{d}z_2)Q_0(s(z), \mathrm{d}z')\bar{\pi}(\mathrm{d}z) \quad \text{(Lemma 18)}$$
$$= \int_{(T^\star M)^4} f(s(z_1), z')Q_0(z_2, \mathrm{d}z_1)Q_{c,1}(s(z), \mathrm{d}z_2)Q_0(z, \mathrm{d}z')\bar{\pi}(\mathrm{d}z) \quad \text{(momentum reversal on } z)$$
$$= \int_{(T^\star M)^4} f(s(z_1), s(z))Q_0(z_2, \mathrm{d}z_1)Q_{c,1}(z', \mathrm{d}z_2)Q_0(z, \mathrm{d}z')\bar{\pi}(\mathrm{d}z) \quad \text{(Lemma 17)}$$
$$= \int_{T^\star M \times T^\star M} f(s(z'), s(z))Q_c(z, \mathrm{d}z')\bar{\pi}(\mathrm{d}z) .$$

Moreover, $s_\# \bar\pi = \bar\pi$. Hence, by combining Definition 4 and Lemma 5, we obtain the result of Theorem 15. □

# F  Numerical integration in HMC

In this section, we first recall the definition of symplectic maps, and then discuss the choice of integrators in RMHMC. Finally, we focus on the implicit integrator defined in (6) and provide some notations, which will be notably used in appendix G.

**Reminders on symplecticity.**    We define $J_d = \begin{pmatrix} 0 & \mathrm{I}_d \\ -\mathrm{I}_d & 0 \end{pmatrix} \in \mathbb{R}^{2d \times 2d}$.

**Definition 19** (Hairer et al. (2006))**.** *A linear mapping $A : \mathbb{R}^{2d} \to \mathbb{R}^{2d}$ is called symplectic if $A^\top J_d A = J_d$.*

**Definition 20** (Hairer et al. (2006))**.** *A differentiable map $F : \mathsf{U} \to \mathbb{R}^{2d}$, where $\mathsf{U} \subset \mathbb{R}^{2d}$ is an open set, is called symplectic if the Jacobian matrix $\mathrm{Jac}[F]$ is symplectic, i.e., if $\mathrm{Jac}[F](z)^\top J_d \, \mathrm{Jac}[F](z) = J_d$ for any $z \in \mathsf{U}$. In particular, if $F$ is symplectic, then $|\det \mathrm{Jac}[F]| \equiv 1$.*

**Choice of the integrators in RMHMC.**    Introducing a Riemannian metric $\mathfrak{g}$ in the Hamiltonian *a priori* makes it non separable. Thus, designing an integrator which is (i) symplectic, (ii) reversible and (iii) not too computationally heavy is challenging. The *generalized Leapfrog* integrator (GLI), or equivalently the Störmer-Verlet integrator, combined with fixed point iterations, as chosen by Girolami & Calderhead (2011), is considered as the standard scheme for RMHMC. Brofos & Lederman (2021b) analyze the impact of GLI on the ergodicity of RMHMC under a metric which is designed in the same manner as Cobb et al. (2019). In particular, they show that the convergence threshold used to compute the fixed-point process only matters to an extent, and identify a diminishing return to using smaller convergence thresholds. They compare the fixed-point approach with Newton's method, which gives less iterations to converge. On the other hand, the *implicit midpoint* integrator (IMI) enjoys the same theoretical properties as GLI, when these two integrators are combined with fixed-point iterations. As shown by Brofos & Lederman (2021a), IMI may also show numerical advantages in terms of reversibility and volume preservation for specific target distributions. However, this integrator is hard to use for reversibility proofs due to its non-separability. Besides this, Cobb et al. (2019) combine a 2-state augmented Hamiltonian, based on the work of Tao (2016), with Strang splitting (Strang, 1968) to propose a symplectic *explicit* integrator which thus has the advantage not to rely on fixed-point iterations. Yet, this integrator does not satisfy reversibility in theory.

**Details on the Störmer-Verlet integrator.**    The scheme defined in (6) can be written as $\mathrm{G}_h = \overline{\mathrm{G}}_h \circ \underline{\mathrm{G}}_h$ where:

(a) $\underline{\mathrm{G}}_h : \mathrm{T}^\star \mathsf{M} \to 2^{\mathrm{T}^\star \mathsf{M}}$ is the set-valued map *implicitly* defined by the first two aligns of (6), *i.e.*, for any $(z, z') \in \mathrm{T}^\star \mathsf{M} \times \mathrm{T}^\star \mathsf{M}$, $z' \in \underline{\mathrm{G}}_h(z)$ if and only if

$$p' = p - \tfrac{h}{2} \partial_x H_2(x, p'), \qquad x' = x + \tfrac{h}{2} [\partial_p H_2(x, p') + \partial_p H_2(x', p')] \,, \qquad (29)$$

We also define the map $\underline{\mathrm{g}}_h : \mathrm{T}^\star \mathsf{M} \times \mathrm{T}^\star \mathsf{M} \to \mathbb{R}^n \times \mathbb{R}^n$ by

$$\underline{\mathrm{g}}_h(z, z') = (x' - x - \tfrac{h}{2} [\partial_p H_2(x, p') + \partial_p H_2(x', p')], p' - p + \tfrac{h}{2} \partial_x H_2(x, p')) \,. \qquad (30)$$

such that $\underline{\mathrm{g}}_h(z, z') = 0$ if and only if $z' \in \underline{\mathrm{G}}_h(z)$.

(b) $\overline{\mathrm{G}}_h : \mathrm{T}^\star \mathsf{M} \to \mathrm{T}^\star \mathsf{M}$ is *explicitly* defined by the last align of (6), *i.e.*, for any $z \in \mathrm{T}^\star \mathsf{M}$,

$$\overline{\mathrm{G}}_h(z) = (x, p - \tfrac{h}{2} \partial_x H_2(x, p)) \,. \qquad (31)$$

# G  Proofs of Section 4.1

In this section, we state several results on the maps defined by the implicit integrators of the Hamiltonian (see Section 3.1), in order to prove Proposition 2.

**Lemma 21.** *Let $\mathsf{U} \subset \mathbb{R}^d$ a non-empty open set and $z \in \mathsf{U}$. Assume there exist a neighbourhood $\mathsf{U}_z \subset \mathsf{U}$ of $z$, $\mathsf{L} > 0$ and a $\mathsf{L}$-Lipschitz-continuous map $\psi : \mathsf{U}_z \to \mathbb{R}^d$ such that $\psi \in \mathrm{C}^1(\mathsf{U}_z, \mathbb{R}^d)$. Then, for any $h \in (0, 1/\mathsf{L})$ and any $a \in \mathbb{R}^d$, the map $\psi_h : \mathsf{U}_z \to \mathbb{R}^d$ defined for any $z' \in \mathsf{U}_z$ by $\psi_h(z') = z' + h\psi(z') + a$ is a $\mathrm{C}^1$-diffeomorphism on a neighbourhood $\mathsf{U}'_z \subset \mathsf{U}_z$ of $z$.*

*Proof.* Assume we are provided with $\mathsf{U}$, $z$, $\mathsf{U}_z$, $\mathsf{L}$ and $\psi$ as described in Lemma 21. Let $h \in (0, 1/\mathsf{L})$ and $a \in \mathbb{R}^d$. We define the map $\psi_h : \mathsf{U}_z \to \mathbb{R}^d$ by $\psi_h(z') = z' + h\psi(z') + a$, for any $z' \in \mathsf{U}_z$. Since $\psi \in \mathrm{C}^1(\mathsf{U}_z, \mathbb{R}^d)$, it is clear that $\psi_h \in \mathrm{C}^1(\mathsf{U}_z, \mathbb{R}^d)$. We have, for any $z' \in \mathsf{U}_z$, $D\psi_h(z') = \mathrm{Id} + hD\psi(z')$. In particular, for any $w \in \mathbb{R}^d$, it comes that

$$\|D\psi_h(z)w\| \geq \|w\| - h\mathsf{L}\|w\| = (1 - h\mathsf{L})\|w\| .$$

Since $1 - h\mathsf{L} > 0$, $\mathrm{Jac}[\psi_h](z)$ is invertible. We conclude the proof by using the local inverse function theorem. $\qquad\square$

We recall that the set-valued map $\mathrm{G}_h$, defined in (6), is the *generalised Leapfrog* integrator, or equivalently the *Störmer-Verlet* integrator, of the non-separable Hamiltonian $H_2$ defined in Section 3.1. We state below a result of local smoothness of this integrator.

**Lemma 22.** *Let $(\mathsf{M}, \mathfrak{g})$ be a a smooth manifold of $\mathbb{R}^d$ and $h > 0$. Then, for any $(x^{(0)}, x^{(1)}) \in \mathsf{M} \times \mathsf{M}$, the vector $p^{(0)} \in \mathrm{T}^\star_{x^{(0)}}\mathsf{M}$ such that $(x^{(1)}, p^{(1)}) \in \mathrm{G}_h(x^{(0)}, p^{(0)})$ for any $p^{(1)} \in \mathrm{T}^\star_{x^{(1)}}\mathsf{M}$ is uniquely determined by $p^{(0)} = \mathrm{G}_{h,x^{(0)}}(x^{(1)})$, where the map $\mathrm{G}_{h,x^{(0)}} : \mathsf{M} \to \mathrm{T}^\star_{x^{(0)}}\mathsf{M}$ is defined by*

$$\mathrm{G}_{h,x^{(0)}}(y) = p^{(1/2)}(y) - \frac{h}{4}D\mathfrak{g}(x^{(0)})[\mathfrak{g}(x^{(0)})^{-1}p^{(1/2)}(y), \mathfrak{g}(x^{(0)})^{-1}p^{(1/2)}(y)] , \tag{32}$$

*with $p^{(1/2)}(y) = \frac{2}{h}(\mathfrak{g}(x^{(0)})^{-1} + \mathfrak{g}(y)^{-1})^{-1}(y - x^{(0)})$. In particular, if $\mathfrak{g} \in \mathrm{C}^2(\mathsf{M}, \mathbb{R}^{d \times d})$, then, for any $x^{(0)} \in \mathsf{M}$, $\mathrm{G}_{h,x^{(0)}} \in \mathrm{C}^1(\mathsf{M}, \mathrm{T}^\star_{x^{(0)}}\mathsf{M})$.*

*For any $\alpha \geq 1$, we define $r^\star_\alpha > 0$ by*

$$r^\star_\alpha = \min(1/50000, 1/\{1000\alpha(\alpha + 1)\}) . \tag{33}$$

*We recall that we denote by $\mathsf{W}^0(x, r)$ the open Dikin ellipsoid (w.r.t. $\mathfrak{g}$) at $x \in \mathsf{M}$ of radius $r > 0$, given by $\mathsf{W}^0(x, r) = \{y \in \mathbb{R}^d : \|y - x\|_{\mathfrak{g}(x)} < r\}$. Assume **A**1, **A**2. Let $x^{(0)} \in \mathsf{M}$ and $r \in (0, r^\star_\alpha]$, where $\alpha$ is given by **A**2. Then, for any $x \in \mathsf{W}^0(x^{(0)}, r)$, $x \in \mathsf{M}$ and $\mathrm{Jac}[\mathrm{G}_{h,x^{(0)}}](x)$ is invertible for any $h > 0$.*

*Proof.* Let $(\mathsf{M}, \mathfrak{g})$ be a a smooth manifold of $\mathbb{R}^d$ and $h > 0$. We first prove the existence and uniqueness of the map defined in (32). Let $(x^{(0)}, x^{(1)}) \in \mathsf{M} \times \mathsf{M}$. We define for any $y \in \mathsf{M}$

$$p^{(1/2)}(y) = \frac{2}{h}(\mathfrak{g}(x^{(0)})^{-1} + \mathfrak{g}(y)^{-1})^{-1}(y - x^{(0)}) , \qquad \tilde{p}(y) = p^{(1/2)}(y) + \frac{h}{2}\partial_x H_2(x^{(0)}, p^{(1/2)}(y)) .$$

Note that these expressions are obtained by simply inverting the first two aligns of (6). Then, by definition of $\mathrm{G}_h$, the vector $p \in \mathrm{T}^\star_{x^{(0)}}\mathsf{M}$ such that $(x^{(1)}, p^{(1)}) \in \mathrm{G}_h(x^{(0)}, p)$ for any $p^{(1)} \in \mathrm{T}^\star_{x^{(1)}}\mathsf{M}$ is uniquely determined by $p = \tilde{p}(x^{(1)})$. We thus obtain (32), using the expression of $\partial_x H_2$ given in Section 3.1. Moreover, it is clear that $\mathrm{G}_{h,x^{(0)}}$ is continuously differentiable if $\mathfrak{g} \in \mathrm{C}^2(\mathsf{M}, \mathbb{R}^{d \times d})$.

We are now going to prove the second result of Lemma 22. Assume **A**1, **A**2. Let $x^{(0)} \in \mathsf{M}$ and $r \in (0, r^\star_\alpha]$, where $r^\star_\alpha$ is defined in (33) and $\alpha$ is given by **A**2. In particular $r^\star_\alpha \leq 1/11$. For the sake of clarity, we will now denote $\mathfrak{g}(x^{(0)})$ by $\mathfrak{g}_0$ for the rest of the proof.

We define the open Dikin ellipsoid $\mathsf{B}_0 = \mathsf{W}^0(x^{(0)}, r)$. Since $r < 1$, $\mathsf{B}_0 \subset \mathsf{M}$ by Lemma 9-(b). Let $h > 0$. We now define the following maps on $\mathsf{B}_0$

$$\mathrm{M}_0 = \left\{ \begin{array}{ll} \mathsf{B}_0 & \to S^{++}_d(\mathbb{R}) \\ y & \mapsto \mathfrak{g}_0^{-1} + \mathfrak{g}(y)^{-1} \end{array} \right. , \qquad \mathrm{K}_0 = \left\{ \begin{array}{ll} \mathsf{B}_0 & \to \mathbb{R}^d \\ y & \mapsto \mathfrak{g}_0^{-1}\mathrm{M}_0(y)^{-1}(y - x^{(0)}) \end{array} \right. ,$$

$$\phi_0 = \left\{ \begin{array}{ll} \mathsf{B}_0 & \to \mathbb{R}^d \\ y & \mapsto \mathrm{M}_0(y)D\mathfrak{g}_0[\mathrm{K}_0(y), \mathrm{K}_0(y)] \end{array} \right. , \qquad \tilde{\phi}_0 = \left\{ \begin{array}{ll} \mathsf{B}_0 & \to \mathbb{R}^d \\ y & \mapsto \frac{h}{2}\mathrm{M}_0(y)\mathrm{G}_{h,x^{(0)}}(y) \end{array} \right. .$$

Note that we have, for any $y \in \mathsf{B}_0$, $\tilde{\phi}(y) = (y - x^{(0)}) - \frac{1}{2}\phi_0(y)$. Since $\mathfrak{g}$ is twice continuously differentiable on $\mathsf{B}_0$, the maps previously defined are continuously differentiable. We now compute the derivatives of these maps. Let $y \in \mathsf{B}_0$ and $w \in \mathbb{R}^d$. We have

$$DM_0(y)[w] = -\mathfrak{g}(y)^{-1}D\mathfrak{g}(y)[w]\mathfrak{g}(y)^{-1} \;,$$

$$\begin{aligned} DK_0(y)[w] &= -\mathfrak{g}_0^{-1}M_0(y)^{-1}DM_0(y)[w]M_0(y)^{-1}(y - x^{(0)}) + \mathfrak{g}_0^{-1}M_0(y)^{-1}w \\ &= -\mathfrak{g}_0^{-1}\{\mathfrak{g}(y)\mathfrak{g}_0^{-1} + I_d\}^{-1}\mathfrak{g}(y)DM_0(y)[w]\mathfrak{g}(y)\{\mathfrak{g}_0^{-1}\mathfrak{g}(y) + I_d\}^{-1}(y - x^{(0)}) \\ &\quad + \mathfrak{g}_0^{-1}M_0(y)^{-1}w \\ &= \mathfrak{g}_0^{-1}\{\mathfrak{g}(y)\mathfrak{g}_0^{-1} + I_d\}^{-1}D\mathfrak{g}(y)[w]\{\mathfrak{g}_0^{-1}\mathfrak{g}(y) + I_d\}^{-1}(y - x^{(0)}) + \mathfrak{g}_0^{-1}M_0(y)^{-1}w \;, \end{aligned}$$

$$\begin{aligned} D\phi_0(y)[w] &= DM_0(y)[w]D\mathfrak{g}_0[K_0(y), K_0(y)] + M_0(y)D\mathfrak{g}_0[DK_0(y)[w], K_0(y)] \\ &\quad + M_0(y)D\mathfrak{g}_0[K_0(y), DK_0(y)[w]] \;, \end{aligned}$$

$$D\tilde{\phi}_0(y)[w] = I_d - \frac{1}{2}D\phi_0(y)[w] \;.$$

We have in particular $\tilde{\phi}_0(x^{(0)}) = 0$ and $D\phi_0(x^{(0)}) = 0$. Let us first study the smoothness of $D\phi_0$ on $\mathsf{B}_0$.

**Smoothness of $D\phi_0$.** We prove here that $D\phi_0$ is $(r_\alpha^\star)^{-1}$-Lipschitz-continuous on $\mathsf{B}_0$ with respect to $\|\cdot\|_{\mathfrak{g}_0}$. Let $(y, y') \in \mathsf{B}_0$ and $w \in \mathbb{R}^d$. We have

$$\begin{aligned} \|D\phi_0(y)[w] &- D\phi_0(y')[w]\|_{\mathfrak{g}_0} \qquad\qquad\qquad\qquad\qquad\qquad\qquad\qquad\qquad (34) \\ &\leq \|DM_0(y)[w]D\mathfrak{g}_0[K_0(y), K_0(y)] - DM_0(y')[w]D\mathfrak{g}_0[K_0(y'), K_0(y')]\|_{\mathfrak{g}_0} \\ &\quad + \|M_0(y)D\mathfrak{g}_0[DK_0(y)[w], K_0(y)] - M_0(y')D\mathfrak{g}_0[DK_0(y')[w], K_0(y')]\|_{\mathfrak{g}_0} \\ &\quad + \|M_0(y)D\mathfrak{g}_0[K_0(y), DK_0(y)[w]] - M_0(y')D\mathfrak{g}_0[K_0(y'), DK_0(y')[w]]\|_{\mathfrak{g}_0} \;. \end{aligned}$$

Remark that the two last terms in (34) are very similar and can be bounded in the same way. The rest of the proof on the Lipschitz-continuity of $D\phi_0$ is separated in two steps, each of them consisting of bounding from above a term of (34).

*Step 1.* Let us first bound the first term in (34). For any $x \in \mathsf{B}_0$, we denote $\overline{DM_0(x)[w]D\mathfrak{g}_0[K_0(x), K_0(x)]}$ by $a_1(x)$. We have

$$\begin{aligned} \mathfrak{g}_0^{1/2}(a_1(y) &- a_1(y')) \qquad\qquad\qquad\qquad\qquad\qquad\qquad\qquad\qquad\qquad (35) \\ &= \{\mathfrak{g}_0^{1/2}(DM_0(y)[w] - DM_0(y')[w])\mathfrak{g}_0^{1/2}\}\{\mathfrak{g}_0^{-1/2}D\mathfrak{g}_0[K_0(y), K_0(y)]\} \\ &\quad + \{\mathfrak{g}_0^{1/2}DM_0(y')[w]\mathfrak{g}_0^{1/2}\}\{\mathfrak{g}_0^{-1/2}(D\mathfrak{g}_0[K_0(y), K_0(y)] - D\mathfrak{g}_0[K_0(y'), K_0(y')])\} \;. \end{aligned}$$

We now aim to bound each one of the four terms that appear in (35).

*Step 1.1.* First, we have

$$\begin{aligned} \mathfrak{g}_0^{1/2}(DM_0(y)[w] - DM_0(y')[w])\mathfrak{g}_0^{1/2} &= \mathfrak{g}_0^{1/2}(\mathfrak{g}(y')^{-1}D\mathfrak{g}(y')[w]\mathfrak{g}(y')^{-1} - \mathfrak{g}(y)^{-1}D\mathfrak{g}(y)[w]\mathfrak{g}(y)^{-1})\mathfrak{g}_0^{1/2} \\ &= \mathfrak{g}_0^{1/2}\{\mathfrak{g}(y')^{-1} - \mathfrak{g}(y)^{-1}\}D\mathfrak{g}(y')[w]\mathfrak{g}(y')^{-1}\mathfrak{g}_0^{1/2} \\ &\quad + \mathfrak{g}_0^{1/2}\mathfrak{g}(y)^{-1}\{D\mathfrak{g}(y')[w] - D\mathfrak{g}(y)[w]\}\mathfrak{g}(y')^{-1}\mathfrak{g}_0^{1/2} \\ &\quad + \mathfrak{g}_0^{1/2}\mathfrak{g}(y)^{-1}D\mathfrak{g}(y)[w]\{\mathfrak{g}(y')^{-1} - \mathfrak{g}(y)^{-1}\}\mathfrak{g}_0^{1/2} \;. \end{aligned}$$

We recall that $r \leq r_\alpha^\star \leq 1/11$, and thus, we have for any $x \in \{y, y'\}$ the following inequalities

$$\|\mathfrak{g}_0^{1/2}\{\mathfrak{g}(y')^{-1} - \mathfrak{g}(y)^{-1}\}\mathfrak{g}_0^{1/2}\|_2 \leq 3(1 - r)^{-3}\|y - y'\|_{\mathfrak{g}_0} \;, \qquad\qquad \text{(on the model of (18))}$$

$$\|\mathfrak{g}_0^{1/2}\mathfrak{g}(x)^{-1}\mathfrak{g}_0^{1/2}\|_2 \leq (1 - r)^{-2} \;, \qquad\qquad\qquad\qquad\qquad \text{(Lemma 9-(b))}$$

$$\|\mathfrak{g}_0^{-1/2}D\mathfrak{g}(x)[w]\mathfrak{g}_0^{-1/2}\|_2 \leq (1 - r)^{-2}\|\mathfrak{g}(x)^{-1/2}D\mathfrak{g}(x)[w]\mathfrak{g}(x)^{-1/2}\|_2 \;.$$

In particular, we have for any $x \in \{y, y'\}$

$$\|\mathfrak{g}(x)^{-1/2} D\mathfrak{g}(x)[w]\mathfrak{g}(x)^{-1/2}\|_2 = \sup_{\{w' \in \mathbb{R}^d \,:\, \|w'\|_2 = 1\}} \|D\mathfrak{g}(x)[w, \mathfrak{g}(x)^{-1/2}w']\|_{\mathfrak{g}(x)^{-1}} \tag{36}$$

$$\leq 2\|w\|_{\mathfrak{g}(x)} \sup_{\{w' \in \mathbb{R}^d \,:\, \|w'\|_2 = 1\}} \|\mathfrak{g}(x)^{-1/2}w'\|_{\mathfrak{g}(x)} \qquad \text{(Lemma 9-(a))}$$

$$\leq 2(1-r)^{-1}\|w\|_{\mathfrak{g}_0} \,, \qquad\qquad\qquad \text{(Lemma 9-(c))}$$

and using again that $r \leq r_\alpha^\star \leq 1/11$, we have

$$\|\mathfrak{g}(y)^{-1/2}\{D\mathfrak{g}(y')[w] - D\mathfrak{g}(y)[w]\}\mathfrak{g}(y')^{-1/2}\|_2 \tag{37}$$

$$= \sup_{\{w' \in \mathbb{R}^d \,:\, \|w'\|_2 = 1\}} \|D\mathfrak{g}(y')[w, \mathfrak{g}(y')^{-1/2}w'] - D\mathfrak{g}(y)[w, \mathfrak{g}(y')^{-1/2}w']\}\|_{\mathfrak{g}(y)^{-1}}$$

$$\leq \alpha(\alpha+1)/3\|w\|_{\mathfrak{g}(y)}((1 - \|y - y'\|_{\mathfrak{g}(y)})^{-3} - 1) \qquad\qquad \text{(Lemma 11-(c))}$$

$$\leq 5\alpha(\alpha+1)/3(1-r)^{-2}\|w\|_{\mathfrak{g}_0}\|y - y'\|_{\mathfrak{g}_0} \,. \qquad\qquad \text{(on the model of (22))}$$

Thus, by using (36) and (37), we have

$$\|\mathfrak{g}_0^{1/2}(DM_0(y)[w] - DM_0(y')[w])\mathfrak{g}_0^{1/2}\|_2 \leq (12(1-r)^{-8} + 5\alpha(\alpha+1)/3(1-r)^{-6})\|y - y'\|_{\mathfrak{g}_0}\|w\|_{\mathfrak{g}_0} \,.$$

*Step 1.2.* Secondly, we have

$$\|\mathfrak{g}_0^{-1/2} D\mathfrak{g}_0[K_0(y), K_0(y)]\|_2 \leq 2\|K_0(y)\|_{\mathfrak{g}_0}^2 \qquad\qquad \text{(Lemma 9-(a))}$$

$$\leq 2\|\mathfrak{g}_0^{-1/2}M_0(y)^{-1}(y - x^{(0)})\|_2^2 \qquad \text{(definition of } K_0\text{)}$$

$$\leq 2\|\mathfrak{g}_0^{-1/2}M_0(y)^{-1}\mathfrak{g}_0^{-1/2}\|_2^2\|y - x^{(0)}\|_{\mathfrak{g}_0}^2$$

$$\leq 2r^2\|\mathfrak{g}_0^{-1/2}M_0(y)^{-1}\mathfrak{g}_0^{-1/2}\|_2^2 \,,$$

where we have by Lemma 9-(b)

$$\mathfrak{g}_0^{-1/2}M_0(y)^{-1}\mathfrak{g}_0^{-1/2} = (I_d + \mathfrak{g}_0^{1/2}\mathfrak{g}(y)^{-1}\mathfrak{g}_0^{1/2})^{-1} \preceq (1 + (1-r)^2)^{-1}I_d \,. \tag{38}$$

Hence, we obtain

$$\|\mathfrak{g}_0^{-1/2} D\mathfrak{g}_0[K_0(y), K_0(y)]\|_2 \leq 2r^2(1 + (1-r)^2)^{-2} \,. \tag{39}$$

*Step 1.3.* Thirdly, we have

$$\|\mathfrak{g}_0^{1/2} DM_0(y')[w]\mathfrak{g}_0^{1/2}\|_2 \leq (1-r)^{-2}\|\mathfrak{g}(x)^{-1/2} D\mathfrak{g}(y')[w]\mathfrak{g}(x)^{-1/2}\|_2 \tag{40}$$

$$\leq 2(1-r)^{-3}\|w\|_{\mathfrak{g}_0} \,. \qquad\qquad \text{(using (36))}$$

*Step 1.4.* Fourthly, we have

$$\|\mathfrak{g}_0^{-1/2}(D\mathfrak{g}_0[K_0(y), K_0(y)] - D\mathfrak{g}_0[K_0(y'), K_0(y')])\|_2 \tag{41}$$

$$\leq 2\|D\mathfrak{g}_0[K_0(y) - K_0(y'), K_0(y)]\|_{\mathfrak{g}_0^{-1}} + \|D\mathfrak{g}_0[K_0(y) - K_0(y'), K_0(y) - K_0(y')]\|_{\mathfrak{g}_0^{-1}}$$

$$\leq 2\|K_0(y) - K_0(y')\|_{\mathfrak{g}_0}\|K_0(y)\|_{\mathfrak{g}_0} + \|K_0(y) - K_0(y')\|_{\mathfrak{g}_0}\|K_0(y) - K_0(y')\|_{\mathfrak{g}_0} \qquad \text{(Lemma 9-(a))}$$

$$\leq 4\|M_0(y)^{-1}(y - x^{(0)}) - M_0(y')^{-1}(y' - x^{(0)})\|_{\mathfrak{g}_0^{-1}}\|M_0(y)^{-1}(y - x^{(0)})\|_{\mathfrak{g}_0^{-1}}$$

$$+ 2\|M_0(y)^{-1}(y - x^{(0)}) - M_0(y')^{-1}(y' - x^{(0)})\|_{\mathfrak{g}_0^{-1}}^2 \,.$$

In particular, $M_0(y)^{-1}(y - x^{(0)}) - M_0(y')^{-1}(y' - x^{(0)}) = M_0(y)^{-1}(y - y') + (M_0(y)^{-1} - M_0(y')^{-1})(y' - x^{(0)})$, and thus we have

$$\|M_0(y)^{-1}(y - x^{(0)}) - M_0(y')^{-1}(y' - x^{(0)})\|_{\mathfrak{g}_0^{-1}} \tag{42}$$

$$\leq \|\mathfrak{g}_0^{-1/2}M_0(y)^{-1}\mathfrak{g}_0^{-1/2}\|_2\|y - y'\|_{\mathfrak{g}_0} + r\|\mathfrak{g}_0^{-1/2}(M_0(y)^{-1} - M_0(y')^{-1})\mathfrak{g}_0^{-1/2}\|_2$$

$$\leq (1 + (1-r)^2)^{-1}\|y - y'\|_{\mathfrak{g}_0} + r\|\mathfrak{g}_0^{-1/2}(M_0(y')^{-1} - M_0(y')^{-1})\mathfrak{g}_0^{-1/2}\|_2 \,. \qquad \text{(using (38))}$$

We now aim to find an upper bound for $\|\mathfrak{g}_0^{-1/2}(M_0(y)^{-1} - M_0(y')^{-1})\mathfrak{g}_0^{-1/2}\|_2$. We have

$$
\begin{aligned}
&\mathfrak{g}_0^{-1/2}(M_0(y)^{-1} - M_0(y')^{-1})\mathfrak{g}_0^{-1/2} \\
&= (I_d + \mathfrak{g}_0^{1/2}\mathfrak{g}(y)^{-1}\mathfrak{g}_0^{1/2})^{-1} - (I_d + \mathfrak{g}_0^{1/2}\mathfrak{g}(y')^{-1}\mathfrak{g}_0^{1/2})^{-1} \\
&= (I_d + \mathfrak{g}_0^{1/2}\mathfrak{g}(y')^{-1}\mathfrak{g}_0^{1/2} + \mathfrak{g}_0^{1/2}\{\mathfrak{g}(y)^{-1} - \mathfrak{g}(y')^{-1}\}\mathfrak{g}_0^{1/2})^{-1} - (I_d + \mathfrak{g}_0^{1/2}\mathfrak{g}(y')^{-1}\mathfrak{g}_0^{1/2})^{-1} \\
&= B(y')^{-1/2}[(I_d + B(y')^{-1/2}\mathfrak{g}_0^{1/2}\{\mathfrak{g}(y)^{-1} - \mathfrak{g}(y')^{-1}\}\mathfrak{g}_0^{1/2}B(y')^{-1/2})^{-1} - I_d]B(y')^{-1/2} ,
\end{aligned}
$$

where $B(y') = I_d + \mathfrak{g}_0^{1/2}\mathfrak{g}(y')^{-1}\mathfrak{g}_0^{1/2}$. In particular, we have

$$
(1 + (1-r)^{-2})^{-1/2}I_d \preceq B(y')^{-1/2} \preceq (1 + (1-r)^2)^{-1/2}I_d ,
$$

by Lemma 9-(b). Note that (17) still holds, where $x, x'$ and $\bar{r}$ are respectively replaced by $y, y'$ and $r$. Thus, on the model on (18), we have

$$
\{(1 - \|y' - y\|_{\mathfrak{g}(y)})^2 - 1\}(1-r)^2 I_d \preceq \mathfrak{g}_0^{1/2}\{\mathfrak{g}(y)^{-1} - \mathfrak{g}(y')^{-1}\}\mathfrak{g}_0^{1/2} \preceq \{(1 - \|y' - y\|_{\mathfrak{g}(y)})^{-2} - 1\}(1-r)^{-2}I_d ,
$$

and then

$$
\begin{aligned}
&(1 + \{(1 - \|y' - y\|_{\mathfrak{g}(y)})^2 - 1\}(1-r)^2(1 + (1-r)^{-2})^{-1})I_d \\
&\qquad \preceq I_d + B(y')^{-1/2}\mathfrak{g}_0^{1/2}\{\mathfrak{g}(y)^{-1} - \mathfrak{g}(y')^{-1}\}\mathfrak{g}_0^{1/2}B(y')^{-1/2} \\
&\qquad\qquad \preceq (1 + \{(1 - \|y' - y\|_{\mathfrak{g}(y)})^{-2} - 1\}(1-r)^{-2}(1 + (1-r)^2)^{-1})I_d .
\end{aligned}
$$

Therefore, we have $\|\mathfrak{g}_0^{-1/2}(M_0(y')^{-1} - M_0(y')^{-1})\mathfrak{g}_0^{-1/2}\|_2 \leq (1 + (1-r)^2)^{-1}\max(|f_1(r)|, f_2(r))$ where

$$
\begin{aligned}
f_1(r) &= (1 + \{(1 - \|y' - y\|_{\mathfrak{g}(y)})^{-2} - 1\}(1-r)^{-2}(1 + (1-r)^2)^{-1})^{-1} - 1, \\
f_2(r) &= (1 + \{(1 - \|y' - y\|_{\mathfrak{g}(y)})^2 - 1\}(1-r)^2(1 + (1-r)^{-2})^{-1})^{-1} - 1 .
\end{aligned}
$$

We now aim to control the upper bound $\max(|f_1(r)|, f_2(r))$.

(a) We first bound $|f_1(r)|$. Since $\|y' - y\|_{\mathfrak{g}(y)} \geq 0$, it is clear that $f_1(r) \leq 0$. Using Inequality (a) with $u = \|y' - y\|_{\mathfrak{g}(y)}$, where $u \leq 1/5$ by (17), we obtain

$$
\begin{aligned}
|f_1(r)| = -f_1(r) &\leq 1 - (1 + 3(1-r)^{-2}(1 + (1-r)^2)^{-1}\|y' - y\|_{\mathfrak{g}(y)})^{-1} \\
&\leq 3(1-r)^{-2}(1 + (1-r)^2)^{-1}\|y' - y\|_{\mathfrak{g}(y)} && \text{(Inequality (d))} \\
&\leq 3(1-r)^{-3}(1 + (1-r)^2)^{-1}\|y' - y\|_{\mathfrak{g}_0} . && \text{(Lemma 9-(c))}
\end{aligned}
$$

(b) We now bound $f_2(r)$. We have

$$
\begin{aligned}
f_2(r) &\leq (1 - 2(1-r)^2(1 + (1-r)^{-2})^{-1}\|y' - y\|_{\mathfrak{g}(y)})^{-1} - 1 \\
&\leq 4(1-r)^2(1 + (1-r)^{-2})^{-1}\|y' - y\|_{\mathfrak{g}(y)} \\
&\leq 4(1-r)(1 + (1-r)^{-2})^{-1}\|y' - y\|_{\mathfrak{g}_0} , && \text{(Lemma 9-(c))}
\end{aligned}
$$

where we used (i) Inequality (d) in the first line with $u = \|y' - y\|_{\mathfrak{g}(y)}$ and (ii) Inequality (b) in the second line with $u = 2(1-r)^2(1 + (1-r)^{-2})^{-1}\|y' - y\|_{\mathfrak{g}(y)} \leq 4r(1-r)(1 + (1-r)^{-2})^{-1} \leq 1/2$ with $r \leq 1/11$.

We recall that $r \in (0, 1)$, and thus, we have respectively $(1-r)^{-3} \geq (1-r)$ and $(1 + (1-r)^2)^{-1} \geq (1 + (1-r)^{-2})^{-1}$. Therefore, $\max(|f_1(r)|, f_2(r)) \leq (4/3)|f_1(r)|$ and

$$
\|\mathfrak{g}_0^{-1/2}(M_0(y')^{-1} - M_0(y')^{-1})\mathfrak{g}_0^{-1/2}\|_2 \leq 4(1-r)^{-3}(1 + (1-r)^2)^{-2}\|y' - y\|_{\mathfrak{g}_0} , \tag{43}
$$

$$
\|K_0(y) - K_0(y')\|_{\mathfrak{g}_0} \leq (1 + (1-r)^2)^{-1}\{1 + 4r(1-r)^{-3}(1 + (1-r)^2)^{-1}\}\|y' - y\|_{\mathfrak{g}_0} .
$$

By combining (41), (42) and (43), we finally have

$$\|\mathfrak{g}_0^{-1/2}(D\mathfrak{g}_0[\mathrm{K}_0(y),\mathrm{K}_0(y)] - D\mathfrak{g}_0[\mathrm{K}_0(y'),\mathrm{K}_0(y')])\|_2$$
$$\leq 4r(1+(1-r)^2)^{-2}\{1+4r(1-r)^{-3}(1+(1-r)^2)^{-1}\}\|y-y'\|_{\mathfrak{g}_0}$$
$$+ 4r(1+(1-r)^2)^{-2}\{1+4r(1-r)^{-3}(1+(1-r)^2)^{-1}\}^2\|y-y'\|_{\mathfrak{g}_0}$$
$$\leq 4r(1+(1-r)^2)^{-2}\{2+4r(1-r)^{-3}(1+(1-r)^2)^{-1}\}^2\|y-y'\|_{\mathfrak{g}_0}\,.$$

*Conclusion of Step 1.* By combining the results of Steps 1.1 to 1.4, it comes that

$$\|DM_0(y)[w]D\mathfrak{g}_0[\mathrm{K}_0(y),\mathrm{K}_0(y)] - DM_0(y')[w]D\mathfrak{g}_0[\mathrm{K}_0(y'),\mathrm{K}_0(y')]\|_{\mathfrak{g}_0}$$
$$= \|\mathfrak{g}_0^{1/2}(a_1(y)-a_1(y'))\|_2$$
$$\leq \{2r^2(1+(1-r)^2)^{-2}(12(1-r)^{-8}+5\alpha(\alpha+1)/3(1-r)^{-6})$$
$$+ 8r(1-r)^{-3}(1+(1-r)^2)^{-2}\{2+4r(1-r)^{-3}(1+(1-r)^2)^{-1}\}^2\}\|y-y'\|_{\mathfrak{g}_0}\|w\|_{\mathfrak{g}_0}\,.$$

*Step 2.* Let us now bound the second term in (34). For any $x \in \mathsf{B}_0$, we denote $\overline{M_0(y)}D\mathfrak{g}_0[DK_0(y)[w],\mathrm{K}_0(y)]$ by $a_2(x)$. We have

$$\mathfrak{g}_0^{1/2}(a_2(y)-a_2(y')) = \{\mathfrak{g}_0^{1/2}(M_0(y)-M_0(y'))\mathfrak{g}_0^{1/2}\}\{\mathfrak{g}_0^{-1/2}D\mathfrak{g}_0[DK_0(y)[w],\mathrm{K}_0(y)]\} \qquad (44)$$
$$+ \{\mathfrak{g}_0^{1/2}M_0(y')\mathfrak{g}_0^{1/2}\}\{\mathfrak{g}_0^{-1/2}(D\mathfrak{g}_0[DK_0(y)[w],\mathrm{K}_0(y)] - D\mathfrak{g}_0[DK_0(y')[w],\mathrm{K}_0(y')])\}\,.$$

We now aim to bound each of the four terms that appear in (44).

*Step 2.1.* First, we have

$$\|\mathfrak{g}_0^{1/2}(M_0(y)-M_0(y'))\mathfrak{g}_0^{1/2}\|_2 = \|\mathfrak{g}_0^{1/2}(\mathfrak{g}(y)^{-1}-\mathfrak{g}(y')^{-1})\mathfrak{g}_0^{1/2}\|_2$$
$$\leq 3(1-r)^{-3}\|y-y'\|_{\mathfrak{g}_0}\,. \qquad \text{(on the model of (18))}$$

*Step 2.2.* Secondly, we have

$$\|\mathfrak{g}_0^{-1/2}D\mathfrak{g}_0[DK_0(y)[w],\mathrm{K}_0(y)]\|_2 \leq 2\|DK_0(y)[w]\|_{\mathfrak{g}_0}\|M_0(y)^{-1}(y-x^{(0)})\|_{\mathfrak{g}_0^{-1}} \quad \text{(Lemma 9-(a))}$$
$$\leq 2r(1+(1-r)^2)^{-1}\|DK_0(y)[w]\|_{\mathfrak{g}_0}\,, \qquad \text{(see Step 1.2.)}$$

where

$$\|DK_0(y)[w]\|_{\mathfrak{g}_0} \leq r\|\mathfrak{g}_0^{-1/2}(\mathfrak{g}(y)\mathfrak{g}_0^{-1}+\mathrm{I}_d)^{-1}\mathfrak{g}_0^{1/2}\|_2^2\|\mathfrak{g}_0^{-1/2}\mathfrak{g}(y)^{1/2}\|_2^2\|\mathfrak{g}(y)^{-1/2}D\mathfrak{g}(y)[w]\mathfrak{g}(y)^{-1/2}\|_2$$
$$+ \quad \|M_0(y)^{-1}w\|_{\mathfrak{g}_0^{-1}}\,.$$

Using (36) and (38), it comes that

$$\|DK_0(y)[w]\|_{\mathfrak{g}_0} \leq \{2r(1-r)^{-3}(1+(1-r)^2)^{-2}+(1+(1-r)^2)^{-1}\}\|w\|_{\mathfrak{g}_0}\,.$$

Then, we have

$$\|\mathfrak{g}_0^{-1/2}D\mathfrak{g}_0[DK_0(y)[w],\mathrm{K}_0(y)]\|_2 \leq 2r(1+(1-r)^2)^{-2}\{1+2r(1-r)^{-3}(1+(1-r)^2)^{-1}\}\|w\|_{\mathfrak{g}_0}\,.$$

*Step 2.3.* Thirdly, we have $\|\mathfrak{g}_0^{1/2}M_0(y')\mathfrak{g}_0^{1/2}\|_2 = \|\mathfrak{g}_0^{1/2}\mathfrak{g}(y')^{-1}\mathfrak{g}_0^{1/2}+\mathrm{I}_d\|_2 \leq 1+(1-r)^{-2}$.

*Step 2.4.* Fourthly, using Lemma 9-(a), we have

$$\|\mathfrak{g}_0^{-1/2}(D\mathfrak{g}_0[DK_0(y)[w],\mathrm{K}_0(y)] - D\mathfrak{g}_0[DK_0(y')[w],\mathrm{K}_0(y')])\|_2$$
$$\leq \|D\mathfrak{g}_0[DK_0(y)[w]-DK_0(y')[w],\mathrm{K}_0(y)]\|_{\mathfrak{g}_0^{-1}} + \|D\mathfrak{g}_0[DK_0(y)[w],\mathrm{K}_0(y)-\mathrm{K}_0(y')]\|_{\mathfrak{g}_0^{-1}}$$
$$+ \|D\mathfrak{g}_0[DK_0(y)[w]-DK_0(y')[w],\mathrm{K}_0(y)-\mathrm{K}_0(y')]\|_{\mathfrak{g}_0^{-1}}$$
$$\leq 2\|DK_0(y)[w]-DK_0(y')[w]\|_{\mathfrak{g}_0}\|\mathrm{K}_0(y)\|_{\mathfrak{g}_0} + 2\|DK_0(y)[w]\|_{\mathfrak{g}_0}\|\mathrm{K}_0(y)-\mathrm{K}_0(y')\|_{\mathfrak{g}_0}$$
$$+ 2\|DK_0(y)[w]-DK_0(y')[w]\|_{\mathfrak{g}_0}\|\mathrm{K}_0(y)-\mathrm{K}_0(y')\|_{\mathfrak{g}_0}$$
$$\leq 2\{\|\mathrm{K}_0(y)\|_{\mathfrak{g}_0}+\|\mathrm{K}_0(y)-\mathrm{K}_0(y')\|_{\mathfrak{g}_0}\}\|DK_0(y)[w]-DK_0(y')[w]\|_{\mathfrak{g}_0}$$
$$+ 2\|DK_0(y)[w]\|_{\mathfrak{g}_0}\|\mathrm{K}_0(y)-\mathrm{K}_0(y')\|_{\mathfrak{g}_0}\,.$$

We recall that the following inequalities hold
$$\|\mathrm{K}_0(y)]\|_{\mathfrak{g}_0} \le r(1 + (1-r)^2)^{-1} ,$$
$$\|D\mathrm{K}_0(y)[w]\|_{\mathfrak{g}_0} \le (1 + (1-r)^2)^{-1}\{1 + 2r(1-r)^{-3}(1 + (1-r)^2)^{-1}\}\|w\|_{\mathfrak{g}_0} ,$$
$$\|\mathrm{K}_0(y) - \mathrm{K}_0(y')\|_{\mathfrak{g}_0} \le (1 + (1-r)^2)^{-1}\{1 + 4r(1-r)^{-3}(1 + (1-r)^2)^{-1}\}\|y' - y\|_{\mathfrak{g}_0} .$$
We are now going to bound $\|D\mathrm{K}_0(y)[w] - D\mathrm{K}_0(y')[w]\|_{\mathfrak{g}_0}$. We have
$$\|D\mathrm{K}_0(y)[w] - D\mathrm{K}_0(y')[w]\|_{\mathfrak{g}_0} \le \|(\mathrm{M}_0(y)^{-1} - \mathrm{M}_0(y')^{-1})w\|_{\mathfrak{g}_0^{-1}} + A_1 + A_2 + A_3 + A_4 ,$$

where
$$A_1 = \|\{(\mathrm{I}_d + \mathfrak{g}_0^{-1/2}\mathfrak{g}(y)\mathfrak{g}_0^{-1/2})^{-1} - (\mathrm{I}_d + \mathfrak{g}_0^{-1/2}\mathfrak{g}(y')\mathfrak{g}_0^{-1/2})^{-1}\}$$
$$\times \mathfrak{g}_0^{-1/2}D\mathfrak{g}(y)[w]\mathfrak{g}_0^{-1/2}(\mathrm{I}_d + \mathfrak{g}_0^{-1/2}\mathfrak{g}(y)\mathfrak{g}_0^{-1/2})^{-1}\mathfrak{g}_0^{1/2}(y - x^{(0)})\|_2 ,$$
$$A_2 = \|(\mathrm{I}_d + \mathfrak{g}_0^{-1/2}\mathfrak{g}(y')\mathfrak{g}_0^{-1/2})^{-1}\mathfrak{g}_0^{-1/2}\{D\mathfrak{g}(y)[w] - D\mathfrak{g}(y')[w]\}\mathfrak{g}_0^{-1/2}(\mathrm{I}_d + \mathfrak{g}_0^{-1/2}\mathfrak{g}(y)\mathfrak{g}_0^{-1/2})^{-1}\mathfrak{g}_0^{1/2}(y - x^{(0)})\|_2 ,$$
$$A_3 = \|(\mathrm{I}_d + \mathfrak{g}_0^{-1/2}\mathfrak{g}(y')\mathfrak{g}_0^{-1/2})^{-1}\mathfrak{g}_0^{-1/2}D\mathfrak{g}(y)[w]\mathfrak{g}_0^{-1/2}$$
$$\times \{(\mathrm{I}_d + \mathfrak{g}_0^{-1/2}\mathfrak{g}(y)\mathfrak{g}_0^{-1/2})^{-1} - (\mathrm{I}_d + \mathfrak{g}_0^{-1/2}\mathfrak{g}(y')\mathfrak{g}_0^{-1/2})^{-1}\}\mathfrak{g}_0^{1/2}(y - x^{(0)})\|_2 ,$$
$$A_3 = \|(\mathrm{I}_d + \mathfrak{g}_0^{-1/2}\mathfrak{g}(y')\mathfrak{g}_0^{-1/2})^{-1}\mathfrak{g}_0^{-1/2}D\mathfrak{g}(y)[w]\mathfrak{g}_0^{-1/2}(\mathrm{I}_d + \mathfrak{g}_0^{-1/2}\mathfrak{g}(y')\mathfrak{g}_0^{-1/2})^{-1}\mathfrak{g}_0^{1/2}(y - y')\|_2 .$$
In particular, we have
$$\|(\mathrm{M}_0(y')^{-1} - \mathrm{M}_0(y')^{-1})w\|_{\mathfrak{g}_0^{-1}} \le 4(1-r)^{-3}(1 + (1-r)^2)^{-2}\|y' - y\|_{\mathfrak{g}_0}\|w\|_{\mathfrak{g}_0} , \qquad ((43))$$
$$\max(A_1, A_3) \le 8r(1-r)^{-6}(1 + (1-r)^2)^{-3}\|y - y'\|_{\mathfrak{g}_0}\|w\|_{\mathfrak{g}_0} , \qquad ((36),\ (38),\ (43))$$
$$A_2 \le 5\alpha(\alpha + 1)/3r(1-r)^{-4}(1 + (1-r)^2)^{-2}\|y - y'\|_{\mathfrak{g}_0}\|w\|_{\mathfrak{g}_0} , \qquad ((37),\ (38))$$
$$A_4 \le 2(1-r)^{-3}(1 + (1-r)^2)^{-2}\|y - y'\|_{\mathfrak{g}_0}\|w\|_{\mathfrak{g}_0} . \qquad ((36)\ \text{and}\ (38))$$
Therefore, we have
$$\|\mathfrak{g}_0^{-1/2}(D\mathfrak{g}_0[D\mathrm{K}_0(y)[w], \mathrm{K}_0(y)] - D\mathfrak{g}_0[D\mathrm{K}_0(y')[w], \mathrm{K}_0(y')])\|_2$$
$$\le 2r(1 + (1-r)^2)^{-1}\{3 + 4r(1-r)^{-3}(1 + (1-r)^2)^{-1})\}$$
$$\times \{4(1-r)^{-3}(1 + (1-r)^2)^{-2} + 16r(1-r)^{-6}(1 + (1-r)^2)^{-3}$$
$$+ 5\alpha(\alpha + 1)/3r(1-r)^{-4}(1 + (1-r)^2)^{-2} + 2(1-r)^{-3}(1 + (1-r)^2)^{-2}\}\|y - y'\|_{\mathfrak{g}_0}\|w\|_{\mathfrak{g}_0}$$
$$+ 2\{1 + 2r(1-r)^{-3}(1 + (1-r)^2)^{-1}\}$$
$$\times (1 + (1-r)^2)^{-2}(1 + 4r(1-r)^{-3}(1 + (1-r)^2)^{-1})\|y - y'\|_{\mathfrak{g}_0}\|w\|_{\mathfrak{g}_0} .$$

*Conclusion of Step 2.* By combining the results of Steps 2.1 to 2.4, it comes that
$$\|\mathrm{M}_0(y)D\mathfrak{g}_0[D\mathrm{K}_0(y)[w], \mathrm{K}_0(y)] - \mathrm{M}_0(y')D\mathfrak{g}_0[D\mathrm{K}_0(y')[w], \mathrm{K}_0(y')]\|_{\mathfrak{g}_0}$$
$$= \|\mathfrak{g}_0^{1/2}(a_2(y) - a_2(y'))\|_2$$
$$\le \{6r(1-r)^{-3}(1 + (1-r)^2)^{-2}[1 + 2r(1-r)^{-3}(1 + (1-r)^2)^{-1}]$$
$$+ 2r[1 + (1-r)^{-2}](1 + (1-r)^2)^{-1}\{3 + 4r(1-r)^{-3}(1 + (1-r)^2)^{-1})\}$$
$$\times \{4(1-r)^{-3}(1 + (1-r)^2)^{-2} + 16r(1-r)^{-6}(1 + (1-r)^2)^{-3}$$
$$+ 5\alpha(\alpha + 1)/3r(1-r)^{-4}(1 + (1-r)^2)^{-2} + 2(1-r)^{-3}(1 + (1-r)^2)^{-2}\}$$
$$+ 2[1 + (1-r)^{-2}]\{1 + 2r(1-r)^{-3}(1 + (1-r)^2)^{-1}\}$$
$$\times (1 + (1-r)^2)^{-2}(1 + 4r(1-r)^{-3}(1 + (1-r)^2)^{-1})\}\|y - y'\|_{\mathfrak{g}_0}\|w\|_{\mathfrak{g}_0} .$$

*Conclusion.* Finally, we have $\|D\phi_0(y)[w] - D\phi_0(y')[w]\|_{\mathfrak{g}_0} \le c(r)\|y - y'\|_{\mathfrak{g}_0}\|w\|_{\mathfrak{g}_0}$, where
$$c(r) \le 24r^2(1-r)^{-12} + 10\alpha(\alpha + 1)/3r^2(1-r)^{-10}$$
$$+ 32r(1-r)^{-7} + 128r^2(1-r)^{-12} + 128r^3(1-r)^{-17}$$
$$+ 12r(1-r)^{-7} + 24r^2(1-r)^{-12}$$
$$+ \{48r(1-r)^{-9} + 192r^2(1-r)^{-14} + 20\alpha(\alpha + 1)r^2(1-r)^{-10} + 24r(1-r)^{-9}\}\{1 + (1-r)^{-2}\}$$
$$+ \{64r^2(1-r)^{-14} + 256r^3(1-r)^{-19} + 80\alpha(\alpha + 1)/3r^3(1-r)^{-15} + 32r^2(1-r)^{-14}\}\{1 + (1-r)^{-2}\}$$
$$+ \{4(1-r)^{-4} + 24r(1-r)^{-9} + 32r^2(1-r)^{-14}\}\{1 + (1-r)^{-2}\} ,$$

by combining the results of Step 1 (two first lines) and Step 2 (following lines). Hence, using that $r \leq 1/11$, we have

$$
\begin{aligned}
c(r) &\leq \max(24 \times 256, 5 \times 80\alpha(\alpha + 1)/3)(1 - r)^{-21} \\
&\leq \max(6144, 400\alpha(\alpha + 1)/3)15/2 \\
&\leq \max(46080, 1000\alpha(\alpha + 1)) \\
&\leq 1/r_\alpha^\star , \qquad\qquad\qquad\qquad\qquad\qquad \text{(see (33))}
\end{aligned}
$$

*i.e.*, for any $(y, y') \in \mathsf{B}_0 \times \mathsf{B}_0$ and $w \in \mathbb{R}^d$, we have

$$
\|D\phi_0(y)[w] - D\phi_0(y)[w]\|_{\mathfrak{g}_0} \leq (r_\alpha^\star)^{-1}\|y - y'\|_{\mathfrak{g}_0}\|w\|_{\mathfrak{g}_0} ,
$$

and thus

$$
\|D\phi_0(y) - D\phi_0(y)\|_{\mathfrak{g}_0} \leq (r_\alpha^\star)^{-1}\|y - y'\|_{\mathfrak{g}_0} .
$$

This last inequality finally proves that $D\phi_0$ is $(r_\alpha^\star)^{-1}$-Lipschitz-continuous on $\mathsf{B}_0$ with respect to $\|\cdot\|_{\mathfrak{g}_0}$.

**Inequality on $D\tilde{\phi}_0$.** Elaborating on the smoothness of $D\phi_0$, we prove here that $\|D\tilde{\phi}_0(y)\|_{\mathfrak{g}_0} > 1/2$ for any $y \in \mathsf{B}_0$. Let $y \in \mathsf{B}_0$, we have

$$
\begin{aligned}
\|\mathrm{I}_d\|_{\mathfrak{g}_0} &= \|D\tilde{\phi}_0(y) + 1/2D\phi_0(y)\|_{\mathfrak{g}_0} , \\
1 &\leq \|D\tilde{\phi}_0(y)\|_{\mathfrak{g}_0} + 1/2\|D\phi_0(y)\|_{\mathfrak{g}_0} , \\
1 - 1/2\|D\phi_0(y)\|_{\mathfrak{g}_0} &\leq \|D\tilde{\phi}_0(y)\|_{\mathfrak{g}_0} .
\end{aligned}
$$

We recall that $D\phi_0(x^{(0)}) = 0$ and that $D\phi_0$ is $(r_\alpha^\star)^{-1}$-Lipschitz-continuous on $\mathsf{B}_0$. Thus, we obtain

$$
\|D\phi_0(y)\|_{\mathfrak{g}_0} = \|D\phi_0(y) - D\phi_0(x^{(0)})\|_{\mathfrak{g}_0} \leq (r_\alpha^\star)^{-1}\|y - x^{(0)}\|_{\mathfrak{g}_0} < (r_\alpha^\star)^{-1}r \leq 1 ,
$$

and then $\|D\tilde{\phi}_0(y)\|_{\mathfrak{g}_0} > 1/2$, which proves the result.

**Smoothness of $\phi_0$ and $\tilde{\phi}_0$.** We prove here that $\phi_0$ (and thus $\tilde{\phi}_0$) is Lipschitz-continuous on $\mathsf{B}_0$ with respect to $\|\cdot\|_{\mathfrak{g}_0}$. Let $(y, y') \in \mathsf{B}_0 \times \mathsf{B}_0$. We have

$$
\begin{aligned}
\|\phi_0(y) - \phi_0(y')\|_{\mathfrak{g}_0} &\leq \|\{\mathrm{M}_0(y) - \mathrm{M}_0(y')\}D\mathfrak{g}_0[\mathrm{K}_0(y), \mathrm{K}_0(y)]\|_{\mathfrak{g}_0} \qquad (45) \\
&\quad + \|\mathrm{M}_0(y')\{D\mathfrak{g}_0[\mathrm{K}_0(y), \mathrm{K}_0(y)] - D\mathfrak{g}_0[\mathrm{K}_0(y'), \mathrm{K}_0(y')]\}\|_{\mathfrak{g}_0} .
\end{aligned}
$$

To prove the Lipschitz-continuity of $\phi_0$, we are going to proceed in two steps, each of them consisting of bounding from above a term of (45).

*Step 1.* Let us first bound the first term in (45). Since $r \leq 1/11$, we have

$$
\begin{aligned}
&\|\{\mathrm{M}_0(y) - \mathrm{M}_0(y')\}D\mathfrak{g}_0[\mathrm{K}_0(y), \mathrm{K}_0(y)]\|_{\mathfrak{g}_0} \\
&\qquad \leq \|\mathfrak{g}_0^{1/2}\{\mathrm{M}_0(y) - \mathrm{M}_0(y')\}\mathfrak{g}_0^{1/2}\|_2\|D\mathfrak{g}_0[\mathrm{K}_0(y), \mathrm{K}_0(y)]\|_{\mathfrak{g}_0^{-1}} \\
&\qquad \leq 3(1 - r)^{-3}\|y - y'\|_{\mathfrak{g}_0} \times 2\|\mathrm{K}_0(y)\|_{\mathfrak{g}_0}^2 \qquad\qquad \text{(see (18) and Lemma 9-(a))} \\
&\qquad \leq 6r^2(1 - r)^{-3}(1 + (1 - r)^2)^{-2}\|y - y'\|_{\mathfrak{g}_0} . \qquad \text{(see (39))}
\end{aligned}
$$

*Step 2.* Let us now bound the second term in (45). We have

$$
\begin{aligned}
&\|\mathrm{M}_0(y')\{D\mathfrak{g}_0[\mathrm{K}_0(y), \mathrm{K}_0(y)] - D\mathfrak{g}_0[\mathrm{K}_0(y'), \mathrm{K}_0(y')]\}\|_{\mathfrak{g}_0} \\
&\qquad \leq \|\mathfrak{g}_0^{1/2}\mathrm{M}_0(y')\mathfrak{g}_0^{1/2}\|_2\|D\mathfrak{g}_0[\mathrm{K}_0(y), \mathrm{K}_0(y)] - D\mathfrak{g}_0[\mathrm{K}_0(y'), \mathrm{K}_0(y')]\|_{\mathfrak{g}_0^{-1}} \\
&\qquad \leq \{1 + (1 - r)^{-2}\}\{2\|D\mathfrak{g}_0[\mathrm{K}_0(y) - \mathrm{K}_0(y'), \mathrm{K}_0(y)]\|_{\mathfrak{g}_0^{-1}} + \|D\mathfrak{g}_0[\mathrm{K}_0(y) - \mathrm{K}_0(y'), \mathrm{K}_0(y) - \mathrm{K}_0(y')]\|_{\mathfrak{g}_0^{-1}}\} \\
&\qquad \leq \{1 + (1 - r)^{-2}\}\{4\|\mathrm{K}_0(y) - \mathrm{K}_0(y')\|_{\mathfrak{g}_0}\|\mathrm{K}_0(y)\|_{\mathfrak{g}_0} + 2\|\mathrm{K}_0(y) - \mathrm{K}_0(y')\|_{\mathfrak{g}_0}^2\} \\
&\qquad \leq 1\{1 + (1 - r)^{-2}\}\|\mathrm{K}_0(y) - \mathrm{K}_0(y')\|_{\mathfrak{g}_0}\{2\|\mathrm{K}_0(y)\|_{\mathfrak{g}_0} + \|\mathrm{K}_0(y) - \mathrm{K}_0(y')\|_{\mathfrak{g}_0}\} ,
\end{aligned}
$$

where we recall that the following inequalities hold

$$\|K_0(y)]\|_{\mathfrak{g}_0} \le r(1 + (1-r)^2)^{-1} ,$$

$$\|K_0(y) - K_0(y')\|_{\mathfrak{g}_0} \le (1 + (1-r)^2)^{-1}\{1 + 4r(1-r)^{-3}(1 + (1-r)^2)^{-1}\}\|y' - y\|_{\mathfrak{g}_0} .$$

Thus, we have

$$\|M_0(y')\{D\mathfrak{g}_0[K_0(y), K_0(y)] - D\mathfrak{g}_0[K_0(y'), K_0(y')]\}\|_{\mathfrak{g}_0}$$
$$\le 4r\{1 + (1-r)^2\}^{-2}\{1 + 4r(1-r)^{-3}(1 + (1-r)^2)^{-1}\}^2\|y - y'\|_{\mathfrak{g}_0} .$$

_Conclusion._ Since $r \le 1/11$, we finally have

$$\|\phi_0(y) - \phi_0(y')\|_{\mathfrak{g}_0} \le \{6r^2(1-r)^{-3}(1 + (1-r)^2)^{-2}$$
$$4r\{1 + (1-r)^2\}^{-2}\{1 + 4r(1-r)^{-3}(1 + (1-r)^2)^{-1}\}^2\}\|y - y'\|_{\mathfrak{g}_0}$$
$$\le \{6r^2(1-r)^{-7} + 4r(1-r)^{-4} + 32r^2(1-r)^{-9} + 64r^3(1-r)^{-14}\}\|y - y'\|_{\mathfrak{g}_0}$$
$$\le 4 \times 64r(1-r)^{-14}\|y - y'\|_{\mathfrak{g}_0}$$
$$\le 4864/5r\|y - y'\|_{\mathfrak{g}_0} .$$

Therefore, $\phi_0$ and $\tilde{\phi}_0$ are respectively $(4864/5r)$ and $(1 + 2432/5r)$-Lipschitz-continuous on $B_0$ with respect to $\|\cdot\|_{\mathfrak{g}_0}$.

**We are now ready to prove the second result of Lemma 22.**

**Invertibility of** $\mathrm{Jac}[G_{h,x^{(0)}}]$. Let $x \in B_0$ and $w \in \mathbb{R}^d$. We have $(h/2)G_{h,x^{(0)}} = M_0(y)^{-1}\tilde{\phi}_0(y)$ and thus,

$$(h/2)DG_{h,x^{(0)}}(x)[w] = M_0(x)^{-1}D\tilde{\phi}_0(x)[w] + DM_0(x)^{-1}[w]\tilde{\phi}_0(x) ,$$
$$D\tilde{\phi}_0(x)[w] = (h/2)M_0(x)DG_{h,x^{(0)}}(x)[w] - M_0(x)DM_0(x)^{-1}[w]\tilde{\phi}_0(x) .$$

Then, we have

$$\|D\tilde{\phi}_0(x)[w]\|_{\mathfrak{g}_0} \le (h/2)\|\mathfrak{g}_0^{1/2}M_0(x)\mathfrak{g}_0^{1/2}\|_2\|DG_{h,x^{(0)}}(x)[w]\|_{\mathfrak{g}_0^{-1}} \tag{46}$$
$$+ \|\mathfrak{g}_0^{1/2}M_0(x)\mathfrak{g}_0^{1/2}\|_2\|\mathfrak{g}_0^{-1/2}DM_0(x)^{-1}[w]\mathfrak{g}_0^{-1/2}\|_2\|\tilde{\phi}_0(x)\|_{\mathfrak{g}_0}$$
$$\le \{1 + (1-r)^{-2}\}\{(h/2)\|DG_{h,x^{(0)}}(x)[w]\|_{\mathfrak{g}_0^{-1}} + \|\mathfrak{g}_0^{-1/2}DM_0(x)^{-1}[w]\mathfrak{g}_0^{-1/2}\|_2\|\tilde{\phi}_0(x)\|_{\mathfrak{g}_0}\} ,$$

where $DM_0(x)^{-1}[w] = -M_0(x)^{-1}DM_0(x)[w]M_0(x)^{-1}$ and

$$\|\mathfrak{g}_0^{-1/2}DM_0(x)^{-1}[w]\mathfrak{g}_0^{-1/2}\|_2 \le \|\mathfrak{g}_0^{-1/2}M_0(x)^{-1}\mathfrak{g}_0^{-1/2}\|_2^2\|\mathfrak{g}_0^{-1/2}DM_0(x)[w]\mathfrak{g}_0^{-1/2}\|_2 \tag{47}$$
$$\le 2(1-r)^{-3}(1 + (1-r)^2)^{-1}\|w\|_{\mathfrak{g}_0} \qquad \text{(see (40))}$$
$$\le 2(1-r)^{-7}\|w\|_{\mathfrak{g}_0} .$$

Moreover, using the Lipschitz-continuity of $\tilde{\phi}_0$ on $B_0$ and $\tilde{\phi}_0(x^{(0)}) = 0$, we have

$$\|\tilde{\phi}_0(x)\|_{\mathfrak{g}_0} \le \|\tilde{\phi}_0(x) - \tilde{\phi}_0(x^{(0)})\|_{\mathfrak{g}_0} + \|\tilde{\phi}_0(x^{(0)})\|_{\mathfrak{g}_0} \le (1 + 2432/5r)\|x - x^{(0)}\|_{\mathfrak{g}_0} < (1 + 2432/5r)r . \tag{48}$$

Since $r \le 1/11$, we have $1 + (1-r)^{-2} \le 5/2$, and by combining (46),(47) and(48), we obtain

$$\|D\tilde{\phi}_0(x)[w]\| \le (5h/4)\|DG_{h,x^{(0)}}(x)[w]\|_{\mathfrak{g}_0^{-1}} + 4874(1-r)^{-7}r\|w\|_{\mathfrak{g}_0} .$$

Since $\|D\tilde{\phi}_0(x)[w]\|_{\mathfrak{g}_0} > \|w\|_{\mathfrak{g}_0}/2$ and $(1-r)^{-7} \le 2$ with $r \le 1/11$, the last inequality becomes

$$(5h/4)\|DG_{h,x^{(0)}}(x)[w]\|_{\mathfrak{g}_0^{-1}} \ge \{1/2 - 9748r\}\|w\|_{\mathfrak{g}_0} ,$$
$$\ge 1/2 - r/(4r_\alpha^\star) \ge 1/4 . \qquad \text{(see (33))}$$

In particular, $\mathrm{Jac}[G_{h,x^{(0)}}](x)$ is invertible, which concludes the proof. $\qquad\square$

We prove below that, for any $z^{(0)} \in T^\star M$, the set $\underline{G}_h(z^{(0)})$ is reduced to a single point, for $h$ small enough depending on $z^{(0)}$.

**Lemma 23.** *For any $w \geq 0$, any $r \in (0,1)$ and any $\alpha \geq 1$, we define*

$$h_1(w,r,\alpha) = \min\left(\frac{1}{1+(1-r)^{-2}+2(2r+w)}, \frac{(1-r)^3}{3(r+w)}, \frac{r}{(r+w)(1+3r/2)+1/2(r+w)^2}\right).$$
(49)

*We recall that the set-valued map $\underline{G}_h$ is defined in (29). Assume A1, A2. Then, for any $r \in (0,1/11]$, any $z = (x,p) \in T^\star M$ and any $h \in (0, h_1(\|p\|_{\mathfrak{g}(x)^{-1}}, r, \alpha))$, there exists a unique $z' \in \mathrm{B}_{\|\cdot\|_z}(z,r) \subset T^\star M$ such that $z' \in \underline{G}_h(z)$.*

*Proof.* Assume **A1**, **A2**. Let $r \in (0,1/11]$, $z = (x,p) \in T^\star M$ and $h \in (0, h_1(\|p\|_{\mathfrak{g}(x)^{-1}}, r, \alpha))$, where $h_1$ is defined in (49). We recall that the map $\underline{g}_h$ is defined in (30). We define $\mathrm{B} = \mathrm{B}_{\|\cdot\|_z}(z,r)$ and the map $\underline{g}_{h,z} : T^\star M \to \mathbb{R}^n \times \mathbb{R}^n$ by $\underline{g}_{h,z}(z') = z' - \underline{g}_h(z,z')$ for any $z' \in T^\star M$. Then, $\underline{g}_{h,z}(z') = z'$ if and only if $z' \in \underline{G}_h(z)$. The proof is divided in two steps.

*Step 1.* We first prove that $\underline{g}_{h,z}(\mathrm{B}) \subset \mathrm{B}$. Let $z' \in \mathrm{B}$, we have by Lemma 9-(a)

$$
\begin{aligned}
\|\underline{g}_{h,z}(z') - z\|_z &= \tfrac{h}{2}\|\{\mathfrak{g}(x)^{-1} + \mathfrak{g}(x')^{-1}\}p'\|_{\mathfrak{g}(x)} + \tfrac{h}{4}\|D\mathfrak{g}(x)[\mathfrak{g}(x)^{-1}p', \mathfrak{g}(x)^{-1}p']\|_{\mathfrak{g}(x)^{-1}} \\
&= \tfrac{h}{2}\|2\mathfrak{g}(x)^{-1}p' + \{\mathfrak{g}(x')^{-1} - \mathfrak{g}(x)^{-1}\}p'\|_{\mathfrak{g}(x)} + \tfrac{h}{4}\|D\mathfrak{g}(x)[\mathfrak{g}(x)^{-1}p', \mathfrak{g}(x)^{-1}p']\|_{\mathfrak{g}(x)^{-1}} \\
&\leq \tfrac{h}{2}(2\|p'\|_{\mathfrak{g}(x)^{-1}} + \|\mathfrak{g}(x)^{1/2}(\mathfrak{g}(x')^{-1} - \mathfrak{g}(x)^{-1})\mathfrak{g}(x)^{1/2}\|_2\|p'\|_{\mathfrak{g}(x)^{-1}}) + \tfrac{h}{2}\|p'\|^2_{\mathfrak{g}(x)^{-1}},
\end{aligned}
$$

where the following inequalities hold

(a) $\|p'\|_{\mathfrak{g}(x)^{-1}} = \|p' - p + p\|_{\mathfrak{g}(x)^{-1}} \leq r + \|p\|_{\mathfrak{g}(x)^{-1}}$.

(b) $\|\mathfrak{g}(x)^{1/2}(\mathfrak{g}(x')^{-1} - \mathfrak{g}(x)^{-1})\mathfrak{g}(x)^{1/2}\|_2 \leq 3\|x - x'\|_{\mathfrak{g}(x)} \leq 3r$, on the model of (18), using $r \leq 1/11$.

Therefore, we have

$$
\begin{aligned}
\|\underline{g}_{h,z}(z') - z\|_z &\leq h((r + \|p\|_{\mathfrak{g}(x)^{-1}}) + \tfrac{3}{2}r(r + \|p\|_{\mathfrak{g}(x)^{-1}}) + \tfrac{1}{2}(r + \|p\|_{\mathfrak{g}(x)^{-1}})^2) \\
&\leq h((r + \|p\|_{\mathfrak{g}(x)^{-1}})(1 + \tfrac{3}{2}r) + \tfrac{1}{2}(r + \|p\|_{\mathfrak{g}(x)^{-1}})^2) \\
&\leq hr/h_1(\|p\|_{\mathfrak{g}(x)^{-1}}, r, \alpha) < r,
\end{aligned}
$$

which proves the statement.

*Step 2.* We now prove that $\underline{g}_{h,z}$ is a contraction on $\mathrm{B}$. This proof notably recovers some elements of the proof of Proposition 13 (see Appendix E). Let $(z_1, z_2) \in \mathrm{B} \times \mathrm{B}$ with $z_1 = (x_1, p_1)$ and $z_2 = (x_2, p_2)$. Remark that $x_1, x_2 \in \mathrm{W}^0(x,1) \times \mathrm{W}^0(x,1)$. Let us first bound $\|\underline{g}^{(1)}_{h,z}(z_1) - \underline{g}^{(1)}_{h,z}(z_2)\|_{\mathfrak{g}(x)}$. We have

$$
\begin{aligned}
\underline{g}^{(1)}_{h,z}(z_1) - \underline{g}^{(1)}_{h,z}(z_2) &= \tfrac{h}{2}\{\mathfrak{g}(x)^{-1}(p_1 - p_2) + \mathfrak{g}(x_1)^{-1}p_1 - \mathfrak{g}(x_2)^{-1}p_2\} \\
&= \tfrac{h}{2}\{\mathfrak{g}(x)^{-1}(p_1 - p_2) + \mathfrak{g}(x_1)^{-1}(p_1 - p_2) + (\mathfrak{g}(x_1)^{-1} - \mathfrak{g}(x_2)^{-1})p_2\},
\end{aligned}
$$

and then, since $r \leq 1/11$, we have on the model of (18)

$$
\begin{aligned}
\|\underline{g}^{(1)}_{h,z}(z_1) - \underline{g}^{(1)}_{h,z}(z_2)\|_{\mathfrak{g}(x)} &\leq \tfrac{h}{2}\{1 + \|\mathfrak{g}(x)^{1/2}\mathfrak{g}(x_1)^{-1}\mathfrak{g}(x)^{1/2}\|_2\}\|p_1 - p_2\|_{\mathfrak{g}(x)^{-1}} \\
&\quad + \tfrac{h}{2}\|\mathfrak{g}(x)^{1/2}(\mathfrak{g}(x_1)^{-1} - \mathfrak{g}(x_2)^{-1})\mathfrak{g}(x)^{1/2}\|_2\|p_2\|_{\mathfrak{g}(x)^{-1}} \\
&\leq \tfrac{h}{2}\{1 + (1-r)^{-2}\}\|p_1 - p_2\|_{\mathfrak{g}(x)^{-1}} + \tfrac{3h}{2}(1-r)^{-3}\|x_1 - x_2\|_{\mathfrak{g}(x)}(r + \|p\|_{\mathfrak{g}(x)^{-1}}).
\end{aligned}
$$

Let us now bound $\|\underline{g}^{(2)}_{h,z}(z_1) - \underline{g}^{(2)}_{h,z}(z_2)\|_{\mathfrak{g}(x)^{-1}}$. We have

$$\underline{g}^{(2)}_{h,z}(z_1) - \underline{g}^{(2)}_{h,z}(z_2) = \tfrac{h}{4}\{D\mathfrak{g}(x)[\mathfrak{g}(x)^{-1}p_1, \mathfrak{g}(x)^{-1}p_1] - D\mathfrak{g}(x)[\mathfrak{g}(x)^{-1}p_2, \mathfrak{g}(x)^{-1}p_2]\},$$

which gives using Lemma 9-(a)

$$\|\underline{\mathfrak{g}}_{h,z}^{(2)}(z_1) - \underline{\mathfrak{g}}_{h,z}^{(2)}(z_2)\|_{\mathfrak{g}(x)^{-1}} = \frac{h}{2}\|D\mathfrak{g}(x)[\mathfrak{g}(x)^{-1}(p_1-p_2),\mathfrak{g}(x)^{-1}p_2]\|_{\mathfrak{g}(x)^{-1}}$$
$$+ \frac{h}{4}\|D\mathfrak{g}(x)[\mathfrak{g}(x)^{-1}(p_1-p_2),\mathfrak{g}(x)^{-1}(p_1-p_2)]\|_{\mathfrak{g}(x)^{-1}}$$
$$\leq h\|\mathfrak{g}(x)^{-1}(p_1-p_2)\|_{\mathfrak{g}(x)}\|\mathfrak{g}(x)^{-1}p_2\|_{\mathfrak{g}(x)} + \frac{h}{2}\|\mathfrak{g}(x)^{-1}(p_1-p_2)\|_{\mathfrak{g}(x)}^2$$
$$\leq h(r + \|p\|_{\mathfrak{g}(x)^{-1}})\|p_1-p_2\|_{\mathfrak{g}(x)^{-1}} + hr\|p_1-p_2\|_{\mathfrak{g}(x)^{-1}} .$$

Finally, it comes that

$$\|\underline{\mathfrak{g}}_{h,z}(z_1) - \underline{\mathfrak{g}}_{h,z}(z_2)\|_z \leq \frac{h}{2}\{1 + (1-r)^{-2} + 4r + 2\|p\|_{\mathfrak{g}(x)^{-1}}\}\|p_1-p_2\|_{\mathfrak{g}(x)^{-1}}$$
$$+ \frac{3h}{2}(1-r)^{-3}(r + \|p\|_{\mathfrak{g}(x)^{-1}})\|x_1-x_2\|_{\mathfrak{g}(x)}$$
$$\leq (1/2)h\{\|x_1-x_2\|_{\mathfrak{g}(x)} + \|p_1-p_2\|_{\mathfrak{g}(x)^{-1}}\}/h_1(\|p\|_{\mathfrak{g}(x)^{-1}},r,\alpha)$$
$$< (1/2)\|z_1-z_2\|_z,$$

which proves that $\underline{\mathfrak{g}}_{h,z}$ is a contraction on B.

_Conclusion._ We obtain the result of Lemma 23 by applying the fixed point theorem on $\underline{\mathfrak{g}}_{h,z}$ and B. $\qquad\square$

Elaborating on Lemma 23, we prove in Lemma 24 that the only element of $\underline{G}_h(z^{(0)})$ verifies smoothness properties if $h$ is chosen small enough, depending on $z^{(0)}$.

**Lemma 24.** _For any_ $z = (x,p) \in T^\star M$, _any_ $r \in (0,1)$ _and any_ $\alpha \geq 1$, _we define_

$$\bar{h}(z,r,\alpha) = \min\left(h_1(\|p\|_{\mathfrak{g}(x)^{-1}},r,\alpha), h_2(z), h_3(z,r)\right) \tag{50}$$

_where_ $h_1$ _is defined in_ (49), _and_ $h_2$ _and_ $h_3$ _are defined by_

$$h_2(z) = \frac{1}{3/2 + 3\|p\|_{\mathfrak{g}(x)^{-1}}} , \qquad h_3(z,r) = \frac{1}{1 + (1-r)^{-1}(r + \|p\|_{\mathfrak{g}(x)^{-1}})} .$$

_We recall that_ $r_\alpha^\star$ _is defined in_ (33). _We also recall that the maps_ $\underline{G}_h$, $\underline{\mathfrak{g}}_h$, $\overline{G}_h$ _and_ $G_{h,x^{(0)}}$ _are respectively defined in_ (29), (30), (31) _and_ (32). _Assume A1, A2. Then, for any_ $z^{(0)} \in T^\star M$ _and any_ $h \in (0,\bar{h}(z^{(0)},r_\alpha^\star,\alpha))$, _there exists a unique element_ $z_h^{(1/2)} \in T^\star M$ _such that_ $z_h^{(1/2)} \in \underline{G}_h(z^{(0)}) \cap B_{\|\cdot\|_{z^{(0)}}}(z^{(0)},r_\alpha^\star)$. _Moreover, we have_

(a) $\mathrm{Jac}_{z'}[\underline{\mathfrak{g}}_h](z^{(0)}, z_h^{(1/2)})$ _and_ $\mathrm{Jac}[\overline{G}_h](z_h^{(1/2)})$ _are invertible,_

(b) $\mathrm{Jac}[G_{h,x^{(0)}}](x_h^{(1/2)})$ _is invertible._

_Proof._ Assume A1, A2. Let $z^{(0)} \in T^\star M$ and $h \in (0,\bar{h}(z^{(0)},r_\alpha^\star,\alpha))$, where $\bar{h}$ is defined in (50). Since $r_\alpha^\star \leq 1/11$, Lemma 23 ensures the existence and uniqueness of $z_h^{(1/2)} = (x_h^{(1/2)}, p_h^{(1/2)}) \in T^\star M$ such that $z_h^{(1/2)} \in \underline{G}_h(z^{(0)}) \cap B_{\|\cdot\|_{z^{(0)}}}(z^{(0)},r_\alpha^\star)$. Moreover, we have $x_h^{(1/2)} \in W^0(x^{(0)},r_\alpha^\star)$, and thus $\mathrm{Jac}[G_{h,x^{(0)}}](x_h^{(1/2)})$ is invertible by Lemma 22. Let us prove that $\mathrm{Jac}_{z'}[\underline{\mathfrak{g}}_h](z^{(0)}, z_h^{(1/2)})$ and $\mathrm{Jac}[\overline{G}_h](z_h^{(1/2)})$ are invertible.

**Invertibility of** $\mathrm{Jac}_{z'}[\underline{\mathfrak{g}}_h](z^{(0)}, z_h^{(1/2)})$. We first remark that $\mathrm{Jac}_{z'}[\underline{\mathfrak{g}}_h](z^{(0)}, z_h^{(1/2)}) = \mathrm{Jac}[\psi_{0,h}](z_h^{(1/2)})$ where $\psi_{0,h} = \mathrm{Id} + \frac{h}{2}\psi_0$ and $\psi_0$ is defined on $T^\star M$ by

$$\psi_0(z) = (-\{\mathfrak{g}(x^{(0)})^{-1} + \mathfrak{g}(y)^{-1}\}^{-1}p, -\tfrac{1}{2}D\mathfrak{g}(x^{(0)})[\mathfrak{g}(x^{(0)})^{-1}p,\mathfrak{g}(x^{(0)})^{-1}p]) .$$

We recall that $\|z_h^{(1/2)} - z^{(0)}\|_{z^{(0)}} < r_\alpha^\star \leq 1/11$ and thus define $r_1 = 1/11 - \|z_h^{(1/2)} - z^{(0)}\|_{z^{(0)}} > 0$ and $B_1 = B_{\|\cdot\|_{z^{(0)}}}(z_h^{(1/2)}, r_1)$. We set $\tilde{r} = 1/11$. Note that $B_1 \subset B_{\|\cdot\|_{z^{(0)}}}(z^{(0)}, 1/11)$. We show

below that $\psi_0$ is Lipschitz-continuous on $B_1$ with respect to $\|\cdot\|_{z^{(0)}}$. Let $(z, z') \in B_1 \times B_1$. Similarly to the proof of Lemma 23, we have

$$\|\psi_0^{(1)}(z) - \psi_0^{(1)}(z')\|_{\mathfrak{g}(x^{(0)})} \leq \|p - p'\|_{\mathfrak{g}(x^{(0)})^{-1}}$$
$$+ \|\mathfrak{g}(x^{(0)})^{1/2}\mathfrak{g}(x)^{-1}\mathfrak{g}(x^{(0)})^{1/2}\|_2 \|p - p'\|_{\mathfrak{g}(x^{(0)})^{-1}}$$
$$+ \|\mathfrak{g}(x^{(0)})^{1/2}\{\mathfrak{g}(x)^{-1} - \mathfrak{g}(x')^{-1}\}\mathfrak{g}(x^{(0)})^{1/2}\|_2 \|p\|_{\mathfrak{g}(x^{(0)})^{-1}}$$
$$\leq (1 + (1 - \tilde{r})^{-2})\|p - p'\|_{\mathfrak{g}(x^{(0)})^{-1}}$$
$$+ 3(1 - \tilde{r})^{-3}(\tilde{r} + \|p^{(0)}\|_{\mathfrak{g}(x^{(0)})^{-1}})\|x - x'\|_{\mathfrak{g}(x^{(0)})} .$$

In the same manner, we have

$$\|\psi_0^{(2)}(z) - \psi_0^{(2)}(z')\|_{\mathfrak{g}(x^{(0)})^{-1}} \leq 2(2\tilde{r} + \|p^{(0)}\|_{\mathfrak{g}(x^{(0)})^{-1}})\|p - p'\|_{\mathfrak{g}(x^{(0)})^{-1}} .$$

Therefore, recalling that $\tilde{r} = 1/11$, we obtain with the previous inequalities

$$\|\psi_0(z) - \psi_0(z')\|_{z^{(0)}} \leq \{3 + 6\|p^{(0)}\|_{\mathfrak{g}(x^{(0)})^{-1}}\}\|z - z'\|_{z^{(0)}}$$
$$\leq 2\|z - z'\|_{z^{(0)}}/h_2(z^{(0)}) .$$

Hence, $\psi_0$ is $(2/h_2(z^{(0)}))$-Lipschitz-continuous on $B_1$ with respect to $\|\cdot\|_{z^{(0)}}$. Then, since $h < h_2(z^{(0)})$, $\mathrm{Jac}[\psi_{0,h}](z)$ is invertible for any $z \in B_1$ by Lemma 21. In particular, $\mathrm{Jac}_{z'}[\underline{\mathrm{g}}_h](z^{(0)}, z_h^{(1/2)}) = \mathrm{Jac}[\psi_{0,h}](z_h^{(1/2)})$ is invertible.

**Invertibility of $\mathrm{Jac}[\overline{\mathrm{G}}_h](z_h^{(1/2)})$.** Let us remark that $\overline{\mathrm{G}}_h = \mathrm{Id} + h\zeta$ where $\zeta$ is defined on $\mathrm{T}^\star\mathrm{M}$ by

$$\zeta(z) = (0, \frac{1}{4}D\mathfrak{g}(x)[\mathfrak{g}(x)^{-1}p, \mathfrak{g}(x)^{-1}p]).$$

We define $r_2 = 1/2$ and $B_2 = B_{\|\cdot\|_{z_h^{(1/2)}}}(z_h^{(1/2)}, r_2)$. We show below that $\zeta$ is Lipschitz-continuous on $B_2$ with respect to $\|\cdot\|_{z_h^{(1/2)}}$, with a Lipschitz constant which does not depend on $z_h^{(1/2)}$. Let $(z, z') \in B_2 \times B_2$. Using Lemma 9-(a) with $r_2 = 1/2$, it is clear that

$$\|\zeta(z) - \zeta(z')\|_{z_h^{(1/2)}} \leq (1 + \|p_h^{(1/2)}\|_{\mathfrak{g}(x_h^{(1/2)})^{-1}})\|z - z'\|_{z_h^{(1/2)}} .$$

Moreover, since $\|z_h^{(1/2)} - z^{(0)}\|_{z^{(0)}} < r_\alpha^\star$, we have by Lemma 12

$$\|p_h^{(1/2)}\|_{\mathfrak{g}(x_h^{(1/2)})^{-1}} \leq (1 - r_\alpha^\star)^{-1}\|p_h^{(1/2)}\|_{\mathfrak{g}(x^{(0)})^{-1}}$$
$$\leq (1 - r_\alpha^\star)^{-1}(r_\alpha^\star + \|p^{(0)}\|_{\mathfrak{g}(x^{(0)})^{-1}}) ,$$

and thus

$$\|\zeta(z) - \zeta(z')\|_{z_h^{(1/2)}} \leq \{1 + (1 - r_\alpha^\star)^{-1}(r_\alpha^\star + \|p^{(0)}\|_{\mathfrak{g}(x^{(0)})^{-1}})\}\|z - z'\|_{z_h^{(1/2)}}$$
$$\leq \|z - z'\|_{z_h^{(1/2)}}/h_3(z^{(0)}, r_\alpha^\star) .$$

Hence, $\zeta$ is $(1/h_3(z^{(0)}, r_\alpha^\star))$-Lipschitz-continuous on $B_2$ with respect to $\|\cdot\|_{z_h^{(1/2)}}$. Since $h < h_3(z^{(0)}, r_\alpha^\star)$, $\mathrm{Jac}[\overline{\mathrm{G}}_h](z)$ is invertible for any $z \in B_2$ by Lemma 21. In particular, $\mathrm{Jac}[\overline{\mathrm{G}}_h](z_h^{(1/2)})$ is invertible, which concludes the proof. $\square$

The following lemma states the existence of a diffeomorphism between $z^{(0)} \in \mathrm{T}^\star\mathrm{M}$ and $z^{(1)} \in \mathrm{G}_h(z^{(0)})$ under smoothness conditions verified by $z^{(1)}$.

**Lemma 25.** *Let $(z^{(0)}, z^{(1)}) \in \mathrm{T}^\star\mathrm{M} \times \mathrm{T}^\star\mathrm{M}$. We recall that the maps $\underline{\mathrm{g}}_h$, $\overline{\mathrm{G}}_h$ and $\mathrm{G}_h$ are respectively defined in* (30), (31) *and* (6). *Assume that $\mathfrak{g} \in \mathrm{C}^2(\mathrm{M}, \mathbb{R}^{d \times d})$ and that there exist $h > 0$ and $z_h^{(1/2)} \in \mathrm{T}^\star\mathrm{M}$ with the following properties:*

*(a) $\underline{\mathrm{g}}_h(z^{(0)}, z_h^{(1/2)}) = 0$ and $z^{(1)} = \overline{\mathrm{G}}_h(z_h^{(1/2)})$.*

*(b)* $\mathrm{Jac}_{z'}[\underline{g}_h](z^{(0)}, z_h^{(1/2)})$ *and* $\mathrm{Jac}[\overline{G}_h](z_h^{(1/2)})$ *are invertible.*

*Then, there exists a neighbourhood* $\mathsf{U} \subset \mathrm{T}^\star\mathsf{M}$ *of* $z^{(0)}$ *and a* $\mathrm{C}^1$*-diffeomorphism* $\xi : \mathsf{U} \to \xi(\mathsf{U}) \subset \mathrm{T}^\star\mathsf{M}$ *such that (i)* $\xi(z^{(0)}) = z^{(1)}$*, (ii) for any* $z \in \mathsf{U}$*,* $\xi(z) \in \mathrm{G}_h(z)$ *and (iii)* $|\det\mathrm{Jac}\,\xi| \equiv 1$*.*

*Proof.* Let $(z^{(0)}, z^{(1)}) \in \mathrm{T}^\star\mathsf{M} \times \mathrm{T}^\star\mathsf{M}$. Assume that $\mathfrak{g} \in \mathrm{C}(\mathsf{M}, \mathbb{R}^{d \times d})$ and consider $h > 0$ and $z_h^{(1/2)} \in \mathrm{T}^\star\mathsf{M}$ as described in Lemma 25. Under the assumption on $\mathfrak{g}$, $\underline{g}_h$ and $\overline{G}_h$ are continuously differentiable respectively on $\mathrm{T}^\star\mathsf{M} \times \mathrm{T}^\star\mathsf{M}$ and $\mathrm{T}^\star\mathsf{M}$. We are going to prove Lemma 25, by first deriving intermediary results on $\underline{g}_h$ and $\overline{G}_h$.

**Result on $\underline{g}_h$.** We recall that $\underline{g}_h(z^{(0)}, z_h^{(1/2)}) = 0$ and $\mathrm{Jac}_{z'}[\underline{g}_h](z^{(0)}, z_h^{(1/2)})$ is invertible. Then, by applying the implicit function theorem on $\underline{g}_h$ at $(z^{(0)}, z_h^{(1/2)})$, we obtain the existence of a neighbourhood $\mathsf{U}_0 \subset \mathrm{T}^\star\mathsf{M}$ of $z^{(0)}$ and a $\mathrm{C}^1$-diffeomorphism $\xi_0 : \mathsf{U}_0 \to \xi_0(\mathsf{U}_0) \subset \mathrm{T}^\star\mathsf{M}$ such that $\xi_0(z^{(0)}) = z_h^{(1/2)}$ and for any $z \in \mathsf{U}_0$, $\underline{g}_h(z, \xi_0(z)) = 0$, *i.e.*, $\xi_0(z) \in \underline{G}_h(z)$. Moreover, for any $z \in \mathsf{U}_0$, the Jacobian of $\xi_0$ at $z$ is given by

$$\mathrm{Jac}[\xi_0](z) = -\mathrm{Jac}_{z'}[\underline{g}_h](z, \xi_0(z))^{-1}\,\mathrm{Jac}_z[\underline{g}_h](z, \xi_0(z))\,.$$

**Result on $\overline{G}_h$.** We recall that $z^{(1)} = \overline{G}_h(z_h^{(1/2)})$ and $\mathrm{Jac}[\overline{G}_h](z_h^{(1/2)})$ is invertible. Then, we apply the inverse function theorem on $\overline{G}_h$ at $z_h^{(1/2)}$ and obtain the existence of a neighbourhood $\mathsf{U}_1 \subset \mathrm{T}^\star\mathsf{M}$ of $z_h^{(1/2)}$ such that $\xi_1 = \overline{G}_h|_{\mathsf{U}_1}$ is a $\mathrm{C}^1$-diffeomorphism on $\mathsf{U}_1$.

**Final result.** We now consider the subset $\mathsf{U}$ and the map $\xi$, respectively defined by

(a) $\mathsf{U} = \xi_0^{-1}\left(\mathsf{U}_1 \cap \xi_0(\mathsf{U}_0)\right) \subset \mathrm{T}^\star\mathsf{M}$, neighbourhood of $z^{(0)}$.

(b) $\xi = \xi_1|_{\xi_0(\mathsf{U})} \circ \xi_0$, $\mathrm{C}^1$-diffeomorphism on $\mathsf{U}$ such that $\xi(z^{(0)}) = z^{(1)}$, and for any $z \in \mathsf{U}$, $\xi(z) \in \mathrm{G}_h(z)$.

Let us now prove that $|\det\mathrm{Jac}\,\xi| \equiv 1$. Let $z \in \mathsf{U}$. We define $z_0 = \xi_0(z)$. By the chain rule, we have

$$\begin{aligned}
\mathrm{Jac}[\xi](z) &= \mathrm{Jac}[\xi_1 \circ \xi_0](z) \\
&= \mathrm{Jac}[\xi_1](\xi_0(z))\,\mathrm{Jac}[\xi_0](z) \\
&= -\mathrm{Jac}[\xi_1](\xi_0(z))\,\mathrm{Jac}_{z'}[\underline{g}_h](z, \xi_0(z))^{-1}\,\mathrm{Jac}_z[\underline{g}_h](z, \xi_0(z))\,,
\end{aligned}$$

where we have

(a) $\mathrm{Jac}[\xi_1](z') = \begin{pmatrix} \mathrm{I}_d & 0 \\ -\frac{h}{2}\partial_{x,x}H_2(z') & \mathrm{I}_d - \frac{h}{2}\partial_{x,p}H_2(z') \end{pmatrix}$,

(b) $\mathrm{Jac}_{z'}[\underline{g}_h](z, z') = \begin{pmatrix} \mathrm{I}_d - \frac{h}{2}\partial_{x,p}H_2(z') & -\frac{h}{2}\{\partial_{p,p}H_2(x, p') + \partial_{p,p}H_2(x', p')\} \\ 0 & \mathrm{I}_d + \frac{h}{2}\partial_{x,p}H_2(x, p') \end{pmatrix}$,

(c) $\mathrm{Jac}_z[\underline{g}_h](z, z') = \begin{pmatrix} -\mathrm{I}_d - \frac{h}{2}\partial_{x,p}H_2(x, p') & 0 \\ \frac{h}{2}\partial_{x,x}H_2(x, p') & -\mathrm{I}_d \end{pmatrix}$.

Therefore, we obtain

$|\det\mathrm{Jac}[\xi](z)|$
$$= \det(\mathrm{I}_d - \frac{h}{2}\partial_{x,p}H_2(z_0))\det(\{\mathrm{I}_d - \frac{h}{2}\partial_{x,p}H_2(z_0)\}\{\mathrm{I}_d + \frac{h}{2}\partial_{x,p}H_2(x, p_0)\})^{-1}\det(\mathrm{I}_d + \frac{h}{2}\partial_{x,p}H_2(x, p_0)) = 1\,,$$

which concludes the proof. $\qquad\square$

**Lemma 26.** *Let* $h > 0$*. We recall that the set-valued map* $\mathrm{F}_h$ *is defined in Section 3.1. Let* $z \in \mathrm{T}^\star\mathsf{M}$*. Assume that there exist* $(z', z'') \in \mathrm{T}^\star\mathsf{M} \times \mathrm{T}^\star\mathsf{M}$ *such that*

*(a)* $z' \in F_h(z)$,

*(b)* $z'' \in F_h(z)$, *and*

*(c)* $x' = x''$.

*Then, we have $p' = p''$, and thus $z' = z''$.*

*Proof.* Let $h > 0$. Let $z \in T^\star M$. We consider $(z', z'') \in F_h(z) \times F_h(z)$ such that $x' = x'' = \tilde{x}$. By using the last two aligns from (6) with $z'$ and $z''$, we have $p' = -p^{(1/2)} + \frac{h}{2} \mathfrak{g}(\tilde{x})^{-1} p^{(1/2)} = p''$, where $p^{(1/2)} = \frac{2}{h} (\mathfrak{g}(x)^{-1} + \mathfrak{g}(\tilde{x})^{-1})^{-1} (\tilde{x} - x)$, which concludes the proof. $\square$

Under the assumption **A2**, we define, for any $z^{(0)} \in T^\star M$, $h^\star(z^{(0)}) = \bar{h}(s(z^{(0)}), r_\alpha^\star, \alpha)$, where $\alpha$ is given by **A2**, and $r_\alpha^\star$ and $\bar{h}$ are respectively defined in (33) and (50). Note that $h^\star$ appears in Proposition 2, for which we derive the proof below.

*Proof of Proposition 2.* We recall that maps $\underline{G}_h$, $\underline{g}_h$ and $\overline{G}_h$ are respectively defined in (29), (30) and (31). Assume **A1**, **A2**. Let $z^{(0)} \in T^\star M$. We define $\tilde{z}^{(0)} = s(z^{(0)})$. Let $h \in (0, h^\star(z^{(0)}))$. By using Lemma 24 on $\tilde{z}^{(0)}$, we obtain the existence of $z_h^{(1/2)} \in T^\star M$ with the following properties:

(a) $z_h^{(1/2)} \in \underline{G}_h(\tilde{z}^{(0)})$, *i.e.*, $\underline{g}_h(\tilde{z}^{(0)}, z_h^{(1/2)}) = 0$,

(b) $\mathrm{Jac}_{z'}[\underline{g}_h](\tilde{z}^{(0)}, z_h^{(1/2)})$ and $\mathrm{Jac}[\overline{G}_h](z_h^{(1/2)})$ are invertible,

(c) $\mathrm{Jac}[G_{h,x^{(0)}}](x_h^{(1/2)})$ is invertible.

We then define $z_h^{(1)} = \overline{G}_h(z_h^{(1/2)})$. In particular, $z_h^{(1)} \in G_h(\tilde{z}^{(0)})$, *i.e.*, $z_h^{(1)} \in F_h(z^{(0)})$ and $\mathrm{Jac}[G_{h,x^{(0)}}](x_h^{(1)}) = \mathrm{Jac}[G_{h,x^{(0)}}](x_h^{(1/2)})$ is invertible. By combining the properties of $z_h^{(1/2)}$ and with Lemma 25, we also obtain the existence of a neighbourhood $U' \subset T^\star M$ of $\tilde{z}^{(0)}$ and a $C^1$-diffeomorphism $\xi_h : U' \to \xi_h(U') \subset T^\star M$ such that (i) $\xi_h(\tilde{z}^{(0)}) = z_h^{(1)}$, (ii) for any $z \in U'$, $\xi_h(z) \in G_h(z)$ and (iii) $|\det \mathrm{Jac}\, \xi_h| \equiv 1$. Therefore, we can define the subset $U$ and the map $\gamma_h$ of Proposition 2 by

(a) $U = s(U')$, neighbourhood of $z^{(0)}$ in $T^\star M$,

(b) $\gamma_h = \xi_h \circ s$, such that (i) $\gamma_h(z^{(0)}) = z_h^{(1)}$, (ii) for any $z \in U$, $\gamma_h(z) \in F_h(z)$ and (iii) $\det \mathrm{Jac}[\gamma_h] \equiv 1$.

We now prove that $U$ can be reduced to a smaller subset such that $\gamma_h(z)$ is the only element of $F_h(z)$ in $\gamma_h(U)$ for any $z \in U$. Motivated by Lemma 22, we first define, for any $(x, x') \in M \times M$, $F_{h,x}(x')$ as the only element $p \in T_x^\star M$ such that $(x', p') \in F_h(x, p)$ for any $p' \in T_{x'}^\star M$. It is clear that $F_{h,x}(x') = -G_{h,x}(x')$, where $G_{h,x}$ is defined in (32). We also define $Z_h : M \times M \to T^\star M$ by

$$Z_h(x, x') = (x, F_{h,x}(x')) = (x, -G_{h,x}(x')) \,,$$

which is continuously differentiable since $\mathfrak{g} \in C^2(M, \mathbb{R}^{d \times d})$. Besides this, we obtain by Lemma 22 that $\mathrm{Jac}[G_{h,x^{(0)}}](x^{(0)}, x_h^{(1)})$ is invertible, and then $\mathrm{Jac}[Z_h](x_h^{(1)})$ is invertible. Therefore, by applying the inverse function theorem on $Z_h$ at $(x^{(0)}, x_h^{(1)})$, it comes that $Z_h$ is a $C^1$-diffeomorphism in a neighbourhood of $(x^{(0)}, x_h^{(1)})$. We are now going to prove the result by contradiction. Assume for now that there is no neighbourhood $U$ of $z^{(0)}$ such that $\gamma_h(z)$ is the only element of $F_h(z)$ in $\gamma_h(U)$, for any $z \in U$. Then, we can find a sequence $(z_i)_{i \in \mathbb{N}}$ which converges to $z^{(0)}$ such that for any $i \in \mathbb{N}$, there exist two different elements $z_{i,1} \in F_h(z_i)$ and $z_{i,2} \in F_h(z_i)$. Therefore, for any $i \in \mathbb{N}$, $x_{i,1} \neq x_{i,2}$ and by Lemma 26, we have

$$F_{h,x_i}(x_{i,1}) = F_{h,x_i}(x_{i,2}) = p_i \,,$$

and thus

$$Z_h(x_i, x_{i,1}) = Z_h(x_i, x_{i,2}) = z_i \ . \tag{51}$$

Moreover, by continuity of $\gamma_h$, the sequences $(z_{i,1})_{i \in \mathbb{N}}$ and $(z_{i,2})_{i \in \mathbb{N}}$ converge to $z_h^{(1)}$, and therefore, $(x_{i,1})_{i \in \mathbb{N}}$ and $(x_{i,2})_{i \in \mathbb{N}}$ also converge to $x^{(1)}$. Combined with (51), this result of convergence is in contradiction with the fact that $Z_h$ is a diffeomorphism in a neighbourhood of $(x^{(0)}, x_h^{(1)})$. Therefore, we can reduce $\mathsf{U}$ to a smaller subset such that $\gamma_h(z)$ is the only element of $\mathrm{F}_h(z)$ in $\gamma_h(\mathsf{U})$ for any $z \in \mathsf{U}$, which concludes the proof of Proposition 2. $\qquad\square$

# H  Modification of n-BHMC algorithm with step-size conditioning

In the rest of the paper, for any $z^{(0)} \in \mathrm{T}^\star\mathsf{M}$, we will denote by $h_\star(z^{(0)})$ the value of $h_\star$ given by **A**3.

**Beyond n-BHMC.**   A crucial part of the proof of reversibility in BHMC, as much for c-BHMC as for n-BHMC, relies on local *symplectic* properties of the integrator of the Hamiltonian dynamics. Although Algorithm 1 can be implemented without any practical limitation, it is hard to state such properties for its numerical integrator $\Phi_h$ under **A**1, **A**2 and **A**3, given *any* value of $h$. Indeed, we know from **A**3 that $\Phi_h$ is a local involution around $z^{(0)} \in \mathrm{T}^\star\mathsf{M}$ when $h < h_\star(z^{(0)})$; however, we cannot ensure this result when $h > h_\star(z^{(0)})$... To circumvent this issue, we propose to study Theoretical n-BHMC (Tn-BHMC), presented in Algorithm 3. In this modified version of n-BHMC, we actually enforce a condition on $h$ to be small enough. We now get into the details of this new algorithm and assume **A**3 for the rest of this section.

**Theoretical motivations.**   Let $h > 0$. We recall the definition of the set $\mathsf{A}_h$ introduced in Section 4.2

$$\mathsf{A}_h = \{ z \in \mathrm{T}^\star\mathsf{M} \ : \ h < \min_{\tilde{z} \in \mathsf{B}_{\|\cdot\|_z}(z,1)} h_\star(\tilde{z}) \} \ .$$

It is clear that $\mathsf{A}_h \subset \mathrm{dom}_{\Phi_h}$: indeed, if $z^{(0)} \in \mathsf{A}_h$, we have in particular that $h < h_\star(z^{(0)})$, and therefore $z^{(0)} \in \mathrm{dom}_{\Phi_h}$ by **A**3. Let $z^{(0)} \in \mathsf{A}_h$. By **A**3, we know that $\Phi_h$ is an involution on a neighbourhood of $z^{(0)}$; in particular, it comes that $(\Phi_h \circ \Phi_h)(z^{(0)}) = z^{(0)}$. Hence, the condition (b) of the "involution checking step" in Algorithm 1 is *de facto* satisfied. This naturally leads to replace the condition "$z \in \mathrm{dom}_{\Phi_h}$" by the more restrictive condition "$z \in \mathsf{A}_h$".

---

**Algorithm 3:** Tn-BHMC with Momentum Refreshment

**Input:** $(x_0, p_0) \in \mathrm{T}^\star\mathsf{M}$, $\beta \in (0,1]$, $N \in \mathbb{N}$, $h > 0$, $\eta > 0$, $\Phi_h$ with domain $\mathrm{dom}_{\Phi_h}$, $\mathsf{A}_h$
**Output:** $(X_n, P_n)_{n \in [N]}$

1  **for** $n = 1, ..., N$ **do**
2      Step 1:  $G_n \sim \mathrm{N}_x(0, \mathrm{I}_d), \tilde{P}_n \leftarrow \sqrt{1-\beta}P_{n-1} + \sqrt{\beta}G_n$
3      Step 2:  solving a discretized version of ODE (3)
4      $X'_n, P'_n \leftarrow X_{n-1}, \tilde{P}_n; \quad X_n^{(0)}, P_n^{(0)} \leftarrow (s \circ \mathrm{S}_{h/2})(X_n, \tilde{P}_n)$
5      **if** $Z_n^{(0)} = (X_n^{(0)}, P_n^{(0)}) \in \mathsf{A}_h$ **then**
6          $Z_n^{(1)} = \Phi_h(Z_n^{(0)})$
7          **if** $Z_n^{(1)} \in \mathsf{A}_h$ **then**
8              $X'_n, P'_n \leftarrow (s \circ \mathrm{S}_{h/2})(Z_n^{(1)})$
9          **end**
10     **end**
11     Step 3:  $A_n \leftarrow \min(1, \exp[-H(X'_n, P'_n) + H(X_{n-1}, \tilde{P}_n)])$
12     $U_n \sim \mathrm{U}[0,1]$
13     **if** $U_n \leq A_n$ **then** $\bar{X}_n, \bar{P}_n \leftarrow X'_n, P'_n$
14     **else** $\bar{X}_n, \bar{P}_n \leftarrow X_{n-1}, \tilde{P}_n$
15     Step 4:  $X_n, \hat{P}_n \leftarrow s(\bar{X}_n, \bar{P}_n)$
16     Step 5:  $G'_n \sim \mathrm{N}_x(0, \mathrm{I}_d), P_n \leftarrow \sqrt{1-\beta}\hat{P}_n + \sqrt{\beta}G'_n$
17 **end**

---

**Implementation of Tn-BHMC.** In Algorithm 3, we highlight in yellow the modifications of Tn-BHMC in contrast to n-BHMC. Namely, we replace

(a) "$Z_n^{(0)} \in \mathrm{dom}_{\Phi_h}$" (Line 5 in Algorithm 1) by "$Z_n^{(0)} \in \mathsf{A}_h$" (Line 5 in Algorithm 3).

(b) "$Z_n^{(1)} \in \mathrm{dom}_{\Phi_h}$" (Line 8 in Algorithm 1) by "$Z_n^{(1)} \in \mathsf{A}_h$" (Line 7 in Algorithm 3).

Note that there is no need to maintain the "involution checking step" of Algorithm 1, since it is automatically verified once $Z_n^{(0)} \in \mathsf{A}_h$. On the whole, these new conditions are more *restrictive* than the conditions of Algorithm 1 since $\mathsf{A}_h \subset \mathrm{dom}_{\Phi_h}$; moreover, they can be thought as conditions directly applied on the step-size $h$. The specific choice of $\mathsf{A}_h$, instead of another subset of $\mathrm{dom}_{\Phi_h}$, is actually sufficient to derive the proof of reversibility of Tn-BHMC (see Section 4.2).

# I   Proofs of Section 4.2

## I.1   Expression and properties of $r^\star$

Given a Riemannian manifold $(\mathsf{M}, \mathfrak{g})$, we define $r^\star(x)$ for any $x \in \mathsf{M}$ by

$$r^\star(x) = \min(\|\mathfrak{g}(x)\|_2^{-1/2}, \|\mathfrak{g}(x)^{-1}\|_2^{-1/2}) . \tag{52}$$

Note that $r^\star$ is used in **A**3 and that $r^\star(x) = 1/C_x$, where $C_x$ is defined in Lemma 12. We prove below that $r^\star$ has a smooth behaviour on $\mathsf{M}$ under our main assumptions.

**Lemma 27.** *Assume **A**1, **A**2. Also assume that $x \in \mathsf{M} \mapsto \|\mathfrak{g}^{-1}(x)\|_2$ is bounded from above. As defined in (52), $r^\star : \mathsf{M} \to (0, +\infty)$ satisfies the following properties:*

*(a) $r^\star(x) \to 0$ as $x \to \partial\mathsf{M}$.*

*(b) There exists $\mathsf{L} > 0$ such that $r^\star$ is $\mathsf{L}$-Lipschitz-continuous on $\mathsf{M}$ with respect to $\| \cdot \|_2$.*

*(c) There exists $\mathsf{M} > 0$ such that $r^\star(x) \leq \mathsf{M}$ for any $x \in \mathsf{M}$.*

*Proof.* Assume **A**1, **A**2. Also assume that $x \in \mathsf{M} \mapsto \|\mathfrak{g}^{-1}(x)\|_2$ is bounded from above. We first define $r_1 : x \in \mathsf{M} \mapsto \|\mathfrak{g}(x)\|_2^{-1/2}$ and $r_2 : x \in \mathsf{M} \mapsto \|\mathfrak{g}(x)^{-1}\|_2^{-1/2}$, such that $r^\star = \min(r_1, r_2)$. Since $\mathfrak{g} \in \mathrm{C}^2(\mathsf{M}, \mathbb{R}^{d \times d})$, it is clear that $r_1$ and $r_2$ are continuously differentiable on $\mathsf{M}$. We have:

(a) $r_1(x) \to 0$ as $x \to \partial\mathsf{M}$ by Lemma 8, and

(b) $r_2(x) \nrightarrow 0$ as $x \to \partial\mathsf{M}$, since $1/r_2^2 : x \mapsto \|\mathfrak{g}(x)^{-1}\|_2$ is bounded on $\mathsf{M}$.

Combining the fact that $r_2(x) > 0$ for any $x \in \mathsf{M}$ and item (a), we obtain item (a) of Lemma 27. We denote by $d$ the distance induced by $\| \cdot \|_2$ and now define, for any $\varepsilon > 0$,

(i) $\mu_\varepsilon = \inf_{\{y \in \mathsf{M} : d(y, \partial\mathsf{M}) \leq \epsilon\}} r_2(y)$.

(ii) $\mathsf{M}_\varepsilon = \mathrm{Int}(\{x \in \mathsf{M} : d(x, \partial\mathsf{M}) \leq \varepsilon, r_1(x) \leq \mu_\varepsilon\})$, open set in $\mathsf{M}$.

(iii) $\mathsf{M}_{-\varepsilon} = \mathsf{M} \backslash \mathsf{M}_\varepsilon$, closed and bounded (and thus, compact) set in $\mathsf{M}$.

Using items (a) and (b), we can ensure the existence of some $\varepsilon \in (0, \mathrm{diam}(\mathsf{M}))$ such that (i) $\mu_\varepsilon > 0$ and (ii) $\mathsf{M}_\varepsilon$ and $\mathsf{M}_{-\varepsilon}$ are not empty. We consider such $\varepsilon$ for the rest of the proof.

**Smoothness of $r_2$.** We define $\delta = d(\mathsf{M}^c, \mathsf{M}_{-\varepsilon})$ and $\mathsf{M}_\delta = \mathsf{M}^c + \bar{\mathsf{B}}(0, \delta/4)$. Note that

(i) $\delta > 0$ since $\mathsf{M}_{-\varepsilon} \subset \mathsf{M}$,

(ii) $\mathsf{M}_\delta$ is closed since the ball $\bar{\mathsf{B}}(0, \delta/4)$ is compact, and

(iii) $\mathsf{M}_\delta \cap \mathsf{M}_{-\varepsilon} = \emptyset$.

According to the smooth version Urysohn's lemma applied to $\mathsf{M}_\delta$ and $\mathsf{M}_{-\varepsilon}$, there exists $\chi \in \mathrm{C}^1(\mathbb{R}^d, [0,1])$ such that $\chi(\mathsf{M}_{-\varepsilon}) = 1$ and $\chi(\mathsf{M}_\delta) = 0$. We then define $\tilde{r}_2 : \mathbb{R}^d \to (0, +\infty)$ by $\tilde{r}_2 = \chi r_2 + (1 - \chi)\mu_\varepsilon$. In particular, (i) there exists $\mathrm{L}_2 > 0$ such that $\tilde{r}_2$ is $\mathrm{L}_2$-Lipschitz-continuous on $\bar{\mathsf{M}}$ with respect to $\|\cdot\|_2$, since $\tilde{r}_2 \in \mathrm{C}^1(\mathbb{R}^d, [0,1])$, and (ii) for any $x \in \mathsf{M}_\varepsilon$, $\tilde{r}_2(x) > \mu_\varepsilon$.

**Smoothness of $r_1$.** We now prove that $r_1$ is 1-Lipschitz continuous on $\mathsf{M}$ with respect to $\|\cdot\|_2$. Let $x \in \mathsf{M}$. Note that $r_1(x) = (\|\mathfrak{g}(x)\|_2^2)^{-1/4}$ and we thus have

$$\nabla r_1(x) = (-1/4)\|\mathfrak{g}(x)\|_2^{-5/2} h(x) : D\mathfrak{g}(x) ,$$

where $h(x) = \partial_{\mathfrak{g}(x)} \|\mathfrak{g}(x)\|_2^2 = 2\|\mathfrak{g}(x)\|_2 u(x)u(x)^\top$, $u(x)$ being a normal eigenvector of $\mathfrak{g}(x)$ corresponding to the eigenvalue $\|\mathfrak{g}(x)\|_2$. Hence,

$$
\begin{aligned}
\|\nabla r_1(x)\|_2 &\le (1/2)\|\mathfrak{g}(x)\|_2^{-3/2} \|u(x)u(x)^\top : D\mathfrak{g}(x)\|_2 \\
&\le (1/2)\|\mathfrak{g}(x)\|_2^{-3/2} \|u(x)u(x)^\top\|_2 \|D\mathfrak{g}(x)\|_2 \\
&\le (1/2)\|\mathfrak{g}(x)\|_2^{-3/2} \times 2\|\mathfrak{g}(x)\|_2^{3/2} \le 1 \qquad \text{(Definition 1-(c))}
\end{aligned}
$$

which proves the result on $r_1$.

**Smoothness of $r$.** Let $x \in \mathsf{M}$. We can face two cases: either, $x \in \mathsf{M}_{-\varepsilon}$, then $r_2(x) = \tilde{r}_2(x)$; or, $x \in \mathsf{M}_\varepsilon$, then $\tilde{r}_2(x) > \mu_\varepsilon \ge r_1(x)$ and $r_2(x) \ge r_1(x)$ by definition of $\mathsf{M}_\varepsilon$. Thus, we have for any $x \in \mathsf{M}$, $r^\star(x) = \min(r_1(x), \tilde{r}_2(x))$ where $r_1$ and $\tilde{r}_2$ are respectively 1 and $\mathrm{L}_2$ Lipschitz-continuous on $\mathsf{M}$ with respect to $\|\cdot\|_2$. By observing that $2\min(r_1, \tilde{r}_2) = r_1 + \tilde{r}_2 - |r_1 - \tilde{r}_2|$, we set $\mathrm{L} = 1 + \mathrm{L}_2$ and thus obtain item (b) of Lemma 27. Finally, item (c) of Lemma 27 directly derives from item (b), since $\mathsf{M}$ is bounded. $\qquad \square$

Note that the extra-assumption on $\mathfrak{g}^{-1}$ used in Lemma 27 is not directly ensured by self-concordance but can be proved when $\mathsf{M}$ is a polytope, as shown below.

**Lemma 28.** *Consider a polytope $\mathsf{M}$ defined by $m$ constraints with $m > d$ such that $\mathsf{M} = \{x : \mathrm{A}x < b\}$, where $\mathrm{A} \in \mathbb{R}^{m \times d}$ is a full-rank matrix and $b \in \mathbb{R}^m$. We endow $\mathsf{M}$ with the Riemannian metric $\mathfrak{g}(x) = D^2\phi(x)$ where $\phi : \mathsf{M} \to \mathbb{R}$ is the logarithmic barrier given for any $x \in \mathsf{M}$ by $\phi(x) = -\sum_{i=1}^m \ln\left(b_i - \mathrm{A}_i^\top x\right)$. In particular, $\mathsf{M}$ and $\mathfrak{g}$ verify A1 and A2. Then, the function $r : \mathsf{M} \to (0, +\infty)$ defined by $r(x) = \|\mathfrak{g}^{-1}(x)\|_2$, for any $x \in \mathsf{M}$, is bounded from above.*

*Proof.* Consider such manifold $\mathsf{M}$ and metric $\mathfrak{g}$. We aim to show that the smallest eigenvalue of $\mathfrak{g}(x)$ is bounded from below for any $x \in \mathsf{M}$ by a constant $c > 0$, which does not depend on $x$, *i.e.*, for any $h \in \mathbb{R}^d$ and any $x \in \mathsf{M}$, $\mathfrak{g}(x)[h, h] \ge c\|h\|_2^2$.

Since $\mathrm{A}$ is full-ranked, $\mathrm{A}^\top \mathrm{A}$ is positive-definite. In particular, for any $h \in \mathbb{R}^d$, $(\mathrm{A}^\top \mathrm{A})[h, h] \ge \lambda_{\min}(\mathrm{A}^\top \mathrm{A})\|h\|_2^2$, where $\lambda_{\min}(\mathrm{A}^\top \mathrm{A}) > 0$ is the smallest eigenvalue of $\mathrm{A}^\top \mathrm{A}$. We recall that we have for any $x \in \mathsf{M}$, $\mathfrak{g}(x) = \mathrm{A}^\top S(x)^{-2} \mathrm{A}$, where $S(x) = \mathrm{Diag}(b_i - \mathrm{A}_i^\top x)_{i \in [m]}$. Let $i \in [m]$. The function $r_i : x \in \mathsf{M} \mapsto S(x)_{i,i}^{-2}$ has the following properties: (i) $r_i$ is continuous on $\mathsf{M}$, (ii) $r_i(x) > 0$ for any $x \in \mathsf{M}$ and (iii) $r(x) \to +\infty$ as $x \to \partial\mathsf{M}$. Thus, there exists $c_i > 0$ such that for any $x \in \mathsf{M}$, $r_i(x) \ge c_i$. We define $\tilde{c} = \min_{i \in [m]} c_i$ and we have for any $x \in \mathsf{M}$, $S(x)^{-2} \succeq \tilde{c}\mathrm{I}_d$. We now define $c = \tilde{c}\lambda_{\min}(\mathrm{A}^\top \mathrm{A})$ and we have for any $x \in \mathsf{M}$

$$\mathfrak{g}(x)[h, h] \ge \tilde{c} \cdot (\mathrm{A}^\top \mathrm{A})[h, h] \ge c\|h\|_2^2 .$$

In particular, $\mathfrak{g}(x)^{-1} \preceq (1/c)\mathrm{I}_d$, *i.e.*, $\|\mathfrak{g}(x)^{-1}\|_2 \le (1/c)$, which concludes the proof. $\qquad \square$

## I.2 Markov kernels of Algorithm 1

Based on the model of $\bar{\mathsf{E}}_h$ defined in (24), we define the set $\bar{\mathsf{E}}_h^\Phi \subset \mathrm{T}^\star\mathsf{M}$, which will ensure that the maps derived from implicit integrators of n-BHMC are properly expressed

$$\bar{\mathsf{E}}_h^\Phi = (s \circ \mathrm{S}_{h/2})^{-1}(\mathrm{dom}_{\Phi_h} \cap \Phi_h^{-1}(\mathrm{dom}_{\Phi_h})) = (s \circ \mathrm{S}_{h/2})(\mathrm{dom}_{\Phi_h} \cap \Phi_h^{-1}(\mathrm{dom}_{\Phi_h})) . \qquad (53)$$

For any $(z, z') \in \bar{\mathsf{E}}_h^\Phi \times \mathrm{T}^\star\mathsf{M}$, we define the acceptance probability $\bar{a}(z'|z)$ to move from $z$ to $z'$ by

$$\bar{a}(z'|z) = a(z' \mid z)\mathbb{1}_{(s \circ \mathsf{S}_{h/2})(z) = (\Phi_h \circ \Phi_h \circ s \circ \mathsf{S}_{h/2})(z)} = a(z' \mid z)\mathbb{1}_{z = (\mathrm{R}_h^\Phi \circ \mathrm{R}_h^\Phi)(z)} \ ,$$

where $a(z' \mid z)$ is the acceptance probability defined in (9). We denote by $\mathrm{Q}_n : \mathrm{T}^\star\mathsf{M} \times \mathcal{B}(\mathrm{T}^\star\mathsf{M}) \to [0, 1]$, the transition kernel of the (homogeneous) Markov chain $(x_n, p_n)_{n \in [N]}$ generated by Algorithm 1. We also denote by:

(a) $\mathrm{Q}_0 : \mathrm{T}^\star\mathsf{M} \times \mathcal{B}(\mathrm{T}^\star\mathsf{M}) \to [0, 1]$, the transition kernel referring to Step 1 (also Step 5) in Algorithm 1, defined in (25).

(b) $\mathrm{Q}_{n,1} : \mathrm{T}^\star\mathsf{M} \times \mathcal{B}(\mathrm{T}^\star\mathsf{M}) \to [0, 1]$, the transition kernel referring to Step 2-3-4 in Algorithm 1.

We provide below details on Markov kernels $\mathrm{Q}_n$ and $\mathrm{Q}_{n,1}$.

**Kernel $\mathrm{Q}_{n,1}$.** This kernel is deterministic and corresponds to the *numerical* integration of the Hamiltonian up until time $h$. For any $(z, z') \in \mathrm{T}^\star\mathsf{M} \times \mathrm{T}^\star\mathsf{M}$, we have

$$\mathrm{Q}_{n,1}(z, \mathrm{d}z') = \mathbb{1}_{\bar{\mathsf{E}}_h^\Phi}(z)(s_\# \mathrm{Q}_{n,2})(z, \mathrm{d}z') + \mathbb{1}_{(\bar{\mathsf{E}}_h^\Phi)^c}(z)\delta_{s(z)}(\mathrm{d}z') \ , \tag{54}$$

where

$$\mathrm{Q}_{n,2}(z, \mathrm{d}z') = \bar{a}(\mathrm{R}_h^\Phi(z) \mid z)\delta_{\mathrm{R}_h^\Phi(z)}(\mathrm{d}z') + [1 - \bar{a}(\mathrm{R}_h^\Phi(z) \mid z)]\delta_z(\mathrm{d}z') \ . \tag{55}$$

**Kernel $\mathrm{Q}_n$.** This kernel corresponds to one step of Algorithm 1 (*i.e.*, comprising Steps 1 to 5). For any $(z, z') \in \mathrm{T}^\star\mathsf{M} \times \mathrm{T}^\star\mathsf{M}$, we have

$$\mathrm{Q}_n(z, \mathrm{d}z') = \int_{\mathrm{T}^\star\mathsf{M} \times \mathrm{T}^\star\mathsf{M}} \mathrm{Q}_0(z, \mathrm{d}z_1)\mathrm{Q}_{n,1}(z_1, \mathrm{d}z_2)\mathrm{Q}_0(z_2, \mathrm{d}z') \ .$$

### I.3 Proof of reversibility in Algorithm 3

Let $h > 0$. Using notation from **A**3, we recall definition of the set $\mathsf{A}_h$ introduced in Section 4.2

$$\mathsf{A}_h = \{z \in \mathrm{T}^\star\mathsf{M} \ : \ h < \min_{\tilde{z} \in \mathsf{B}_{\|\cdot\|_z}(z,1)} h_\star(\tilde{z})\} \subset \mathrm{dom}_{\Phi_h} \ . \tag{56}$$

We also recall that $\bar{\pi}$, as defined in (2), admits a density with respect to the product Lebesgue measure given for any $z = (x, p) \in \mathrm{T}^\star\mathsf{M}$ by

$$(\mathrm{d}\bar{\pi}/(\mathrm{d}x\mathrm{d}p))(x, p) = (1/Z)\exp[-(1/2)\|p\|_{\mathfrak{g}(x)^{-1}}^2]\det(\mathfrak{g}(x))^{-1/2}\exp[-V(x)] \ .$$

Since Algorithm 3 can be thought as a *restrictive* version of Algorithm 1, the Markov kernels from Tn-BHMC are similar to the kernels from n-BHMC, defined in Appendix I.2. In particular, the transition kernel corresponding to the Gaussian momentum update (Step 1 and 5 in Algorithm 3, Step 1 and 5 in Algorithm 1) is the same and is reversible (up to momentum reversal) with respect to $\bar{\pi}$ (see Lemma 17). Namely, we replace

(a) the set $\bar{\mathsf{E}}_h^\Phi$, defined in (53), by $\tilde{\mathsf{E}}_h^\Phi$,
(b) the kernels $\mathrm{Q}_{n,1}$ and $\mathrm{Q}_{n,2}$, respectively defined in (54) and (55), by $\mathrm{Q}_1$ and $\mathrm{Q}_2$,

where

$$\tilde{\mathsf{E}}_h^\Phi = (s \circ \mathsf{S}_{h/2})^{-1}(\mathsf{A}_h \cap \Phi_h^{-1}(\mathsf{A}_h)) = (s \circ \mathsf{S}_{h/2})(\mathsf{A}_h \cap \Phi_h^{-1}(\mathsf{A}_h)) \ , \tag{57}$$

$$\mathrm{Q}_1(z, \mathrm{d}z') = \mathbb{1}_{\tilde{\mathsf{E}}_h^\Phi}(z)(s_\# \mathrm{Q}_2)(z, \mathrm{d}z') + \mathbb{1}_{(\tilde{\mathsf{E}}_h^\Phi)^c}(z)\delta_{s(z)}(\mathrm{d}z') \ , \tag{58}$$

$$\mathrm{Q}_2(z, \mathrm{d}z') = a(\mathrm{R}_h^\Phi(z) \mid z)\delta_{\mathrm{R}_h^\Phi}(\mathrm{d}z') + [1 - a(\mathrm{R}_h^\Phi(z) \mid z)]\delta_z(\mathrm{d}z') \ . \tag{59}$$

We denote by $\mathrm{Q} : \mathrm{T}^\star\mathsf{M} \times \mathcal{B}(\mathrm{T}^\star\mathsf{M}) \to [0, 1]$, the transition kernel of the (homogeneous) Markov chain $(x_n, p_n)_{n \in [N]}$ generated by Algorithm 3. For any $(z, z') \in \mathrm{T}^\star\mathsf{M} \times \mathrm{T}^\star\mathsf{M}$, we have

$$\mathrm{Q}(z, \mathrm{d}z') = \int_{\mathrm{T}^\star\mathsf{M} \times \mathrm{T}^\star\mathsf{M}} \mathrm{Q}_0(z, \mathrm{d}z_1)\mathrm{Q}_1(z_1, \mathrm{d}z_2)\mathrm{Q}_0(z_2, \mathrm{d}z') \ . \tag{60}$$

We first turn to the reversibility up to momentum reversal of $\mathrm{Q}_1$ with respect to $\bar{\pi}$. We start with the following lemma which is key to establish this result.

**Lemma 29.** *Assume* **A**1, **A**2, **A**3. *Then, for any compact set* $G \subset T^*M$, *for any* $g \in C(T^*M \times T^*M, \mathbb{R})$ *such that* $\mathrm{supp}(g) \subset G \times G$, *we have*

$$\int_{G_h} g(z, \Phi_h(z)) \mathrm{d}z = \int_{G_h} g(\Phi_h(z), z) \mathrm{d}z \; ,$$

*where* $G_h = F_h \cap G \cap \Phi_h^{-1}(G)$, $F_h = A_h \cap \Phi_h^{-1}(A_h)$, $A_h$ *being defined in* (56).

*Proof.* Assume **A**1, **A**2, **A**3. Let $G \subset T^*M$ be a compact set, $g \in C(T^*M \times T^*M, \mathbb{R})$ such that $\mathrm{supp}(g) \subset G \times G$. We first define the sets $F_h = A_h \cap \Phi_h^{-1}(A_h)$, $G_h = F_h \cap G \cap \Phi_h^{-1}(G)$ and the integrals $I'$ and $J'$

$$I' = \int_{G_h} g(z, \Phi_h(z)) \mathrm{d}z \; , \quad J' = \int_{G_h} g(\Phi_h(z), z) \mathrm{d}z \; . \tag{61}$$

We recall the existence under **A**1 and **A**2 of $L > 0$ and $M > 0$ such that $r^*$, given by **A**3 and defined in (52), is L-Lipschitz-continuous on $M$ with respect to $\| \cdot \|_2$ and bounded from above by $M$ (see Lemma 27).

We define $r : T^*M \to (0, +\infty)$ by $r(x, p) = r^*(x)/m$ for any $(x, p) \in T^*M$, where $m = \max\{1/\lambda, 4M, 4L\}$ and $\lambda$ is given by **A**3. Using the properties of $r^*$, it comes that

(a) $r$ is L-Lipschitz on $T^*M$ with respect to $\| \cdot \|_2$,

(b) $r(x, p) \leq 1/(4LC_x)$ for any $(x, p) \in T^*M$, where $C_x$ is defined in Lemma 12,

(c) $r \leq 1/4$, and

(d) $r \leq \lambda r^*$.

Note that $G \subset \cup_{z \in G} B_{\|\cdot\|_z}(z, r(z))$. Since $G$ is a compact set, there exist $K \in \mathbb{N}$ and $(z_i)_{i \in [K]} \in T^*M^K$ such that $G \subset \bigcup_{i=1}^K B_i$, where $B_i = B_{\|\cdot\|_{z_i}}(z_i, r(z_i))$.

We consider the sequence $\{V_i\}_{i=1}^K$ constructed as follows: $V_1 = G \cap B_1$ and for any $i \in \{2, \ldots, K\}$, $V_i = (G \cap B_i) \cap (\cup_{j=1}^{i-1} V_j)^c$. Then, we have that, for any $i \in \{1, \ldots, N\}$, $\cup_{j=1}^i V_j = G \cap (\cup_{j=1}^i B_j)$, and that for any $i_1, i_2 \in \{1, \ldots, K\}$, $V_{i_1} \cap V_{i_2} = \emptyset$ if $i_1 \neq i_2$. Therefore, we get that $G = \sqcup_{i=1}^K V_i$ and $\Phi_h^{-1}(G) = \sqcup_{i=1}^K \Phi_h^{-1}(V_i)$. In particular, for any $A \in \mathcal{B}(T^*M)$ and any $\zeta \in C_c(T^*M, \mathbb{R})$

$$\int_{G \cap A} \zeta(z) \mathrm{d}z = \sum_{i=1}^K \int_{V_i \cap A} \zeta(z) \mathrm{d}z \; , \tag{62}$$

and for any $\tilde{A} \in \mathcal{B}(T^*M)$ and any $\tilde{\zeta} \in C_c(T^*M, \mathbb{R})$

$$\int_{\Phi_h^{-1}(G) \cap \tilde{A}} \tilde{\zeta}(z) \mathrm{d}z = \sum_{i=1}^K \int_{\Phi_h^{-1}(V_i) \cap \tilde{A}} \tilde{\zeta}(z) \mathrm{d}z \; . \tag{63}$$

Using (62) and (61), we obtain $I' = \sum_{i=1}^K I_i'$, where $I_i' = \int_{V_i \cap \Phi_h^{-1}(G) \cap F_h} g(z, \Phi_h(z)) \mathrm{d}z$ for any $i \in [K]$. We are now going to show that, for any $i \in [K]$

$$I_i' = \int_{\Phi_h^{-1}(V_i) \cap G \cap F_h} g(\Phi_h(z), z) \mathrm{d}z \; . \tag{64}$$

Let $i \in [K]$. We proceed by making the following case disjunction:

(a) Either $V_i \cap \Phi_h^{-1}(G) \cap F_h = \emptyset$, and then $I_i' = 0$. We prove by contradiction in this case that $\Phi_h^{-1}(V_i) \cap G \cap F_h = \emptyset$. Assume that there exists $z \in \Phi_h^{-1}(V_i) \cap G \cap F_h$. By definition of $F_h$, $z \in A_h$, $\Phi_h(z) \in A_h$, and we get that $\Phi_h(\Phi_h(z)) = z$ using **A**3. Hence, we have:

- $\Phi_h(z) \in V_i$,
- $\Phi_h(\Phi_h(z)) = z \in G$,
- $\Phi_h(z) \in A_h$,
- $\Phi_h(\Phi_h(z)) = z \in A_h$.

Therefore, we get $\Phi_h(z) \in V_i \cap \Phi_h^{-1}(G) \cap F_h = \emptyset$, which is absurd. Finally, (64) holds since

$$I_i' = 0 = \int_{\Phi_h^{-1}(V_i) \cap G \cap F_h} g(\Phi_h(z), z) \mathrm{d}z \; .$$

(b) Either, there exists some $\tilde{z} \in V_i \cap \Phi_h^{-1}(G)$ such that $\tilde{z} \in F_h$. In particular, $\tilde{z} \in B_i$, and thus $\|\tilde{z} - z_i\|_{z_i} < r(z_i) < 1$. In this case, by combining **A**3 with Lemma 9-(c), we have for any $z' \in T^\star M$

$$\|z'\|_{\tilde{z}} \leq (1 - \|\tilde{z} - z_i\|_{z_i})^{-1}\|z'\|_{z_i} < (1 - r(z_i))^{-1}\|z'\|_{z_i} \ .$$

By considering $z' = z_i - \tilde{z}$, it comes that

$$\|z_i - \tilde{z}\|_{\tilde{z}} < (1 - r(z_i))^{-1}\|\tilde{z} - z_i\|_{z_i} < (1 - r(z_i))^{-1}r(z_i) \ .$$

By combining properties (a) and (b) of $r$ with Lemma 12, we have the following upper bound of $r(z_i)$

$$r(z_i) \leq r(\tilde{z}) + \mathsf{L}\|\tilde{z} - z_i\|_2 \leq r(\tilde{z}) + \mathsf{L}C_{x_i}\|\tilde{z} - z_i\|_{z_i} < r(\tilde{z}) + \mathsf{L}C_{x_i}r(z_i) < r(\tilde{z}) + 1/4 \ ,$$

and thus

$$\|z_i - \tilde{z}\|_{\tilde{z}} < (1 - r(\tilde{z}) - 1/4)^{-1}(r(\tilde{z}) + 1/4) < 1 \ ,$$

using property (c) of $r$. Hence, $z_i \in \mathsf{B}_{\|\cdot\|_{\tilde{z}}}(\tilde{z}, 1)$. Moreover, $\tilde{z} \in F_h \subset A_h$, and then it comes that $h < h_\star(z_i)$. Therefore, by combining property (d) of $r$ with **A**3, we have

(i) $V_i \subset B_i \subset \mathsf{B}_{\|\cdot\|_{z_i}}(z_i, \lambda r^\star(z_i)) \subset \mathrm{dom}_{\Phi_h}$,

(ii) the restriction of $\Phi_h$ to $V_i$ is a $\mathrm{C}^1$-diffeomorphism such that $\Phi_h \circ \Phi_h = \mathrm{Id}$.

This last result provides proper assumptions on $\Phi_h$ to operate a change of variable in $I_i'$. We now define $G_{1,h} = \Phi_h(V_i \cap \Phi_h^{-1}(G) \cap F_h)$ and $G_{2,h} = \Phi_h^{-1}(V_i) \cap G \cap F_h$, and prove that $G_{1,h} = G_{2,h}$ in two steps.

(i) We first prove that $G_{1,h} \subset G_{2,h}$. Let $z \in G_{1,h}$. Then, there exists $z' \in V_i \cap \Phi_h^{-1}(G) \cap F_h$ such that $z = \Phi_h(z')$. Since $\Phi_h$ is an involution on $V_i$, we have $\Phi_h(z) = z'$. Since $z' \in V_i \cap A_h$, it comes that $z \in \Phi_h^{-1}(V_i) \cap \Phi_h^{-1}(A_h)$. Moreover, $z' \in \Phi_h^{-1}(G) \cap \Phi_h^{-1}(A_h)$, and thus, $z \in G \cap A_h$. Then, we have $z \in G_{2,h}$, which proves this first result.

(ii) We now prove that $G_{2,h} \subset G_{1,h}$. Let $z \in G_{2,h}$. In particular, $z \in G$. Then, there exists $j \in [K]$ such that $z \in V_j$. Therefore, $z \in B_j$ and thus $\|z - z_j\|_{z_j} < r(z_j) < 1$. Since $z \in A_h$, we obtain that $h < h_\star(z_j)$ with the same computations as those written above. In particular, this proves with **A**3 that $\Phi_h$ is an involution on $V_j$. Then, $z = \Phi_h(\Phi_h(z))$, with $\Phi_h(z) \in V_i \cap A_h$ and $\Phi_h(z) \in \Phi_h^{-1}(G) \cap \Phi_h^{-1}(A_h)$, since $z \in G \cap A_h$. Therefore, we have $z \in G_{1,h}$, which proves this last result.

Given the fact that $\Phi_h$ is an involution on $V_i$, we operate the change of variable $z \mapsto \Phi_h(z)$ in $I_i'$ and obtain

$$I_i' = \int_{\Phi_h(V_i \cap \Phi_h^{-1}(G) \cap F_h)} g(\Phi_h(z), z)\mathrm{d}z = \int_{\Phi_h^{-1}(V_i) \cap G \cap F_h} g(\Phi_h(z), z)\mathrm{d}z \ ,$$

which gives (64).

Therefore, combining (62), (63) and (64), we get

$$I' = \sum_{i=1}^K \int_{V_i \cap \Phi_h^{-1}(G) \cap F_h} g(z, \Phi_h(z))\mathrm{d}z = \sum_{i=1}^K \int_{\Phi_h^{-1}(V_i) \cap G \cap F_h} g(\Phi_h(z), z)\mathrm{d}z = J' \ ,$$

which concludes the proof. $\qquad\square$

We are now ready to establish the reversibility up to momentum reversal of $Q_1$, defined in (58), with respect to $\bar{\pi}$.

**Lemma 30.** *Assume* **A**1*,* **A**2*,* **A**3*. Then, for any $h > 0$, the Markov kernel $Q_1$ with step-size $h$, defined in (58), is reversible up to momentum reversal with respect to $\bar{\pi}$.*

*Proof.* Assume **A**1, **A**2, **A**3. Let $h > 0$. We recall that the kernels $Q_1$ and $Q_2$ are respectively defined in (58) and (59). We define the transition kernel $Q_3 : T^\star M \times \mathcal{B}(T^\star M) \to [0, 1]$ by

$$Q_3(z, \mathrm{d}z') = \mathbb{1}_{\bar{\mathsf{E}}_h^\Phi}(z)Q_2(z, \mathrm{d}z') + \mathbb{1}_{(\bar{\mathsf{E}}_h^\Phi)^c}(z)\delta_z(\mathrm{d}z') \ , \tag{65}$$

such that $Q_1 = s_\# Q_3$. The rest of the proof is divided into two parts. First, we prove that $Q_3$ is reversible with respect to $\bar{\pi}$. Then, we prove that $Q_1$ is reversible up to momentum reversal with respect to $\bar{\pi}$.

(a) Let $f \in C(\mathrm{T}^\star\mathsf{M} \times \mathrm{T}^\star\mathsf{M}, \mathbb{R})$ with compact support. We consider a compact set $\mathsf{K}$ with respect to the topology induced by the set $\{\mathsf{B}_{\|\cdot\|_z}(z, r), z \in \mathrm{T}^\star\mathsf{M}, r \in (0, 1)\}$ such that $\mathrm{supp}(f) \subset \mathsf{K} \times \mathsf{K}$. According to Definition 4, we aim to show that

$$\int_{\mathrm{T}^\star\mathsf{M}\times\mathrm{T}^\star\mathsf{M}} f(z, z')\mathsf{Q}_3(z, \mathrm{d}z')\bar{\pi}(\mathrm{d}z) = \int_{\mathrm{T}^\star\mathsf{M}\times\mathrm{T}^\star\mathsf{M}} f(z', z)\mathsf{Q}_3(z, \mathrm{d}z')\bar{\pi}(\mathrm{d}z) . \tag{66}$$

We denote by $I$ the left integral of (66). By combining (59) and (65), we have $I = I_1 + I_2 + I_3$ where

$$I_1 = \int_{\tilde{\mathsf{E}}_h^\Phi} \bar{\pi}(z)a(\mathrm{R}_h^\Phi(z) \mid z)f(z, \mathrm{R}_h^\Phi(z))\mathrm{d}z ,$$

$$I_2 = \int_{\tilde{\mathsf{E}}_h^\Phi} \bar{\pi}(z)[1 - a(\mathrm{R}_h^\Phi(z) \mid z)]f(z, z)\mathrm{d}z ,$$

$$I_3 = \int_{(\tilde{\mathsf{E}}_h^\Phi)^c} \bar{\pi}(z)f(z, z)\mathrm{d}z .$$

Since $\mathrm{supp}(f) \subset \mathsf{K} \times \mathsf{K}$, we have for any $z \in \mathsf{K}^c$, $f(z, \cdot) = 0$ and $f(\cdot, z) = 0$. Note also that for any $z \in ((\mathrm{R}_h^\Phi)^{-1}(\mathsf{K}))^c$, $\mathrm{R}_h^\Phi(z) \notin \mathsf{K}$, and thus $f(\mathrm{R}_h^\Phi(z), \cdot) = 0$ and $f(\cdot, \mathrm{R}_h^\Phi(z)) = 0$. By combining these preliminary results with (9), we get

$$I_1 = \int_{\tilde{\mathsf{E}}_h^\Phi \cap \mathsf{K} \cap (\mathrm{R}_h^\Phi)^{-1}(\mathsf{K})} \bar{\pi}(z)a(\mathrm{R}_h^\Phi(z) \mid z)f(z, \mathrm{R}_h^\Phi(z))\mathrm{d}z$$

$$= (1/Z)\int_{\tilde{\mathsf{E}}_h^\Phi \cap \mathsf{K} \cap (\mathrm{R}_h^\Phi)^{-1}(\mathsf{K})} \min\{e^{-H(z)}, e^{-H(\mathrm{R}_h^\Phi(z))}\}f(z, \mathrm{R}_h^\Phi(z))\mathrm{d}z$$

$$I_2 = \int_{\tilde{\mathsf{E}}_h^\Phi \cap \mathsf{K}} \bar{\pi}(z)[1 - a(\mathrm{R}_h^\Phi(z) \mid z)]f(z, z)\mathrm{d}z ,$$

$$I_3 = \int_{(\tilde{\mathsf{E}}_h^\Phi)^c} \bar{\pi}(z)f(z, z)\mathrm{d}z .$$

We denote by $J$ the right integral of (66). By symmetry, we have $J = J_1 + J_2 + J_3$ where

$$J_1 = \int_{\tilde{\mathsf{E}}_h^\Phi \cap \mathsf{K} \cap (\mathrm{R}_h^\Phi)^{-1}(\mathsf{K})} \bar{\pi}(z)a(\mathrm{R}_h^\Phi(z) \mid z)f(\mathrm{R}_h^\Phi(z), z)\mathrm{d}z$$

$$= (1/Z) \int_{\tilde{\mathsf{E}}_h^\Phi \cap \mathsf{K} \cap (\mathrm{R}_h^\Phi)^{-1}(\mathsf{K})} \min\{e^{-H(z)}, e^{-H(\mathrm{R}_h^\Phi(z))}\}f(\mathrm{R}_h^\Phi(z), z)\mathrm{d}z ,$$

$$J_2 = \int_{\tilde{\mathsf{E}}_h^\Phi \cap \mathsf{K}} \bar{\pi}(z)[1 - a(\mathrm{R}_h^\Phi(z) \mid z)]f(z, z)\mathrm{d}z ,$$

$$J_3 = \int_{(\tilde{\mathsf{E}}_h^\Phi)^c} \bar{\pi}(z)f(z, z)\mathrm{d}z .$$

We directly have $I_2 = J_2$ and $I_3 = J_3$. Let us now prove that $I_1 = J_1$.

We recall that $s \circ \mathrm{S}_{h/2}$ is a symplectic $\mathrm{C}^1$-diffeomorphism (see Section 3.1) and $\tilde{\mathsf{E}}_h^\Phi = (s \circ \mathrm{S}_{h/2})(\mathsf{F}_h)$ where $\mathsf{F}_h = \mathsf{A}_h \cap \Phi_h^{-1}(\mathsf{A}_h)$, using (57). We define $\mathsf{K}_h = \mathsf{F}_h \cap (s \circ \mathrm{S}_{h/2})(\mathsf{K}) \cap \Phi_h^{-1}((s \circ \mathrm{S}_{h/2})(\mathsf{K}))$, such that $\tilde{\mathsf{E}}_h^\Phi \cap \mathsf{K} \cap (\mathrm{R}_h^\Phi)^{-1}(\mathsf{K}) = (s \circ \mathrm{S}_{h/2})(\mathsf{K}_h)$, and we operate the change of variable $z \mapsto (s \circ \mathrm{S}_{h/2})(z)$ in $I_1$ and $J_1$

$$I_1 = (1/Z) \int_{\mathsf{K}_h} \min\{e^{-H((s \circ \mathrm{S}_{h/2})(z))}, e^{-H((s \circ \mathrm{S}_{h/2} \circ \Phi_h)(z))}\}f((s \circ \mathrm{S}_{h/2})(z), (s \circ \mathrm{S}_{h/2} \circ \Phi_h)(z))\mathrm{d}z ,$$

$$J_1 = (1/Z) \int_{\mathsf{K}_h} \min\{e^{-H((s \circ \mathrm{S}_{h/2})(z))}, e^{-H((s \circ \mathrm{S}_{h/2} \circ \Phi_h)(z))}\}f((s \circ \mathrm{S}_{h/2} \circ \Phi_h)(z), (s \circ \mathrm{S}_{h/2})(z))\mathrm{d}z .$$

We now define the map $g : \mathrm{T}^\star\mathsf{M} \times \mathrm{T}^\star\mathsf{M} \to \mathbb{R}$ and the set $\mathsf{G} \subset \mathrm{T}^\star\mathsf{M}$ by

$$g(z, z') = \min\{e^{-H((s \circ \mathrm{S}_{h/2})(z))}, e^{-H((s \circ \mathrm{S}_{h/2})(z'))}\}f((s \circ \mathrm{S}_{h/2})(z), (s \circ \mathrm{S}_{h/2})(z')) ,$$

$$\mathsf{G} = (s \circ \mathrm{S}_{h/2})(\mathsf{K}) .$$

Note that $\mathsf{G}$ is a compact set of $\mathrm{T}^\star\mathsf{M}$ by continuity of $s \circ \mathrm{S}_{h/2}$ and $g$ is a continuous function by continuity of $H$ and $s \circ \mathrm{S}_{h/2}$. Then, we obtain $I_1 = J_1$, by applying Lemma 29 with $g$ and $\mathsf{G}$. Finally, we obtain $I = J$ and thus prove (66) for any continuous function $f$ with compact support.

(b) Let $f : \mathrm{T}^\star\mathsf{M} \times \mathrm{T}^\star\mathsf{M} \to \mathbb{R}$ be a continuous function with compact support. We have

$$\int_{\mathrm{T}^\star\mathsf{M}\times\mathrm{T}^\star\mathsf{M}} f(z, z')\mathsf{Q}_1(z, \mathrm{d}z')\bar{\pi}(\mathrm{d}z)$$

$$= \int_{\mathrm{T}^\star\mathsf{M}\times\mathrm{T}^\star\mathsf{M}} f(z, z')(s_\# \mathsf{Q}_3)(z, \mathrm{d}z')\bar{\pi}(\mathrm{d}z)$$

$$= \int_{\mathrm{T}^\star\mathsf{M}\times\mathrm{T}^\star\mathsf{M}} f(z, s(z'))\mathsf{Q}_3(z, \mathrm{d}z')\bar{\pi}(\mathrm{d}z) \qquad \text{(momentum reversal on } z')$$

$$= \int_{\mathrm{T}^\star\mathsf{M}\times\mathrm{T}^\star\mathsf{M}} f(z', s(z))\mathsf{Q}_3(z, \mathrm{d}z')\bar{\pi}(\mathrm{d}z) \qquad \text{(using (66))}$$

$$= \int_{\mathrm{T}^\star\mathsf{M}\times\mathrm{T}^\star\mathsf{M}} f(s(z'), s(z))(s_\# \mathsf{Q}_3)(z, \mathrm{d}z')\bar{\pi}(\mathrm{d}z) \qquad \text{(momentum reversal on } z')$$

$$= \int_{\mathrm{T}^\star\mathsf{M}\times\mathrm{T}^\star\mathsf{M}} f(s(z'), s(z))\mathsf{Q}_1(z, \mathrm{d}z')\bar{\pi}(\mathrm{d}z) ,$$

which concludes the proof.

$\square$

We are now ready to prove Theorem 3, which states that Q is reversible up to momentum reversal with respect to $\bar{\pi}$.

*Proof of Theorem 3.* Assume **A**1, **A**2, **A**3. This proof is very similar to the proof of Theorem 15 (see Appendix E.2). Let $f : T^\star M \times T^\star M \to \mathbb{R}$ be a continuous function with compact support. We have

$\int_{T^\star M \times T^\star M} f(z, z') Q(z, dz') \bar{\pi}(dz)$

$\quad = \int_{(T^\star M)^4} f(z, z') Q_0(z, dz_1) Q_1(z_1, dz_2) Q_0(z_2, dz') \bar{\pi}(dz)$          (see (60))

$\quad = \int_{(T^\star M)^4} f(s(z_1), z') Q_0(z, dz_1) Q_1(s(z), dz_2) Q_0(z_2, dz') \bar{\pi}(dz)$   (Lemma 17)

$\quad = \int_{(T^\star M)^4} f(s(z_1), z') Q_0(s(z), dz_1) Q_1(z, dz_2) Q_0(z_2, dz') \bar{\pi}(dz)$   (momentum reversal on $z$)

$\quad = \int_{(T^\star M)^4} f(s(z_1), z') Q_0(z_2, dz_1) Q_1(z, dz_2) Q_0(s(z), dz') \bar{\pi}(dz)$   (Lemma 30)

$\quad = \int_{(T^\star M)^4} f(s(z_1), z') Q_0(z_2, dz_1) Q_1(s(z), dz_2) Q_0(z, dz') \bar{\pi}(dz)$   (momentum reversal on $z$)

$\quad = \int_{(T^\star M)^4} f(s(z_1), s(z)) Q_0(z_2, dz_1) Q_1(z', dz_2) Q_0(z, dz') \bar{\pi}(dz)$   (Lemma 17)

$\quad = \int_{T^\star M \times T^\star M} f(s(z'), s(z)) Q(z, dz') \bar{\pi}(dz) \,.$

Moreover, $s_\# \bar{\pi} = \bar{\pi}$. Hence, by combining Definition 4 and Lemma 5, we obtain the result of Theorem 3. $\qquad\square$

## J   Comparison with Kook et al. (2022a).

In this section, we provide a precise comparison between our work and the results stated in Kook et al. (2022a). We first dwell on the differences related to the general setting and the algorithms, and then explain how our methodology may solve the limitations of Kook et al. (2022a).

### J.1   General framework

Given a convex body $K \subset \mathbb{R}^d$ and a function $c : \mathbb{R}^d \to \mathbb{R}^m$, consider
$$M = \{x \in K : c(x) = 0\} \,. \tag{67}$$
In their paper, Kook et al. (2022a) aim at sampling from a target distribution $\pi$, with density given for $x \in M$ by
$$d\pi(x)/dx = \exp[-V(x)]/Z \,,$$
where $V \in C^2(M, \mathbb{R})$, $Z = \int_M \exp[-V(x)]dx$. They make the following assumptions:

(a) $K$ is provided with a self-concordant barrier $\phi$.

(b) The differential of $c$ at $x$, $Dc(x)$, is full-ranked for any $x \in M$.

Although this setting might be more general than ours (consider for instance the case where $K$ is a polytope and $c$ is a non-trivial function), Kook et al. (2022a) focus on the case
$$K = \{x \in \mathbb{R}^d : \ell \le x \le u\}, \quad c(x) = Ax - b \tag{68}$$
where $\ell, u \in \mathbb{R}^d$ with $\ell < u$, $b \in \mathbb{R}^m$, and $A \in \mathbb{R}^{d \times m}$. Combined with assumptions (a) and (b), $A$ must have independent rows and the Hessian of $\phi$ at $x$ given by $\mathfrak{g}(x) = D^2\phi(x)$ is a diagonal positive matrix for any $x$. Moreover, $M$ defined by (67)-(68) is a polytope. However, not any polytope can be rewritten in a manner akin to (68), and therefore, the setting chosen by Kook et al. (2022a) can actually be considered as *a special case of our setting*, see Section 2.

Then, they consider the extended state-space $\bar{M} = \{(x, p) \in \mathbb{R}^d \times \mathbb{R}^d : x \in M, p \in \mathrm{Ker}(Dc(x))\}$ and define on $\bar{M}$ a *constrained* Hamiltonian $H$, given by
$$H(x, p) = \bar{H}(x, p) + \lambda(x, p)^\top c(x) \,,$$
$$\text{with } \bar{H}(x, p) = V(x) + \frac{1}{2} \log \mathrm{pdet}\, M(x) + \frac{1}{2} p^\top M(x)^\dagger p \,,$$
$$\text{and } M(x) = Q(x)^\top \mathfrak{g}(x) Q(x) \,,$$

where $Q$ is the orthogonal projection into $\mathrm{Ker}(Dc(x))$, implying that $M$ is semi-definite positive, and $\lambda(x,p)$ is a Lagrangian term given by

$$\lambda(x,p) = (Dc(x)Dc(x)^\top)^{-1}\{D^2c(x)[p, \mathrm{d}x/\mathrm{d}t] - Dc(x)\partial\bar{H}(x,p)/\partial x\}\,.$$

Remark that $\lambda$ simplifies when $c$ is chosen as in (68), but may not be well suited for higher order constraints. Then, Kook et al. (2022a) provide simplifications of $H$ for the setting (68) based on formulas for $\mathrm{pdet}\,M$ and $M^\dagger$.

Finally, they consider the joint distribution $\bar{\pi}$ given for any $(x,p) \in \bar{\mathsf{M}}$ by

$$\mathrm{d}\bar{\pi} = (1/\bar{Z})\exp[-H(x,p)]\mathrm{d}x\mathrm{d}p,$$

where $\bar{Z} = \int_{\bar{\mathsf{M}}}\exp[-H(x,p)]\mathrm{d}x\mathrm{d}p$, for which the first marginal is $\pi$. They naturally propose to sample from $\bar{\pi}$ by implementing a version of RMHMC relying on the *constrained* Hamiltonian $H$, which they call CRHMC.

Similarly to our approach, they consider two cases.

(a) First, Kook et al. (2022a) assume that the Hamiltonian dynamics given by $H$ can be explicitly computed in continuous time (which is not the case in practice). They present the continuous version of RMHMC which incorporates the simplified *constrained* Hamiltonian $H$, see Algorithm 1. We refer to this algorithm as c-CRHMC. This algorithm is identical to our algorithm c-BHMC provided in Appendix D, apart from the involution checking step (see Line 4 in Algorithm 2). They state in Theorem 6 that the Markov kernel corresponding to one iteration of c-CRHMC satisfies detailed balance with respect to $\bar{\pi}$, thus ensuring that it preserves $\bar{\pi}$.

(b) Then, Kook et al. (2022a) provide a practical implementation of CRHMC, which is similar to n-BHMC, based on a time-discretization of the Hamiltonian dynamics for a given time-step $h > 0$. They first split the simplified *constrained* Hamiltonian $H$ into $H_1$ and $H_2$, by leveraging the non-separable aspect of $H$ in $H_2$, as we do in Section 3.1. Then, they propose to use the first-order Euler method to solve the discretized ODE associated to $H_1$, as we also suggest, and the *Implicit Midpoint Integrator* (IMI) to solve the discretized ODE associated with $H_2$, while we implement the *Generalized Leapfrog Integrator* (GLI). We insist on the fact that these two second-order methods share the same theoretical properties (Hairer et al., 2006). Since IMI is implicit, they propose to approximate it with a numerical integrator $\Phi_h$, which is computed with a fixed-point method (as we do) detailed in Algorithm 3. Finally, they implement CRHMC which contains the same steps as n-BHMC *apart from the involution checking step* (see Line 8 in Algorithm 1): (i) refreshing the momentum with a symplectic scheme (Step 1 in both algorithms), (ii) solving the discretized ODE associated to $H$ in three steps according to the splitting given by $H_1$ and $H_2$ (Step 2 in both algorithms), (iii) applying a Metropolis-Hastings filter (Step 3 in both algorithms), and (iv) operating a final momentum reversal (Step 4 in n-BHMC). They state in Theorem 8 that the Markov kernel corresponding to one iteration of CRHMC satisfies detailed balance with respect to $\bar{\pi}$, thus ensuring that it preserves $\bar{\pi}$.

### J.2 Theoretical gaps in the reversibility of CRHMC Kook et al. (2022a)

First, we call into question the proof of the reversibility of c-CRHMC stated in (Kook et al., 2022a, Theorem 6). Indeed, Kook et al. (2022a) implicitly assume that the time-continuous Hamiltonian dynamics have a solution at any time, but do not provide any proof of this result. We emphasize here that this result is not trivial and possibly may not hold due to the pathological behaviour of the barrier at the border of $\mathsf{M}$, see Proposition 14. In contrast, the involution checking step implemented in c-BHMC verifies this condition for the *forward* and the *reversed* dynamics after momentum reversal, which then allows us to derive the reversibility of c-BHMC, see Theorem 15.

Finally, we report some gaps in the proof of reversibility of CRHMC provided in (Kook et al., 2022a, Theorem 8). Indeed, while Kook et al. (2022a) claim that (Hairer et al., 2006, Theorem VI.3.5.) applies to CRHMC which contains the *numerical* integrator detailed in (Kook et al., 2022a, Algorithm 3), in fact (Hairer et al., 2006, Theorem VI.3.5.) only applies to the *implicit* integrator (IMI). This is a fundamental limitation of their theory. This theoretical confusion thus leads to numerical errors, as we detail in Section 6. In contrast, we present in Section 4 theoretical results on the implicit integrator (GLI), which allows us to derive reasonable assumptions on the numerical integrator $\Phi_h$ to obtain reversibility of n-BHMC in Theorem 3. Contrary to Kook et al. (2022a), our proof relies on the properties of $\Phi_h$ and highlights the critical role of the involution checking step.

# K   Experimental details

The numerical experiments presented in Section 6 are based on the MATLAB implementation of CRHMC provided by Kook et al. (2022a). We adapt this code for n-BHMC for sake of fairness. In particular, our implementation differs from CRHMC by the use of the Generalized Leapfrog integrator and the "involution checking" mechanism.

**Details on experiments with synthetic data.**    In these experiments, the hypercube refers to the set $[-1/2, 1/2]^d$ where $d \in \{5, 10\}$. We recall that the ground truth on the quantities we estimate is given by the Metropolis Adjusted Langevin Algorithm (MALA) (Roberts & Stramer, 2002) for the hypercube and the Independent Metropolis Hastings (IMH) sampler (Liu, 1996) for the simplex. We now give details on how the parameters of these two algorithms are chosen. For MALA, we use a constant step-size $h = 0.05$ for both dimensions, which results in an average acceptance probability equal to $0.55$ for $d = 5$ and $0.44$ for $d = 10$. For IMH, we use as proposal the uniform distribution, which is simply a Dirichlet distribution, resulting in an average acceptance probability of order $0.36$. We recall that we use an adaptive step-size $h$ in CRHMC and n-BHMC such that we obtain an average acceptance probability of order $0.5$ in the MH filter. We now discuss the setting of the tolerance parameter $\eta$ in n-BHMC. We choose $\eta$ so that (i) the step-size $h$ is at least greater than $10^{-2}$ over the iterations of the algorithm and (ii) the average acceptance probability is roundly equal to $0.5$. This heuristic results in defining (a) for the hypercube, $\eta = 5$ if $d = 5$ and $\eta = 10$ if $d = 10$ and (b) for the simplex, $\eta = 10$ if $d = 5$ and $\eta = 200$ if $d = 10$.

**Details on experiments with real-world data.**    For these experiments, we consider the exact same setting as in Kook et al. (2022a). In particular, we use the same adaptation strategies to update the step-size and the barrier functions.

**Influence of the norm in** (8).    The norm chosen for the "involution checking step" is arbitrary and many options are available. In particular, one could design $\|z^{(0)} - \Phi_h(z^{(1)})\|_2 > \eta$, and thus pick the Euclidean norm in (8). However, while this approach should theoretically also reduce the bias in the method, we remark that the obtained Markov chains have very poor mixing time. This is due to the fact that in practice the update on the momentum is of order $\|\mathfrak{g}(x)\|_2^{1/2}$ while the update on the position is of order $\|\mathfrak{g}(x)^{-1}\|_2^{1/2}$. Hence the Euclidean norm is ill-suited for the "involution checking step" and the proposal states are rejected a lot more near the boundaries, see Figure 3.

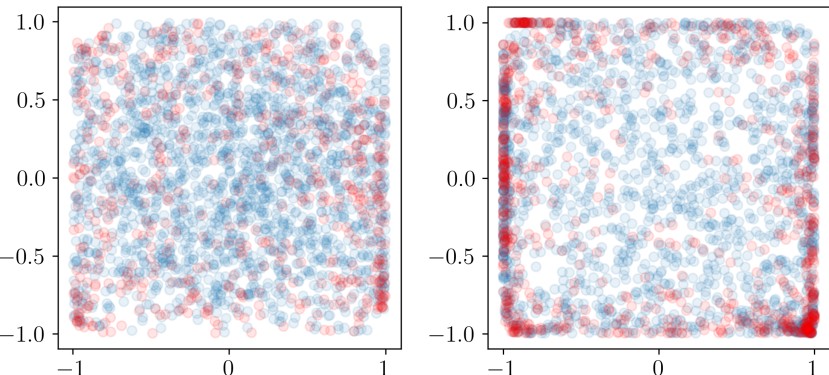

Figure 3: Outputs of n-BHMC after 25k iterations to sample from the uniform distribution over $[-1, 1]^2$ with $\eta = 10^{-3}$, $h = 0.8$, using in (8) the "self-concordant" norm (left) or the Euclidean norm (right). Red samples are rejected and blue samples are accepted.

