which concludes the proof. $\qquad\qquad\square$

**Lemma 6.** *Let* $Q : T^\star M \times \mathcal{B}(T^\star M) \to [0,1]$ *be a transition probability kernel and let* $\bar{\pi}$ *be a probability distribution on* $T^\star M$. *Assume that* $s_\# Q = Q$ *and that* $Q$ *is reversible with respect to* $\bar{\pi}$. *Then,* $Q$ *is reversible up to momentum reversal with respect to* $\bar{\pi}$.

*Proof.* Let $f \in C(T^\star M, \mathbb{R})$ with compact support. We have

$$\int_{T^\star M \times T^\star M} f(z, z') Q(z, dz') \bar{\pi}(dz)$$
$$= \int_{T^\star M \times T^\star M} f(z, s(z'))(s_\# Q)(z, dz') \bar{\pi}(dz) \qquad \text{(momentum reversal on } z')$$
$$= \int_{T^\star M \times T^\star M} f(z', s(z)) Q(z, dz') \bar{\pi}(dz) \qquad \text{(Definition 4)}$$
$$= \int_{T^\star M \times T^\star M} f(s(z'), s(z))(s_\# Q)(z, dz') \bar{\pi}(dz) \qquad \text{(momentum reversal on } z')$$
$$= \int_{T^\star M \times T^\star M} f(s(z'), s(z)) Q(z, dz') \bar{\pi}(dz) \,,$$

which concludes the proof. $\qquad\qquad\square$

**Useful inequalities.** The following inequalities hold:

(a) For any $u \in [0, 1/5]$, $(1-u)^{-2} \le 1 + 3u$ and $(1-u)^{-3} \le 1 + 5u$.

(b) For any $u \in [0, 1/2]$, $(1-u)^{-1} \le 1 + 2u$.

(c) For any $u \in [0, 1]$, $|(1-u)^2 - 1| \le 3u$.

(d) For any $u \ge 0$, we have $1 - (1+u)^{-1} \le u$ and $(1-u)^2 - 1 \ge -2u$.

# B  Details on Riemannian metrics

Let $M$ be a smooth $d$-dimensional manifold, endowed with a metric $\mathfrak{g}$. We recall that the *Riemannian volume element* corresponding to $(M, \mathfrak{g})$ is given in local coordinates by $\mathrm{dvol}_M(x) = \sqrt{\det(\mathfrak{g})} dx$, where $dx$ is a dual coframe, (Lee, 2006, Lemma 3.2.). For any $x \in M$, we respectively denote by $T_x M$ and $T_x^\star M$, the tangent space at $x$ and its dual space, *i.e.*, the cotangent space at $x$. Note that $T_x M$ and $T_x^\star M$ are space vectors, and $T_x M$ is endowed with the scalar product $\langle \cdot, \cdot \rangle_{\mathfrak{g}(x)}$ by definition of the Riemannian metric. For clarity sake, we denote by $v$ (resp. $p$) an element of the tangent (resp. cotangent) space. We recall that the tangent bundle $TM$ and the cotangent bundle $T^\star M$ are respectively defined by $TM = \sqcup_{x \in M} \{x\} \cup T_x M$ and $T^\star M = \sqcup_{x \in M} \{x\} \cup T_x^\star

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

 4\|\mathrm{M}_0(y)^{-1}(y - x^{(0)}) - \mathrm{M}_0(y')^{-1}(y' - x^{(0)})\|_{\mathfrak{g}_0^{-1}}\|\mathrm{M}_0(y)^{-1}(y - x^{(0)})\|_{\mathfrak{g}_0^{-1}}$$

$$+ 2\|\mathrm{M}_0(y)^{-1}(y - x^{(0)}) - \mathrm{M}_0(y')^{-1}(y' - x^{(0)})\|_{\mathfrak{g}_0^{-1}}^2 .$$

In particular, $\mathrm{M}_0(y)^{-1}(y - x^{(0)}) - \mathrm{M}_0(y')^{-1}(y' - x^{(0)}) = \mathrm{M}_0(y)^{-1}(y - y') + (\mathrm{M}_0(y)^{-1} - \mathrm{M}_0(y')^{-1})(y' - x^{(0)})$, and thus we have

$$\|\mathrm{M}_0(y)^{-1}(y - x^{(0)}) - \mathrm{M}_0(y')^{-1}(y' - x^{(0)})\|_{\mathfrak{g}_0^{-1}} \tag{42}$$

$$\leq \|\mathfrak{g}_0^{-1/2}\mathrm{M}_0(y)^{-1}\mathfrak{g}_0^{-1/2}\|_2\|y - y'\|_{\mathfrak{g}_0} + r\|\mathfrak{g}_0^{-1/2}(\mathrm{M}_0(y)^{-1} - \mathrm{M}_0(y')^{-1})\mathfrak{g}_0^{-1/2}\|_2$$

$$\leq (1 + (1-r)^2)^{-1}\|y - y'\|_{\mathfrak{g}_0} + r\|\mathfrak{g}_0^{-1/2}(\mathrm{M}_0(y')^{-1} - \mathrm{M}_0(y)^{-1})\mathfrak{g}_0^{-1/2}\|_2 . \qquad \text{(using (38))}$$

We now aim to find an upper bound for $\|\mathfrak{g}_0^{-1/2}(\mathrm{M}_0(y)^{-1} - \mathrm{M}_0(y')^{-1})\mathfrak{g}_0^{-1/2}\|_2$. We have

$$\mathfrak{g}_0^{-1/2}(\mathrm{M}_0(y)^{-1} - \mathrm{M}_0(y')^{-1})\mathfrak{g}_0^{-1/2}$$
$$= (\mathrm{I}_d + \mathfrak{g}_0^{1/2}\mathfrak{g}(y)^{-1}\mathfrak{g}_0^{1/2})^{-1} - (\mathrm{I}_d + \mathfrak{g}_0^{1/2}\mathfrak{g}(y')^{-1}\mathfrak{g}_0^{1/2})^{-1}$$
$$= (\mathrm{I}_d + \mathfrak{g}_0^{1/2}\mathfrak{g}(y')^{-1}\mathfrak{g}_0^{1/2} + \mathfrak{g}_0^{1/2}\{\mathfrak{g}(y)^{-1} - \mathfrak{g}(y')^{-1}\}\mathfrak{g}_0^{1/2})^{-1} - (\mathrm{I}_d + \mathfrak{g}_0^{1/2}\mathfrak{g}(y')^{-1}\mathfrak{g}_0^{1/2})^{-1}$$
$$= \mathrm{B}(y')^{-1/2}[(\mathrm{I}_d + \mathrm{B}(y')^{-1/2}\mathfrak{g}_0^{1/2}\{\mathfrak{g}(y)^{-1} - \mathfrak{g}(y')^{-1}\}\mathfrak{g}_0^{1/2}\mathrm{B}(y')^{-1/2})^{-1} - \mathrm{I}_d]\mathrm{B}(y')^{-1/2} \, ,$$

where $\mathrm{B}(y') = \mathrm{I}_d + \mathfrak{g}_0^{1/2}\mathfrak{g}(y')^{-1}\mathfrak{g}_0^{1/2}$. In particular, we have

$$(1 + (1-r)^{-2})^{-1/2}\mathrm{I}_d \preceq \mathrm{B}(y')^{-1/2} \preceq (1 + (1-r)^2)^{-1/2}\mathrm{I}_d \, ,$$

by Lemma 9-(b). Note that (17) still holds, where $x, x'$ and $\bar{r}$ are respectively replaced by $y, y'$ and $r$. Thus, on the model on (18), we have

$$\{(1 - \|y' - y\|_{\mathfrak{g}(y)})^2 - 1\}(1-r)^2\mathrm{I}_d \preceq \mathfrak{g}_0^{1/2}\{\mathfrak{g}(y)^{-1} - \mathfrak{g}(y')^{-1}\}\mathfrak{g}_0^{1/2} \preceq \{(1 - \|y' - y\|_{\mathfrak{g}(y)})^{-2} - 1\}(1-r)^{-2}\mathrm{I}_d \, ,$$

and then

$$(1 + \{(1 - \|y' - y\|_{\mathfrak{g}(y)})^2 - 1\}(1-r)^2(1 + (1-r)^{-2})^{-1})\mathrm{I}_d$$

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

{\mathrm{g}}_{h,z}(\mathsf{B}) \subset \mathsf{B}$. Let $z' \in \mathsf{B}$, we have by Lemma 9-(a)

$$\|\underline{\mathrm{g}}_{h,z}(z') - z\|_z = \tfrac{h}{2}\|\{\mathfrak{g}(x)^{-1} + \mathfrak{g}(x')^{-1}\}p'\|_{\mathfrak{g}(x)} + \tfrac{h}{4}\|D\mathfrak{g}(x)[\mathfrak{g}(x)^{-1}p', \mathfrak{g}(x)^{-1}p']\|_{\mathfrak{g}(x)^{-1}}$$

$$= \tfrac{h}{2}\|2\mathfrak{g}(x)^{-1}p' + \{\mathfrak{g}(x')^{-1} - \mathfrak{g}(x)^{-1}\}p'\|_{\mathfrak{g}(x)} + \tfrac{h}{4}\|D\mathfrak{g}(x)[\mathfrak{g}(x)^{-1}p', \mathfrak{g}(x)^{-1}p']\|_{\mathfrak{g}(x)^{-1}}$$

$$\leq \tfrac{h}{2}(2\|p'\|_{\mathfrak{g}(x)^{-1}} + \|\mathfrak{g}(x)^{1/2}(\mathfrak{g}(x')^{-1} - \mathfrak{g}(x)^{-1})\mathfrak{g}(x)^{1/2}\|_2\|p'\|_{\mathfrak{g}(x)^{-1}}) + \tfrac{h}{2}\|p'\|_{\mathfrak{g}(x)^{-1}}^2 ,$$

where the following inequalities hold

(a) $\|p'\|_{\mathfrak{g}(x)^{-1}} = \|p' - p + p\|_{\mathfrak{g}(x)^{-1}} \leq r + \|p\|_{\mathfrak{g}(x)^{-1}}$.

(b) $\|\mathfrak{g}(x)^{1/2}(\mathfrak{g}(x')^{-1} - \mathfrak{g}(x)^{-1})\mathfrak{g}(x)^{1/2}\|_2 \leq 3\|x - x'\|_{\mathfrak{g}(x)} \leq 3r$, on the model of (18), using $r \leq 1/11$.

Therefore, we have

$$\|\underline{\mathrm{g}}_{h,z}(z') - z\|_z \leq h((r + \|p\|_{\mathfrak{g}(x)^{-1}}) + \tfrac{3}{2}r(r + \|p\|_{\mathfrak{g}(x)^{-1}}) + \tfrac{1}{2}(r + \|p\|_{\mathfrak{g}(x)^{-1}})^2)$$

$$\leq h((r + \|p\|_{\mathfrak{g}(x)^{-1}})(1 + \tfrac{3}{2}r) + \tfrac{1}{2}(r + \|p\|_{\mathfrak{g}(x)^{-1}})^2)$$

$$\leq hr/h_1(\|p\|_{\mathfrak{g}(x)^{-1}}, r, \alpha) < r ,$$

which proves the statement.

*Step 2.* We now prove that $\underline{\mathrm{g}}_{h,z}$ is a contraction on $\mathsf{B}$. This proof notably recovers some elements of the proof of Proposition 13 (see Appendix E). Let $(z_1, z_2) \in \mathsf{B} \times \mathsf{B}$ with $z_1 = (x_1, p_1)$ and $z_2 = (x_2, p_2)$. Remark that $x_1, x_2 \in \mathsf{W}^0(x, 1) \times \mathsf{W}^0(x, 1)$. Let us first bound $\|\underline{\mathrm{g}}_{h,z}^{(1)}(z_1) - \underline{\mathrm{

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

 $\mathtt{L} > 0$ and $\mathtt{M} > 0$ such that $r^\star$, given by **A**3 and defined in (52), is $\mathtt{L}$-Lipschitz-continuous on $\mathsf{M}$ with respect to $\|\cdot\|_2$ and bounded from above by $\mathtt{M}$ (see Lemma 27).

We define $r : \mathrm{T}^\star\mathsf{M} \to (0, +\infty)$ by $r(x, p) = r^\star(x)/\mathtt{m}$ for any $(x, p) \in \mathrm{T}^\star\mathsf{M}$, where $\mathtt{m} = \max\{1/\lambda, 4\mathtt{M}, 4\mathtt{L}\}$ and $\lambda$ is given by **A**3. Using the properties of $r^\star$, it comes that

(a) $r$ is $\mathtt{L}$-Lipschitz on $\mathrm{T}^\star\mathsf{M}$ with respect to $\|\cdot\|_2$,

(b) $r(x, p) \le 1/(4\mathtt{L}C_x)$ for any $(x, p) \in \mathrm{T}^\star\mathsf{M}$, where $C_x$ is defined in Lemma 12,

(c) $r \le 1/4$, and

(d) $r \le \lambda r^\star$.

Note that $\mathsf{G} \subset \cup_{z \in \mathsf{G}} \mathsf{B}_{\|\cdot\|_z}(z, r(z))$. Since $\mathsf{G}$ is a compact set, there exist $K \in \mathbb{N}$ and $(z_i)_{i \in [K]} \in \mathrm{T}^\star\mathsf{M}^K$ such that $\mathsf{G} \subset \bigcup_{i=1}^K \mathsf{B}_i$, where $\mathsf{B}_i = \mathsf{B}_{\|\cdot\|_{z_i}}(z_i, r(z_i))$.

We consider the sequence $\{\mathsf{V}_i\}_{i=1}^K$ constructed as follows: $\mathsf{V}_1 = \mathsf{G} \cap \mathsf{B}_1$ and for any $i \in \{2, \ldots, K\}$, $\mathsf{V}_i = (\mathsf{G} \cap \mathsf{B}_i) \cap (\cup_{j=1}^{i-1} \mathsf{V}_j)^c$. Then, we have that, for any $i \in \{1, \ldots, N\}$, $\cup_{j=1}^i \mathsf{V}_j = \mathsf{G} \cap (\cup_{j=1}^i \mathsf{B}_j)$, and that for any $i_1, i_2 \in \{1, \ldots, K\}$, $\mathsf{V}_{i_1} \cap \mathsf{V}_{i_2} = \emptyset$ if $i_1 \neq i_2$. Therefore, we get that $\mathsf{G} = \sqcup_{i=1}^K \mathsf{V}_i$ and $\Phi_h^{-1}(\mathsf{G}) = \sqcup_{i=1}^K \Phi_h^{-1}(\mathsf{V}_i)$. In particular, for any $\mathsf{A} \in \mathcal{B}(\mathrm{T}^\star\mathsf{M})$ and any $\zeta \in \mathrm{C}_c(\mathrm{T}^\star\mathsf{M}, \mathbb{R})$

$$\textstyle\int_{\mathsf{G} \cap \mathsf{A}} \zeta(z)\mathrm{d}z = \sum_{i=1}^K \int_{\mathsf{V}_i \cap \mathsf{A}} \zeta(z)\mathrm{d}z \; , \tag{62}$$

and for any $\tilde{\mathsf{A}} \in \mathcal{B}(\mathrm{T}^\star\mathsf{M})$ and any $\tilde{\zeta} \in \mathrm{C}_c(\mathrm{T}^\star\mathsf{M}, \mathbb{R})$

$$\textstyle\int_{\Phi_h^{-1}(\mathsf{G}) \cap \tilde{\mathsf{A}}} \tilde{\zeta}(z)\mathrm{d}z = \sum_{i=1}^K \int_{\Phi_h^{-1}(\mathsf{V}_i) \cap \tilde{\mathsf{A}}} \tilde{\zeta}(z)\mathrm{d}z \; . \tag{63}$$

Using (62) and (61), we obtain $I' = \sum_{i=1}^K I_i'$, where $I_i' = \int_{\mathsf{V}_i \cap \Phi_h^{-1}(\mathsf{G}) \cap \mathsf{F}_h} g(z, \Phi_h(z))\mathrm{d}z$ for any $i \in [K]$. We are now going to show that, for any $i \in [K]$

$$I_i' = \textstyle\int_{\Phi_h^{-1}(\mathsf{V}_i) \cap \mathsf{G} \cap \mathsf{F}_h} g(\Phi_h(z), z)\mathrm{d}z \; . \tag{64}$$

Let $i \in [K]$. We proceed by making the following case disjunction:

(a) Either $\mathsf{V}_i \cap \Phi_h^{-1}(\mathsf{G}) \cap \mathsf{F}_h = \emptyset$, and then $I_i' = 0$. We prove by contradiction in this case that $\Phi_h^{-1}(\mathsf{V}_i) \cap \mathsf{G} \cap \mathsf{F}_h = \emptyset$. Assume that there exists $z \in \Phi_h^{-1}(\mathsf{V}_i) \cap \mathsf{G} \cap \mathsf{F}_h$. By definition of $\mathsf{F}_h$, $z \in \mathsf{A}_h$, $\Phi_h(z) \in \mathsf{A}_h$, and we get that $\Phi_h(\Phi_h(z)) = z$ using **A**3. Hence, we have:

- $\Phi_h(z) \in \mathsf{V}_i$,
- $\Phi_h(\Phi_h(z)) = z \in \mathsf{G}$,
- $\Phi_h(z) \in \mathsf{A}_h$,
- $\Phi_h(\Phi_h(z)) = z \in \mathsf{A}_h$.

Therefore, we get $\Phi_h(z) \in \mathsf{V}_i \cap \Phi_h^{-1}(\mathsf{G}) \cap \mathsf{F}_h = \emptyset$, which is absurd. Finally, (64) holds since

$$I_i' = 0 = \textstyle\int_{\Phi_h^{-1}(\mathsf{V}_i) \cap \mathsf{G} \cap \mathsf{F}_h} g(\Phi_h(z), z)\mathrm{d}z \; .$$

(b) Either, there exists some $\tilde{z} \in \mathsf{V}_i \cap \Phi_h^{-1}(\mathsf{G})$ such that $\tilde{z} \in \mathsf{F}_h$. In particular, $\tilde{z} \in \mathsf{B}_i$, and thus $\|\tilde{z} - z_i\|_{z_i} < r(z_i) < 1$. In this case, by combining **A**3 with Lemma 9-(c), we have for any $z' \in \mathrm{T}^\star\mathsf{M}$

$$\|z'\|_{\tilde{z}} \le (1 - \|\tilde{z} - z_i\|_{z_i})^{-1}\|z'\|_{z_i} < (1 - r(z_i))^{-1}\|z'\|_{z_i} .$$

By considering $z' = z_i - \tilde{z}$, it comes that

$$\|z_i - \tilde{z}\|_{\tilde{z}} < (1 - r(z_i))^{-1}\|\tilde{z} - z_i\|_{z_i} < (1 - r(z_i))^{-1}r(z_i) .$$

By combining properties (a) and (b) of $r$ with Lemma 12, we have the following upper bound of $r(z_i)$

$$r(z_i) \le r(\tilde{z}) + \mathrm{L}\|\tilde{z} - z_i\|_2 \le r(\tilde{z}) + \mathrm{L}C_{x_i}\|\tilde{z} - z_i\|_{z_i} < r(\tilde{z}) + \mathrm{L}C_{x_i}r(z_i) < r(\tilde{z}) + 1/4 ,$$

and thus

$$\|z_i - \tilde{z}\|_{\tilde{z}} < (1 - r(\tilde{z}) - 1/4)^{-1}(r(\tilde{z}) + 1/4) < 1 ,$$

using property (c) of $r$. Hence, $z_i \in \mathsf{B}_{\|\cdot\|_{\tilde{z}}}(\tilde{z}, 1)$. Moreover, $\tilde{z} \in \mathsf{F}_h \subset \mathsf{A}_h$, and then it comes that $h < h_\star(z_i)$. Therefore, by combining property (d) of $r$ with **A**3, we have

(i) $\mathsf{V}_i \subset \mathsf{B}_i \subset \mathsf{B}_{\|\cdot\|_{z_i}}(z_i, \lambda r^\star(z_i)) \subset \mathrm{dom}_{\Phi_h}$,

(ii) the restriction of $\Phi_h$ to $\mathsf{V}_i$ is a $\mathrm{C}^1$-diffeomorphism such that $\Phi_h \circ \Phi_h = \mathrm{Id}$.

This last result provides proper assumptions on $\Phi_h$ to operate a change of variable in $I'_i$. We now define $\mathsf{G}_{1,h} = \Phi_h(\mathsf{V}_i \cap \Phi_h^{-1}(\mathsf{G}) \cap \mathsf{F}_h)$ and $\mathsf{G}_{2,h} = \Phi_h^{-1}(\mathsf{V}_i) \cap \mathsf{G} \cap \mathsf{F}_h$, and prove that $\mathsf{G}_{1,h} = \mathsf{G}_{2,h}$ in two steps.

(i) We first prove that $\mathsf{G}_{1,h} \subset \mathsf{G}_{2,h}$. Let $z \in \mathsf{G}_{1,h}$. Then, there exists $z' \in \mathsf{V}_i \cap \Phi_h^{-1}(\mathsf{G}) \cap \mathsf{F}_h$ such that $z = \Phi_h(z')$. Since $\Phi_h$ is an involution on $\mathsf{V}_i$, we have $\Phi_h(z) = z'$. Since $z' \in \mathsf{V}_i \cap \mathsf{A}_h$, it comes that $z \in \Phi_h^{-1}(\mathsf{V}_i) \cap \Phi_h^{-1}(\mathsf{A}_h)$. Moreover, $z' \in \Phi_h^{-1}(\mathsf{G}) \cap \Phi_h^{-1}(\mathsf{A}_h)$, and thus, $z \in \mathsf{G} \cap \mathsf{A}_h$. Then, we have $z \in \mathsf{G}_{2,h}$, which proves this first result.

(ii) We now prove that $\mathsf{G}_{2,h} \subset \mathsf{G}_{1,h}$. Let $z \in \mathsf{G}_{2,h}$. In particular, $z \in \mathsf{G}$. Then, there exists $j \in [K]$ such that $z \in \mathsf{V}_j$. Therefore, $z \in \mathsf{B}_j$ and thus $\|z - z_j\|_{z_j} < r(z_j) < 1$. Since $z \in \mathsf{A}_h$, we obtain that $h < h_\star(z_j)$ with the same computations as those written above. In particular, this proves with **A**3 that $\Phi_h$ is an involution on $\mathsf{V}_j$. Then, $z = \Phi_h(\Phi_h(z))$, with $\Phi_h(z) \in \mathsf{V}_i \cap \mathsf{A}_h$ and $\Phi_h(z) \in \Phi_h^{-1}(\mathsf{G}) \cap \Phi_h^{-1}(\mathsf{A}_h)$, since $z \in \mathsf{G} \cap \mathsf{A}_h$. Therefore, we have $z \in \mathsf{