# OpenReview forum: "Unbiased constrained sampling with Self-Concordant Barrier Hamiltonian Monte Carlo"
_NeurIPS.cc/2023/Conference — NeurIPS 2023 poster_

### Official Review · Reviewer_SWNi · 2023-06-25

**Soundness:** 3 good
**Presentation:** 2 fair
**Contribution:** 3 good
**Rating:** 6
**Confidence:** 2

**Summary:**

This paper propose a computational framework for sampling via Hamiltonian Monte Carlo via self-concordant barrier functions for the purposes of constrained sampling on suitably nice Riemannian manifolds, where the Riemannian metric defines the barrier function. They provide computational and theoretical guarantees on their procedure, which hinges on an "involution checking step", making it more theoretically sound than other papers on this matter, in particular this makes the procedure unbiased. The authors supplement their theoretical findings with computational experiments in both real-world and synthetic data.

**Strengths:**

Strong theoretical contributions relative to prior work --- kudos to the authors for fixing the discrepancies in Kook et al (2022a), and proposing their solution (note, I didn't read the proofs in this paper either, only skimmed, but the results appear reasonable. I thought the background section, etc, contained all the relevant information, and the related works section also provided ample context.

**Weaknesses:**

I think the presentation is a bit suboptimal, namely the paper is very dense and hard to read in some parts. Otherwise, I really enjoyed this paper. I think the technical contributions appears to be somewhat minimal, though surely it took a lot of work to justify the "involution checking step", but again, not an expert in this area.

I also understand that Kook et al. (2022b) provided some theoretical understanding of the algorithm from the previous paper --- would the authors comment a bit more on that? Do the theoretical results in that paper suffer from the same problems as in this one, or is everything remedied and this paper is concurrent with that one?



**Questions:**

- What is this [x,y] or x : y notation in lines 101-102 (and other places)? I was not able to parse this...

- I was also largely confused and unconvinced by the numerical experiments... In Table 1 it appears what the authors propose is about "as good" as CRHMC (as in, 50/50 chance it is better performing). I also cannot understand Fig 2 at all --- what does the y-axis mean, and what is the ground truth value the algorithms are trying to estimate? Is the goal to simply be as close to MALA as possible? Seems odd, but I'm far from an expert

**Limitations:**

Yes

---

> ### Author Rebuttal · Authors · 2023-08-08
>
> We thank the reviewer for their feedback and their positive evaluation.
>
> > What is this [x,y] or x : y notation in lines 101-102 (and other places)? I was not able to parse this...
>
> We refer to the general response for more details about our presentation. We hope that these explanations clarify our notation and we are ready to further update our paper if the reviewer has specific concerns regarding our terminology.
>
> > I think the technical contributions appears to be somewhat minimal, though surely it took a lot of work to justify the "involution checking step", but again, not an expert in this area.
>
> We acknowledge that our algorithmic modification is minimal compared to the algorithm of [1], but we strongly believe that the study of such implementation is of fundamental interest for the MCMC community. Indeed, in a similar way as [2], our contribution (i) highlights that the **use of well known implicit schemes for ODE integration** (see [3] for a complete reference) **combined with space constraints leads to asymptotic bias when it comes to their numerical implementation**, and (ii) proposes to solve it in a straightforward manner. More generally, we are convinced that our work is a first step for future work on designing implicit integration schemes for constrained sampling methods.
>
> > I also understand that Kook et al. (2022b) provided some theoretical understanding of the algorithm from the previous paper --- would the authors comment a bit more on that? Do the theoretical results in that paper suffer from the same problems as in this one, or is everything remedied and this paper is concurrent with that one?
>
> We refer to the general response for a detailed comparison with [4]. In particular, we show that [4] suffers from the same shortcomings as [1] when it comes to the assumption made on the implicit integrator.
>
> > I was also largely confused and unconvinced by the numerical experiments... In Table 1 it appears what the authors propose is about "as good" as CRHMC (as in, 50/50 chance it is better performing). I also cannot understand Fig 2 at all --- what does the y-axis mean, and what is the ground truth value the algorithms are trying to estimate? Is the goal to simply be as close to MALA as possible?
>
> We refer to the general response for more details on the setting of our experiments, including a comparison with [1] on real-world data and and a discussion on the setting chosen for synthetic data. In particular, experiments on real-world data are meant to show that CRHMC [1] and BHMC have roughly the same complexity, while BHMC implements an additive involution checking step (which may have an effect on the autocorrelation of the obtained Markov chain). Moreover, we explain that we take the estimators of $\int_{\mathsf{M}}\langle x, \mu \rangle \mathrm{d}\pi(x)$ given by MALA and IMH as ground truth for our experiments on synthetic datasets, where $\pi$ refers to the Gaussian target distribution with mean $\mu\in \mathbb{R}^d$. In this setting, we prove that the estimator given by BHMC is closer to this ground truth than the one given by CRHMC [1].
>
>
> [1] Sampling with Riemannian Hamiltonian Monte Carlo in a constrained space. Kook et al., 2022a.
>
> [2] Geometric numerical integration. Hairer et al., 2006.
>
> [3] Multiple projection Markov Chain Monte Carlo algorithms on submanifolds. Lelièvre et al, 2022.
>
> [4] Condition-number-independent convergence rate of Riemannian Hamiltonian Monte Carlo with numerical integrators. Kook et al., 2022b.

---

### Official Review · Reviewer_iZM7 · 2023-07-07

**Soundness:** 4 excellent
**Presentation:** 3 good
**Contribution:** 3 good
**Rating:** 6
**Confidence:** 3

**Summary:**

The paper proposes a new algorithm for sampling on a constrained space via a Hamiltonian Monte Carlo algorithm, which uses a Riemannian Manifold HMC algorithm, with the Hessian metric given by a self-concordant barrier. This generalizes and improves upon existing work in constrained sampling, while removing a source of bias by adding an “involution checking step”.

**Strengths:**

The intuition for the algorithm, including the choice of self-concordant barrier, is well-informed and intuitive, although it has appeared in some form in Kook et al. (2022). Thus, I will discuss many of the strengths and weaknesses in relation to the prior work of Kook et al. (2022).

This paper possesses two generalizations of Kook et al. Firstly, the primary problem setting in Kook et al. (Eq 11 of that paper) is less general than that in this paper. The proof for this method is also cleaner, and purports to avoid some technical issues present in the proof Kook et al. (I did not carefully verify this point). Furthermore, it avoids the problems of bias presented in Kook et al., which is carefully justified by a Theorem.

The method is clearly unbiased when evaluated on the hypercube and simplex, by a margin that is usually many standard deviations.

The broad details of the proof seem to be correct and well-written, although I did not check every detail.


**Weaknesses:**

No rigorous non-asymptotic theory was presented, although this is also true of Kook et al., likely because of the complexity of these algorithms.

It is difficult to draw firm conclusions from the real-world instances in the experiments, since the method does not show improvement in all/a clear majority of cases.

The notation in this paper is a bit cumbersome, particularly in the Appendix. Nonetheless this is understandable given the complexity of the algorithm, and the authors did a good job in reducing the notational burden within the main text.

To summarize, it was difficult to review all the details/proofs in this manuscript, given its length and technical nature. Nonetheless, I feel like it presents an important development within constrained sampling and merits publication. This informs my final score.


**Questions:**

How difficult is it to establish any non-asymptotic convergence results in this context? What would be the expected rates, and would they differ (in the order of dependencies) with Kook et al.?

As noted under “weaknesses”, the experiments on “real-world data” do not seem fully convincing. Can the authors elaborate further on why this would be the case, and if there was extensive tuning of the parameters involved? It seems from the Appendix that the parameters of both algorithms were not extensively tuned, which then begs the question whether this is really a fair comparison.

It is curious that a (constrained) Gaussian distribution is chosen for the synthetic data, when Kook et al. consider a uniform distribution instead. Their choice allows for some simple tests of uniformity. Does considering a uniform distribution make any appreciable difference in the output?

The discussion in Appendix J is quite useful and I think some more of these details could be also summarized in the main text.

It is my personal preference, but I believe text in the paper should not be highlighted. Use boldface or perhaps underlining to emphasize the text in Ll. 192-193 and in Algorithm 1.

L. 91 “motivates to” -> “motivates us to”

L. 170 “the definition domain” -> “the domain”

L. 213 “enables” -> “enables one/enables us”

The authors should be careful about the formatting of some equations in the Appendix, since they exceed the column width.


**Limitations:**

None beyond those raised in earlier sections.

---

> ### Author Rebuttal · Authors · 2023-08-08
>
> Thank you for your comments, we appreciate your acknowledgment of the paper’s merits.
>
> > No rigorous non-asymptotic theory was presented, although this is also true of Kook et al., likely because of the complexity of these algorithms.
>
> > How difficult is it to establish any non-asymptotic convergence results in this context? What would be the expected rates, and would they differ (in the order of dependencies) with Kook et al.?
>
> We believe that the non-asymptotic convergence results of [1] could be extended to our framework, i.e., comprising the involution checking step. However, such a study would be very technical and is outside of the scope of the current paper. We conjecture that the convergence rate obtained in [1] would also hold in our case. In particular, we do not expect a change in the dependency of this rate with respect to the hyperparameters of the algorithm.
>
> > It is difficult to draw firm conclusions from the real-world instances in the experiments, since the method does not show improvement in all/a clear majority of cases.
>
> > It is curious that a (constrained) Gaussian distribution is chosen for the synthetic data, when Kook et al. consider a uniform distribution instead. Their choice allows for some simple tests of uniformity. Does considering a uniform distribution make any appreciable difference in the output?
>
> We refer to the general response for more details on the setting of our experiments, including a comparison with [2] on real-world data and a discussion on the setting chosen for synthetic data. In particular, we explain that the experiments on real-world data are meant to show that the involution checking step does not hurt the complexity of BHMC, and justify the choice of a 'non-uniform' target distribution for synthetic datasets.
>
> > As noted under “weaknesses”, the experiments on “real-world data” do not seem fully convincing. Can the authors elaborate further on why this would be the case, and if there was extensive tuning of the parameters involved? It seems from the Appendix that the parameters of both algorithms were not extensively tuned, which then begs the question whether this is really a fair comparison.
>
> Regarding the choice of the hyperparameters except $\eta$ (including $h$, $\beta$, the number of iterations in the implicit solver..), we rely on the code provided by [2].  In particular, we do not fine-tune these hyperparameters for both CRHMC and BHMC. Since we are not provided with a truthful baseline due to the high-dimensional setting, we do not tune the parameter $\eta$ and set $\eta=10$ as in the experiments on toy datasets.
>
> > The notation in this paper is a bit cumbersome, particularly in the Appendix.
>
> > It is my personal preference, but I believe text in the paper should not be highlighted.
>
> We refer to the general response for more details about our presentation. We hope that these explanations clarify our notation and we are ready to further update our paper if the reviewer has specific concerns regarding our terminology.
>
> > To summarize, it was difficult to review all the details/proofs in this manuscript, given its length and technical nature.
>
> We are aware that our work is technical, and for completeness, we provide below a short description of our main theoretical results (Proposition 2 and Theorem 3), as well as a sketch of their proof. Proposition 2 justifies Assumption A3 made on the numerical integrator. Theorem 3 proves the reversibility of Algorithm 3. Below, we give more details on these results and provide a sketch of their proof.
>
> **Proposition 2**. In this proposition, we prove that in the self-concordant setting, for any starting point $z^0$, the implicit scheme $F_h=G_h \circ s$ is locally an involution as long as $h$ is small enough. To prove this result, we proceed as follows. We first prove that we can build a symplectic diffeomorphism $\gamma_h$ between $z^0$ and a solution $z^1 \in F_h(z^0)$ on a neighborhood of $z^0$ (see Lemma 23, 24-(a) and 25). This proves the first item of Proposition 2. Next, using Lemma 22, 24-(b) and 26, we prove that $F_h$ coincides with $\gamma_h$ on a given neighborhood of $z^0$. The main difficulty when establishing Proposition 2 comes from Lemma 22, which identifies the momentum $p^0$ used to go from $x^0$ to $x^1$ and shows that it is locally invertible as a perturbation of the identity. Proving this result requires computing higher order derivatives of the Riemannian metric and controlling their smoothness.
>
> **Theorem 3**. In this theorem, we prove the reversibility up to momentum reversal of the whole Markov kernel of Algorithm 3 (including momentum sampling, ODE integration, MH filter). The critical step of this procedure is to ensure that $\Phi_h$ preserves the volume form on some compact sets under Assumption A3, derived from Proposition 2. This is proved in Lemma 29. To do so, we have to restrict the cotangent bundle to the set of points $A_h$, where h is small enough so that $\Phi_h$ acts like an involution. Then, the proof consists in operating a change of variable on a partition of open subsets of a given compact set. The main difficulty in this proof is the control the compact sets on the domain of definition of $\Phi_h$ in order to perform the change of variable in Lemma 29.
>
>
> [1] Condition-number-independent convergence rate of Riemannian Hamiltonian Monte Carlo with numerical integrators. Kook et al., 2022b.
>
> [2] Sampling with Riemannian Hamiltonian Monte Carlo in a constrained space. Kook et al., 2022a.

---

> > ### Comment · Reviewer_iZM7 · 2023-08-20
> > **Response**
> >
> > I thank the authors for their extremely detailed response. I appreciate the explanation on why a non-uniform example was chosen and I believe that this work makes an important contribution in terms of identifying a true gap in Kook et al. For me, one major drawback of this paper continues to be the lack of non-asymptotic convergence results, which would be really necessary to make this paper fully self-contained. As a result, I will maintain my current score.

---

### Official Review · Reviewer_prii · 2023-07-09

**Soundness:** 3 good
**Presentation:** 2 fair
**Contribution:** 3 good
**Rating:** 6
**Confidence:** 4

**Summary:**

This paper introduces a "involution checking step" in Riemannian Hamiltonian Monte Carlo methods that ensures time reversibility of the resulting Markov chain. The numerical experiments demonstrate the competitiveness of the proposed algorithm with regard to the method proposed in Kook et al. (2022a).

---

The rebuttal addressed my quetions. I have raised the rating to 6.

**Strengths:**

1. The algorithms are novel.
2. The algorithms and analyses take practical numerical integrators into consideration.
3. Appendix J points out clearly the theoretical gaps in Kook (2022a).

**Weaknesses:**

1. **Comparison with existing work.** The most relevant previous work seems to be Kook et al. (2022a) and Kook et al. (2022b). The second one appeared later and focused on theoretical analyses, but this submission only discusses about the gaps in the analysis of the first one.
2. **Gap between theory and practice.** The algorithm that actually ensures time reversibility is Algorithm 3, but the numerical experiments are conducted with Algorithm 1.
3. **Imprecise statement about existing work.** In Ln. 304, it is claimed that the convex constraint set considered by Kook et al. (2022a) takes a specific form. But that is simply a special case of the class of constraint sets in Kook et al. (2022a).
4. **Presentation.** The presentation involves many terminologies without explanations. The paper may not be readable to a layperson.
    1. There are some highlights with unclear meanings.
    2. What do the vertical axes mean in Figure 2?
5. **Typos.**
    1. Ln. 115: Concordance -> self-concordace
    2. Ln. 118: polynomial -> polynomial-time
    3. What does the colon mean in (4)?
    4. Ln. 287: a involution -> an involution
    5. Some words in the reference list are not properly capitalized.

**Questions:**

1. How does this work compare with Kook et al. (2022b)?
1. Please address the gap between theory and practice in the weaknesses block.
1. What do the highlights in the submitted files mean?

**Limitations:**

This is a paper on theoretically rigorous algorithm design. The assumptions are explicitly stated.

---

> ### Author Rebuttal · Authors · 2023-08-08
>
> Thank you for taking the time to review our submission and for your constructive feedback.
>
> > 1. How does this work compare with Kook et al. (2022b)?
>
> We refer to the general response for a detailed comparison with [1]. In particular, we show that [1] suffers from the same shortcomings as [2] when it comes to the assumption made on the implicit integrator. We will add this detail in the section of the related work.
>
> > 2. Please address the gap between theory and practice in the weaknesses block.
>
> We are aware of this slight discrepancy between theory and practice, and will highlight this in the Discussion section as a limitation of our study. We emphasize that **directly analyzing Algorithm 1 as such is a very challenging task**. Indeed, the self-concordance properties are only defined locally, whereas deriving reversibility results require global control. More precisely, given a certain current state, we have to **ensure that $h$ is small enough** (depending on this current state) to obtain that the implicit integrator (or its numerical approximation) is locally an involution (see Proposition 2 and Assumption 3). However, in practice, $h$ is fixed by the user. Thus guaranteeing that Proposition 2 holds for any element of the constrained set and a fixed step size $h$ is difficult.
>
> To overcome this limitation, we study a modified version of Algorithm 1, given by Algorithm 3, where the involution checking step (ICS) is replaced by a stronger checking step involving the set $A_h$, *which implies ICS* (see Section 4.2) and for which we are able to state its reversibility. Although this algorithm may be implemented, this would come with a huge (and unrealistic) computational cost since verifying that $z \in A_h$ implies to find the minimal critical step-size $h_\star$ on a neighborhood of $z$. The question of the existence of a global step-size $h$ such that Algorithm 1 can be properly analyzed is not the topic of this paper and is left for future work.
>
> > 3. Imprecise statement about existing work
>
> Indeed, the setting presented in the introduction of [2] is a convex body $K$ provided with a self concordant barrier $\phi$, combined with a linear equality constraint $Ax=b$ (equation 1.1). We remark that this setting is still a **special case of ours** by rewriting this whole set as $K’$={$A^\dagger b +u \in K, u \in \text{Ker} A$}, which is associated with the self-concordant barrier $u \to \phi(A^\dagger b +u)$. We will clarify this result in the updated version of the paper.
>
> However, we emphasize that the **main case of interest in [2] is the case where $K$={$x: \ell < x <u$}**. Indeed, the authors state in their introduction that their algorithm is ‘[efficient] when $K$ is a product of convex bodies $K_i$, each with small dimension’, i.e., $\mathfrak{g}$ is a block-diagonal matrix. Indeed, the design of their numerical integrator heavily relies on the computation of the Cholesky decomposition of the matrix $A \mathfrak{g}^-1 A^T$, which is efficient in the case where $\mathfrak{g}$ is a (block-)diagonal matrix. In addition, through their paper, the authors consider the particular case where the $K_i$ are 1D, both in theory and in the experiments. The efficiency of the whole procedure in this specific case is justified in Appendix B.2.3. For these reasons, we highlighted this framework of [2] in the related work section.
>
> We will change our discussion regarding [2] (lines 302-306) as follows:
>
> 'On the other hand, Kook et al. (2022a) integrate the Hamiltonian dynamics via implicit schemes without any “involution checking step” in a similar self-concordant setting. In particular, they consider a convex body $K$ equipped with a self concordant barrier $\phi$, combined with a linear equality constraint $Ax=b$. This setting is a special case of our framework (see A1 and A2) by rewriting this whole set as $K’$={$A^\dagger b +u \in K, u \in \text{Ker} A$}, equipped with the self concordant barrier $u \to \phi(A^\dagger b +u)$. Kook et al. (2022a) provide an efficient implementation of their algorithm in the case of a convex bounded manifold of the form $\mathsf{M}$={$x \in \mathbb{R}^d : Ax = b, \ell < x < u$}'.
>
> > 4. Presentation
>
> We refer to the general response for more details about our presentation. We hope that these explanations clarify our notation and we are ready to further update our paper if the reviewer has specific concerns regarding our terminology.
>
>
> [1] Condition-number-independent convergence rate of Riemannian Hamiltonian Monte Carlo with numerical integrators. Kook et al., 2022b.
>
> [2] Sampling with Riemannian Hamiltonian Monte Carlo in a constrained space. Kook et al., 2022a.

---

> > ### Comment · Reviewer_prii · 2023-08-14
> > **Thanks**
> >
> > The rebuttal addressed my quetions. I have raised the rating to 6.

---

### Official Review · Reviewer_nmMp · 2023-07-10

**Soundness:** 3 good
**Presentation:** 3 good
**Contribution:** 3 good
**Rating:** 7
**Confidence:** 3

**Summary:**

The authors resolve an existing bias with naive manifold versions of HMC by adding an involution checking step, and therefore leading the new HMC algorithm to be unbiased.

I'm not super familiar with this problem, and I would like the authors to explain some of the technical details, as the proofs are quite involved for a conference review.

**Strengths:**

The authors proposes a solution to an existing bias in manifold HMC, both justifying theoretically and proposing convincing empirical simulations.

**Weaknesses:**

The technical details are quite involved and I believe could be explained better.

**Questions:**

While I have many questions, I am still open to raising my score once these questions are adequately addressed.

1. When you say there's no guarantee that $G_h$ has a unique solution, do you mean that (1) there are cases with multiple solutions, or (2) there are cases with no solutions, or both? If possible, can you also provide an intuitive example where this failure occurs?

2. Can the authors explain to me the main intuitions behind why the naive implementations of Kook et al. (2022a) leads to a biased stationary measure? In particular, why does the Metropolis accept-reject step not fix this bias?

3. To add to the previous question, on an intuitive level, how does this involution check fix this issue?

4. It appears that this is not a problem when we can simulate the ideal Hamiltonian dynamics, is this related to the fact that the domain of flow is the entire cotangent bundle $T^* M$?

5. The proofs are quite involved. Can you explain to me on a high level (a) how does the proof work, and (b) where do the complicated calculations mostly arise from?

---

> ### Author Rebuttal · Authors · 2023-08-08
>
> Thank you for your comment and thoughtful questions.
>
> ### Answer to Question 1.
>
> We refer to the general response for a discussion on the number of solutions of the implicit mapping $G_h$. We provide a one-dimensional example, where $G_h$ may admit 0,1 or 2 solutions for the implicit update of the momentum, depending on the initial condition and $h$.
>
> ### Answer to Questions 2 & 3.
>
> The main reason for the bias in the implementation of [1] is that **there is no guarantee that the numerical integrator used to solve the implicit scheme (Alg. 3 in [1]) is involutive**. This is however essential to derive theoretical results relying on the Metropolis Hastings (MH) filter in the Hamiltonian setting. Indeed, to obtain the reversibility of the Markov kernel corresponding to the Hamiltonian integration step followed by the MH step (see Lemma 30 in our paper), we need to show that for any compactly supported function $f$
>
> $ \int \bar{\pi}(z)   a(R^{\Phi}_h(z) \mid z)f(z,R^{\Phi}_h(z)) \mathrm{d} z= \int \bar{\pi}(z)   a(R^{\Phi}_h(z) \mid z)f(R^{\Phi}_h(z),z) \mathrm{d} z$  ,  (a)
>
> where $\bar \pi$ is the extended target distribution, $a(z' \mid z)=\min(1, \exp(-H(z')+H(z))$ is the MH acceptance filter and
> $R^{\Phi}_h$ is the numerical integrator of the Hamiltonian $H$ with a step-size $h$. Since $R^{\Phi}_h= T_h \circ \Phi_h \circ T_h$, where $T_h$  is an involutive function, and $\Phi_h$ is the numerical approximation of the implicit scheme $G_h \circ s$ (see Section 3.1 in our paper), showing (a) amounts to prove that
>
> $ \int   g(z,R^{\Phi}_h(z)) \mathrm{d} z = \int g(R^{\Phi}_h(z),z) \mathrm{d} z$ , (b)
>
> for a certain function $g$. **A sufficient condition to obtain (b) is thus to ensure that $\Phi_h$ is an involution**. However, this condition is not satisfied in the CRHMC [1], and **the application of the MH filter does not solve this issue**.
>
> In our setting, we directly enforce this condition through the involution checking step (up to some numerical relaxation). To support our point, we would like to mention the paper [2], where the authors also enforce an involution condition in a slightly different fashion than ours. Namely, given a current state $z$, they compute the whole set of solutions $\Phi_h(z)$, choose $z’\in \Phi_h(z)$ according to some probability distribution defined on this set and then verify that $z \in \Phi_h(z’)$ (see line 7, Alg. 2 in [2]).
>
> ### Answer to Question 4.
>
> In the case where the Hamiltonian flow is well defined for all times, the domain of the flow is indeed the entire cotangent bundle but this is not sufficient to ensure the reversibility of HMC. The reversibility actually comes from the fact that the flow is involutive (up to momentum reversal) by definition of the Hamiltonian.
>
> However, in our manifold setting, there is one subtlety that needs to be checked, namely that the Hamiltonian flow does not leave the cotangent bundle in finite time. As detailed in Proposition 14, there is indeed no guarantee that this flow is defined for all times, due to the ill-conditioned behavior of the dynamics near the boundary of the manifold. Therefore, given a time horizon $h>0$, we restrict the cotangent bundle to the points where the flow is defined (see the definition of $E_h$ and $\bar{E}_h$ in Appendix D) on the time interval $[0,h]$. Then, we prove that **the Hamiltonian flow is involutive up to momentum reversal on this subset (see Lemma 16)**, which is exactly the property needed to derive the reversibility of the algorithm with the ideal dynamics. Hence, we do not need to perform an *involution check* in this case. As a result, we introduce c-BHMC in Appendix D, where **the only checking step verifies that the flow is well defined on $[0,h]$, starting at the current state**.
>
> ### Answer to Question 5.
>
> Our main theoretical results are Proposition 2 and Theorem 3. Proposition 2 justifies Assumption A3 made on the numerical integrator. Theorem 3 proves the reversibility of Algorithm 3. Below, we give more details on these results and provide a sketch of their proof.
>
> **Proposition 2**. In this proposition, we prove that in the self-concordant setting, for any starting point $z^0$, the implicit scheme $F_h=G_h \circ s$ is locally an involution as long as $h$ is small enough. To prove this result, we proceed as follows. We first prove that we can build a symplectic diffeomorphism $\gamma_h$ between $z^0$ and a solution $z^1 \in F_h(z^0)$ on a neighborhood of $z^0$ (see Lemma 23, 24-(a) and 25). This proves the first item of Proposition 2. Next, using Lemma 22, 24-(b) and 26, we prove that $F_h$ coincides with $\gamma_h$ on a given neighborhood of $z^0$. The main difficulty when establishing Proposition 2 comes from Lemma 22, which identifies the momentum $p^0$ used to go from $x^0$ to $x^1$ and shows that it is locally invertible as a perturbation of the identity. Proving this result requires computing higher order derivatives of the Riemannian metric and controlling their smoothness.
>
> **Theorem 3**. In this theorem, we prove the reversibility up to momentum reversal of the whole Markov kernel of Algorithm 3 (including momentum sampling, ODE integration, MH filter). The critical step of this procedure is to ensure that $\Phi_h$ preserves the volume form on some compact sets under Assumption A3, derived from Proposition 2. This is proved in Lemma 29. To do so, we have to restrict the cotangent bundle to the set of points $A_h$, where h is small enough so that $\Phi_h$ acts like an involution. Then, the proof consists in operating a change of variable on a partition of open subsets of a given compact set. The main difficulty in this proof is the control the compact sets on the domain of definition of $\Phi_h$ in order to perform the change of variable in Lemma 29.
>
>
> [1] Sampling with Riemannian Hamiltonian Monte Carlo in a constrained space. Kook et al., 2022a
>
> [2] Multiple projection Markov Chain Monte Carlo algorithms on submanifolds. Lelièvre et al. 2022.

---

> > ### Comment · Reviewer_nmMp · 2023-08-12
> > **Response**
> >
> > Hi Everyone,
> >
> > Thank you for the response. Let me start by saying I think the 1D example is incredibly convincing, and this situation finally made sense for me for the first time. The rest of my questions are also addressed, albeit I would hope the authors consider rewriting some portions of the paper, and maybe add some more intuitive content in the extra page (if accepted).
> >
> > I will raise my score to 7.

---

### Author Rebuttal · Authors · 2023-08-08

We thank the reviewers for their insightful comments and are encouraged by their positive feedback regarding the proposition, soundness, and clarity of our work. We provide detailed responses to each reviewer but summarize here their main feedback.

### 1. Comparison with [1] and counterexample

In [1], Kook et al. analyze the convergence rate of time-discretized Riemannian Hamiltonian Monte Carlo (RHMC) in a self-concordant setting (i.e., the Riemannian metric is the Hessian of the self-concordant barrier of the manifold), either relying on the Implicit midpoint integrator (Alg. 2 in [1]) or using the Generalized Leapfrog (GL) integrator (Alg. 3 in [1]). Both of these integrators are implicit, and in practice, must be replaced by numerical schemes such as fixed-point procedure. Crucially, **in [1], the authors assume that these implicit integrators admit a unique solution given any starting point and any step-size h**. However, proving such a statement in the self-concordant setting is highly non-trivial and might not even be true, as we explain below with a simple example.

In our paper, we do not make the assumption from [1] on the GL integrator, and rather consider a numerical solver which outputs a unique approximate solution to this implicit scheme. By doing so, **our analysis is meant to be as close as possible to the practical implementation of RHMC**. In this case, the involution checking step (ICS) is needed to ensure the measure preservation property in RHMC.

Therefore, **the theoretical (and practical) limitation about the reversibility of CRHMC [5] is still an issue in [1]**. To go further, we show below that the assumption made by [1] and [5] may not hold in the case of the GL integrator, defined in (6) in our paper.

Consider the 1-dimensional setting $\mathsf{M}=(-\infty, 0)$. In this case, $\mathfrak{g}(x)=1/x^2$. Let $h>0$, $X_0\in \mathsf{M}$. Assume that $\beta=1$, and consider the first iteration of BHMC. We have $\tilde{P}_1 \sim \mathrm{N}(0, 1/X_0^2)$ and $P_1^{(0)}=\tilde{P}_1-\frac{h}{2}\partial_x H_1(X_0,\tilde{P}_1)$, where $H_1$ is defined in Section 3.1. As mentioned in our paper, $\partial_x H_1$ does not depend on the momentum variable, and thus, $P_1^{(0)} \sim \mathrm{N}(\mu, 1/X_0^2)$, where $\mu=-\frac{h}{2}\partial_x H_1(X_0,\tilde{P}_1)$. Using (5) in our paper, the implicit equation in $P_1^{(1/2)}$ in the GL integrator is actually a polynomial equation of degree 2: $\frac{h}{2}X_0 (P_1^{(1/2)})^2 + P_1^{(1/2)} -P_1^{(0)}=0$.

Denote $\Delta=1+2 h X_0 P_1^{(0)}$. Then, whenever $P_1^{(0)}> -1/(2 h X_0)$, i.e., $\Delta < 0$, this equation admits no solution. Recalling that $P_1^{(0)} \sim \mathrm{N}(\mu, 1/X_0^2)$, this event occurs with positive probability, thus violating the setting of [1] and [5].

### 2. Details on the experiments

**Synthetic data**. In this setting, the target distribution $\pi$ is a constrained Gaussian distribution with mean $\mu=(10 / \sqrt{d-1}) *(0, \sqrt{d-1}, 1,…,1)^T$, variance $1$ and the domain is the hypercube or the simplex. This choice of a ‘non-uniform’ distribution is motivated by the experimental observation that the importance of the reversibility condition is **best revealed when the mass of $\pi$ is not evenly distributed on the boundary of the domain**. The choice of the target quantity $A=\int_{\mathsf{M}} \langle x, \mu \rangle \mathrm{d}\pi(x)$  is arbitrary but is enough to highlight the bias in CRHMC [5], which is corrected when using BHMC. In our experiments, we aim at computing $A$ with the best accuracy. To do so, we consider as ground truth the estimators obtained with competitive algorithms such as MALA (for the hypercube) and IMH (for the simplex). Note that these two algorithms are run 10 times longer than CRHMC [5] and BHMC (i.e., $10^6$ iterations) to ensure accurate unbiased estimation. Finally, each algorithm  (CRHMC [5], BHMC, IMH, MALA) is run 10 times to obtain confidence intervals. We present boxplots in Figure 2 (here the vertical axis refers to the estimated target quantity).

**Real-world data**. In this setting, **we do not have access to a realistic baseline that yields an unbiased estimator**, because the dimension of the polytope is too high and running MALA or IMH would be too costly. Hence, the goal of this experiment is instead to highlight that the ICS does not hurt the convergence properties of the algorithm. More precisely, we show that our method is comparable with the one from [5] in terms of time complexity to obtain independent samples. While it is clear that adding an ICS implies a tradeoff between accuracy and complexity of the method, our results demonstrate that this tradeoff does not penalize BHMC in the considered settings.

### 3. Precisions on the notation
We precise the following notations: (i) for any vectors $u$ and $v$, $D\mathfrak{g}(x)[u,v]$ is the vector $u^T D\mathfrak{g}(x) v$, (ii) $\mathfrak{g}(x)^{−1} : D\mathfrak{g}(x)$ stands for the vector $Tr(\mathfrak{g}(x)^{-1} D\mathfrak{g}(x))$. Regarding the yellow highlighted parts: these highlights are used in Alg. 1 to emphasize our main algorithmic contribution, compared to the implementation of [5], namely the ICS. We will add a sentence to explain this in the updated version of the manuscript. In the appendix, highlights are used in Alg. 2 to emphasize our contribution compared [5] in the case of continuous RHMC and are also used in Alg. 3 to emphasize the differences with Alg. 1.

[1] Condition-number-independent convergence rate of Riemannian Hamiltonian Monte Carlo with numerical integrators. Kook et al., 2022b.

[2] A family of MCMC methods on implicitly defined manifolds. Brubaker et al., 2012.

[3] Monte Carlo on manifolds: sampling densities and integrating functions. Zappa et al., 2018.

[4] Multiple projection Markov Chain Monte Carlo algorithms on submanifolds. Lelièvre et al., 2022.

[5] Sampling with Riemannian Hamiltonian Monte Carlo in a constrained space. Kook et al., 2022a.

---

### Decision · Program_Chairs · 2023-09-21

**Decision:**

Accept (poster)

**Comment:**

The paper considers the problem of sampling on a constrained space using a variant of Hamiltonian Monte Carlo algorithm. Authors suggest using a Riemannian Manifold HMC algorithm with the Hessian metric given by a self-concordant barrier. Their framework improves the existing work in the context of constrained sampling, removing the bias of the algorithm through novel techniques.

The paper is reviewed by 4 expert reviewers with the following Rating/Confidence scores: 6/4, 6/2, 6/3, 7/3. All 4 reviewers agree that the paper is well written and has significant contributions. I also agree with the reviewers and recommend accepting this paper.